# Bayesian inference of kinetic schemes for ion channels by Kalman filtering

**Jan L Münch[1]\*, Fabian Paul[2], Ralf Schmauder[1], Klaus Benndorf[1]\***

[1]Institut für Physiologie II, Universitätsklinikum Jena, Friedrich Schiller University Jena, Jena, Germany; [2]Department of Biochemistry and Molecular Biology, University of Chicago, Chicago, United States

**Abstract** Inferring adequate kinetic schemes for ion channel gating from ensemble currents is a daunting task due to limited information in the data. We address this problem by using a parallelized Bayesian filter to specify hidden Markov models for current and fluorescence data. We demonstrate the flexibility of this algorithm by including different noise distributions. Our generalized Kalman filter outperforms both a classical Kalman filter and a rate equation approach when applied to patch-clamp data exhibiting realistic open-channel noise. The derived generalization also enables inclusion of orthogonal fluorescence data, making unidentifiable parameters identifiable and increasing the accuracy of the parameter estimates by an order of magnitude. By using Bayesian highest credibility volumes, we found that our approach, in contrast to the rate equation approach, yields a realistic uncertainty quantification. Furthermore, the Bayesian filter delivers negligibly biased estimates for a wider range of data quality. For some data sets, it identifies more parameters than the rate equation approach. These results also demonstrate the power of assessing the validity of algorithms by Bayesian credibility volumes in general. Finally, we show that our Bayesian filter is more robust against errors induced by either analog filtering before analog-to-digital conversion or by limited time resolution of fluorescence data than a rate equation approach.

**\*For correspondence:**
jan.muench@med.uni-jena.de
(JLM);
KLAUS.BENNDORF@med.uni-jena.de (KB)

**Competing interest:** The authors declare that no competing interests exist.

## Editor's evaluation

The authors develop a Bayesian approach to modeling signals arising from ensembles of ion channels that can incorporate multiple simultaneously recorded signals such as fluorescence and ionic current. For simulated data from a simple ion channel model where ligand binding drives pore opening, they show that their approach enhances parameter identifiability and/or estimates of parameter uncertainty over more traditional approaches. The developed approach provides a valuable tool for modeling macroscopic time series data including data with multiple observation channels.

## Introduction

Ion channels are essential proteins for the homeostasis of an organism. Disturbance of their function by mutations often causes severe diseases, such as epilepsy (*Oyrer et al., 2018*; *Goldschen-Ohm et al., 2010*), sudden cardiac death (*Clancy and Rudy, 2001*), or sick sinus syndrome (*Verkerk and Wilders, 2014*) indicating a medical need (*Goldschen-Ohm et al., 2010*) to gain further insight into the biophysics of ion channels. The gating of ion channels is usually interpreted by kinetic schemes which are inferred either from macroscopic currents with rate equations (REs) (*Colquhoun and Hawkes, 1995b*; *Celentano and Hawkes, 2004*; *Milescu et al., 2005*; *Stepanyuk et al., 2011*; *Wang et al., 2012*) or from single-channel currents using dwell time distributions (*Neher and Sakmann, 1976*; *Colquhoun et al., 1997a*; *Horn and Lange, 1983*; *Qin et al., 1996*; *Epstein et al., 2016*; *Siekmann*

*et al., 2016*) or hidden Markov models (HMMs) (*Chung et al., 1990*; *Fredkin and Rice, 1992*; *Qin et al., 2000*; *Venkataramanan and Sigworth, 2002*). A HMM consists of a discrete set of metastable states. Changes of their occupation occur as random events over time. Each state is characterized by transition probabilities, related to transition rates, and a probability distribution of the observed signal (*Rabiner, 1989*). It is becoming increasingly clear that the use of Bayesian statistics in HMM estimation constitutes a major advantage (*Ball et al., 1999*; *De Gunst et al., 2001*; *Rosales et al., 2001*; *Rosales, 2004*; *Gin et al., 2009*; *Siekmann et al., 2012*; *Siekmann et al., 2011*; *Hines et al., 2015*; *Sgouralis and Pressé, 2017b*; *Sgouralis and Pressé, 2017a*; *Kinz-Thompson and Gonzalez, 2018*). In ensemble patches, simultaneous orthogonal fluorescence measurement of either conformational changes (*Zheng and Zagotta, 2000*; *Taraska and Zagotta, 2007*; *Taraska et al., 2009*; *Bruening-Wright et al., 2007*; *Kalstrup and Blunck, 2013*; *Kalstrup and Blunck, 2018*; *Wulf and Pless, 2018*) or ligand binding itself (*Biskup et al., 2007*; *Kusch et al., 2010*; *Kusch et al., 2011*; *Wu et al., 2011*) has increased insight into the complexity of channel activation.

Currently, a Bayesian estimator that can collect information from cross-correlations and time correlations inherent in multi-dimensional signals of ensembles of ion channels is still missing. Traditionally, macroscopic currents are analyzed with solutions of REs which yield a point estimate of the rate matrix or its eigenvalues (*Colquhoun et al., 1997a*; *Sakmann and Neher, 2013*; *d'Alcantara et al., 2002*; *Milescu et al., 2005*; *Wang et al., 2012*) if they are fitted to the data. The RE approach is based on a deterministic differential equation derived by averaging the chemical master equation (CME) for the underlying kinetic scheme (*Kurtz, 1972*; *Van Kampen, 1992*; *Jahnke and Huisinga, 2007*). Its accuracy can be improved by processing the information contained in the intrinsic noise (stochastic gating and binding) (*Milescu et al., 2005*; *Munsky et al., 2009*). Nevertheless, all deterministic approaches do not use the information of the time- and cross-correlations of the intrinsic noise. These deterministic approaches are asymptotically valid for an infinite number of channels. Thus, a time trace with a finite number of channels contains, strictly speaking, only one independent data point. Previous rigorous attempts to incorporate the autocorrelation of the intrinsic noise of current data into the estimation (*Celentano and Hawkes, 2004*) suffer from cubic computational complexity (*Stepanyuk et al., 2011*) in the amount of data points, rendering the algorithm non-optimal or even impractical for a Bayesian analysis of larger data set. To understand this, note, that a maximum likelihood optimization (ML) usually takes several orders of magnitude fewer likelihood evaluations to converge compared to the number of posterior evaluations when one samples the posterior. One Monte Carlo iteration (*Betancourt, 2017*) evaluates the posterior distribution and its derivatives many times to propose one sample from the posterior. Stepanyuk suggested an algorithm (*Stepanyuk et al., 2011*; *Stepanyuk et al., 2014*) which derives from the algorithm of *Celentano and Hawkes, 2004* but evaluates the likelihood quicker. Under certain conditions, Stepanyuk's algorithm can be faster than the Kalman filter (*Moffatt, 2007*). The algorithm by *Milescu et al., 2005* achieves its superior computation time efficiency at the cost of ignoring the time correlations of the fluctuations. A further argument for our approach, independent of the Bayesian context, is investigated in this paper: The KF is the minimal variance filter (*Anderson and Moore, 2012*). Instead of strong analog filtering of currents to reduce the noise, but with the inevitable signal distortions (*Silberberg and Magleby, 1993*), we suggest to apply the KF with higher analyzing frequency on minimally filtered data.

On the one hand, a complete HMM analysis (forward algorithm) would deliver the most exact likelihood of macroscopic data. On the other hand, the computational complexity of the forward algorithm limits this type of analysis in ensemble patches to no more than a few hundred channels per time trace (*Moffatt, 2007*). To tame the computational complexity (*Jahnke and Huisinga, 2007*), we approximate the solution of the CME with a Kalman filter (KF), thereby remaining in a stochastic framework *Kalman, 1960*. This allows us to explicitly model the time evolution of the first two moments (mean value and covariance matrix) of the probability distribution of the hidden channel states. Notably, for linear (first or pseudo) Gaussian system dynamics, the KF is optimal in producing a minimal prediction error for the mean state. KFs have been used previously in several protein expression studies which also demonstrate the connection of the KF to the linear noise approximation (*Komorowski et al., 2009*; *Finkenstädt et al., 2013*; *Fearnhead et al., 2014*; *Folia and Rattray, 2018*; *Calderazzo et al., 2019*; *Gopalakrishnan et al., 2011*).

Our approach generalizes the work of *Moffatt, 2007* by including state-dependent fluctuations such as open-channel noise and Poisson noise in additional fluorescence data. A central technical

## Box 1. Phenomenological difference between an RE approach and our Bayesian filter

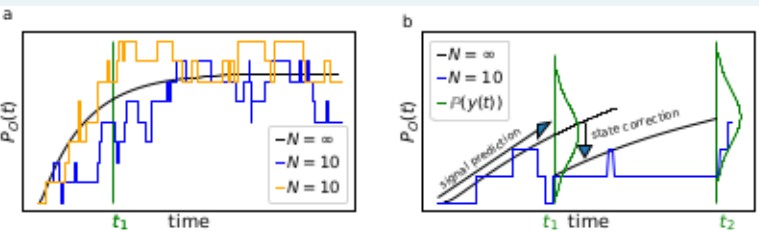

**Box 1—figure 1.** The First-order Markov property of the data requires a recursive prediction of the signal (by the model) and correction (by the data) scheme of the algorithm.
(**a**) Idealized patch-clamp (PC) data in the absence of instrumental noise for either ten (colored) or an infinite number of channels generating the mean time trace (black). The fluctuations with respect to the mean time trace (black) reveal autocorrelation (**b**) Conceptual idea of the Kalman Filter (KF): the stochastic evolution of the ensemble signal is predicted and the prediction model updated recursively.

Two major problems for parameter inference for the dynamics of the ion channel ensemble $\mathbf{n}(t)$ are: (I) that currents are only low-dimensional observations (e.g. one dimension for patch-clamp or two for cPCF) of a high-dimensional process (dimension being the number of model states) blurred by noise and (II) the fluctuations due to the stochastic gating and binding process cause autocorrelation in the signal. Traditional analyses for macroscopic PC data (and also for related fluorescence data) by the RE approach, e.g. *Milescu et al., 2005* ignores the long-lasting autocorrelations of the deviations (*Box 1—figure 1a*) blue and orange curves from the mean time trace (black) that occur in real data measured from a *finite* ensemble. To account for the autocorrelation in the signal, an optimal prediction (*Box 1—figure 1b*) of the future signal distribution $\mathbb{P}(y(t_2))$ should use the measurement $y(t_1)$ from the current time step $t_1$ to update the belief about the underlying $\mathbf{n}(t_1)$. Based on stochastic modelling of the time evolution of the channel ensemble, it then predicts $\mathbb{P}(y(t_2))$.

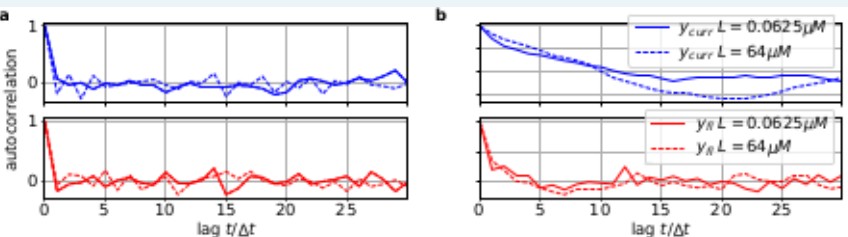

**Box 1—figure 2.** The residuals between model prediction and data reveal long autocorrelations if the analysis algorithm ignores the first-order Markov property.
(**a**) Autocorrelation of the residuals $r_I$ of two ligand concentrations of currents (blue) and of the fluorescence (red) after the data have been analyzed with the KF approach. (**b**) autocorrelation of $r_I$ after analysing with he RE approach.
To demonstrate the difference how the two algorithms analyze the data, we compute the autocorrelation of the residuals of the data. After the analysis with either the RE approach or the KF, we can construct from the model with the mean predicted signal $\mathbf{H}\mathbb{E}[\mathbf{n}(t_i)]$ (see Eq. 4 for the definition of $\mathbf{H}$) and the predicted standard deviation $\sqrt{\mathrm{var}[y(t_i)]}$ the normalized residual time trace of the data which are defined as

$$r(t_i) := \frac{y(t_i) - \left(\mathbf{H}\mathbb{E}[\mathbf{n}(t_i)]\right)}{\sqrt{\mathrm{var}[y(t_i)]}}.$$

(1)

Filtering (fitting) with the KF (given the true kinetic scheme) one expects to find a white-noise process for the residuals. Plots of the autocorrelation function of both signal components (*Box 1—figure 2a*) confirms our expectation. The estimated autocorrelation vanishes after one multiple of the lag time (the interval between sampling points), which means that the residuals are indeed a white-noise process. In contrast, the residuals derived from the RE approach (*Box 1—figure 2b*) display long lasting periodic autocorrelations.

difficulty which we solved is that due to the state-dependent noise the central Bayesian update equation loses its analytical solution. We derived an approximation which is correct for the first two moments of the probability distributions. Stochastic rather than deterministic modeling is generally preferable for small systems or non-linear dynamics (*Van Kampen, 1992*; *Gillespie and Golightly, 2012*). However, even with simulated data of unrealistic high numbers of channels per patch (more than several thousands within one patch), the KF outperforms the deterministic approach in estimating the model parameters. *Moffatt, 2007* already demonstrated the advantage of the KF to learn absolute rates from time traces at equilibrium. Like all algorithms that estimate the variance and the mean (*Milescu et al., 2005*) the KF can infer the number of channels $N_{ch}$ for each time trace, the single-channel current   and analogous in optical recordings the mean number $\lambda_b$ of photons from bound ligands per recorded frame. To select models and to identify parameters, stochastic models are formulated within the framework of Bayesian statistics where parameters are assigned uncertainties by treating them as random variables (*Hines, 2015*; *Ball, 2016*). In contrast, previous work on ensemble currents combined the KF only with ML estimation (*Moffatt, 2007*). Difficulties in treating simple stochastic models by ML approaches in combination with the KF (*Auger-Méthé et al., 2016*), especially with non-observable dynamics, justify the computational burden of Bayesian statistics. Bayesian statistics has an intuitive way to incorporate soft or hard constrains from diverse sources of prior information. Those sources include mathematical prerequisites, other experiments, simulations or theoretical assumptions. They are applied as additional model assumptions by a prior probability distribution over the possible parameter space. Hence, knowledge of the model parameters prior to the experiment are correctly accounted for in the analyzes of the new data. Alternatively, some of these benefits of prior knowledge can be incorporated by penalized maximum likelihood (*Salari et al., 2018*; *Navarro et al., 2018*). Bayesian inference provides outmatching tools for modeling over point estimates: First, the Bayesian approach is still applicable in situations where parameters are not identifiable (*Hines et al., 2014*; *Middendorf and Aldrich, 2017b*) or posteriors are non-Gaussian, whereas ML fitting ceases to be valid (*Calderhead et al., 2013*; *Watanabe, 2007*). Second, a Bayesian approach provides superior model selection tools for singular models such as HMMs or KFs Gelman et al. (2014). Third, Bayesian statistics has a correct uncertainty quantification (*Gillespie and Golightly, 2012*) based on the data and the prior for the statistical problem. In contrast, ML or maximum posterior approaches lack uncertainty quantification based on one data set (*Joshi et al., 2006*). Only under optimal conditions their uncertainty quantification becomes equivalent to Bayesian credibility volumes (*Jaynes and Kempthorne, 1976*). This study focuses on the effects on the posterior due to formulating the likelihood via a KF instead of an RE approach and the benefits of adding a second dimension of observation. We consider the performance of our algorithm against the gold standards in four different aspects: (I) The relative distance of the posterior to the true values, (II) the uncertainty quantification, here in the form of the shape of the posterior, (III) parameter identifiability, and (IV) robustness against typical misspecifications of the likelihood (such as ignoring that currents are filtered or that the integration time of fluorescence data points is finite) of real experimental data.

## Results and discussion
### Simulation of ligand-gated ion-channel data

Here we treat an exemplary ligand-gated channel with two ligand binding steps and one open-closed isomerization described by an HMM (see *Figure 1a*). For this model, confocal patch-clamp fluorometry (cPCF) data were simulated: time courses of ligand binding and channel current upon concentration jumps were generated (see Appendix 5 and Materials and methods section). Idealized example

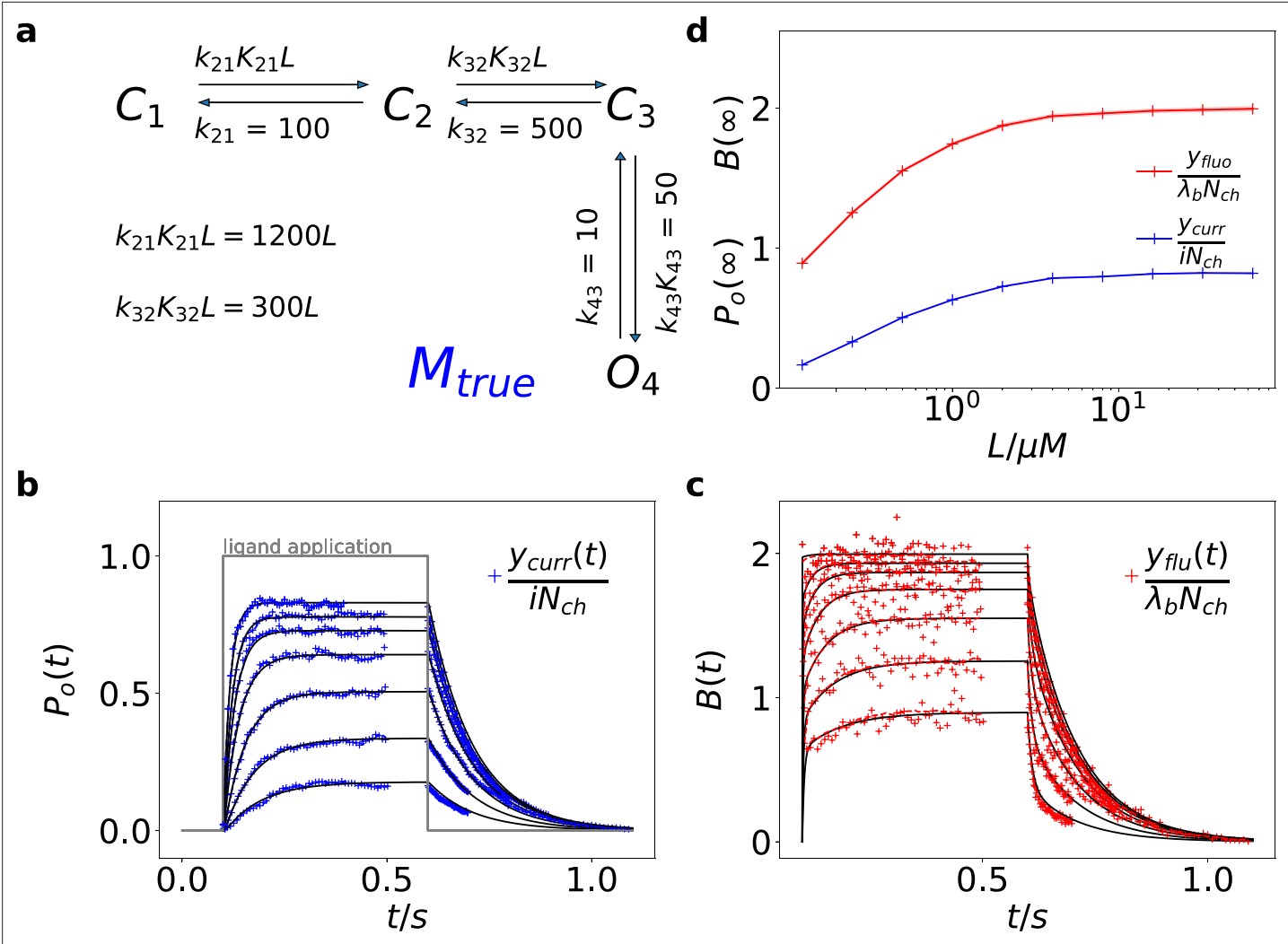

**Figure 1.** Kinetic scheme and simulated data. (**a**) The Markov state model (kinetic scheme) consists of two binding steps and one opening step. The rate matrix $\mathbf{K}$ is parametrized by the absolute rates $k_{ij}$, the ratios $K_{ij}$ between on and off rates (i.e. equilibrium constants) and $L$, the ligand concentration in the solution. The units of the rates are $\mathrm{s}^{-1}$ and $\mu M^{-1}\mathrm{s}^{-1}$, respectively. The liganded states are $C_2$, $C_3$, $O_4$. The open state $O_4$ conducts a mean single-channel current $i = 1$. Note, that absolute magnitude of the single channel current is irrelevant regarding this study what matters is its relative magnitude compared with $\sigma_{\mathrm{op}}$ and $\sigma_{\mathrm{ex}}$. Simulations were performed with 10 kHz or 100 kHz (for **Figures 11 and 12**) sampling, KF analysis frequency $f_{ana}$ for cPCF data is in the range of (200-500) Hz while pure current data is analyzed at 2-5 kHz. (**b-c**) Normalized time traces of simulated relaxation experiments of ligand concentration jumps with $N_{ch} = 10^3$ channels, $\lambda_b = 0.375$ mean photons per bound ligand per frame and single-channel current $i = 1$, open-channel noise with $\sigma_{\mathrm{op}}^2 = 0.1i^2$ and an instrumental noise with the variance $\sigma_{\mathrm{m}}^2 = i^2$. The current $y_{curr}$ and fluorescence $y_{flu}$ time courses are calculated from the same simulation. For visualization, the signals are normalized by the respective median estimates of the KF. The black lines are the theoretical open probabilities $P_o(t)$ and the average binding per channel $B(t)$ for $N_{ch} \to \infty$ of the used model. Typically, we used 10 ligand concentrations which are (0.0625, 0.125, 0.25, 0.5, 1, 2, 4, 8, 16, 64) $\mu M$. d, Equilibrium binding and open probability as function of the ligand concentration $L$.

The online version of this article includes the following source data for figure 1:

**Source data 1.** The example data is provided.

data with added white noise are shown in **Figure 1b–d**. We added realistic instrumental noise to the simulated data (see Appendix 5). A qualitative description of the statistical problem that needs to be addressed when modeling time series data such as the simulated is outlined in Box. 1.

## Kalman filter derived from a Bayesian filter

Here and in the Materials and methods section, we derive the mathematical tools to account correctly for the stochastic Markov dynamics of single molecules in the fluctuations of macroscopic signals. The

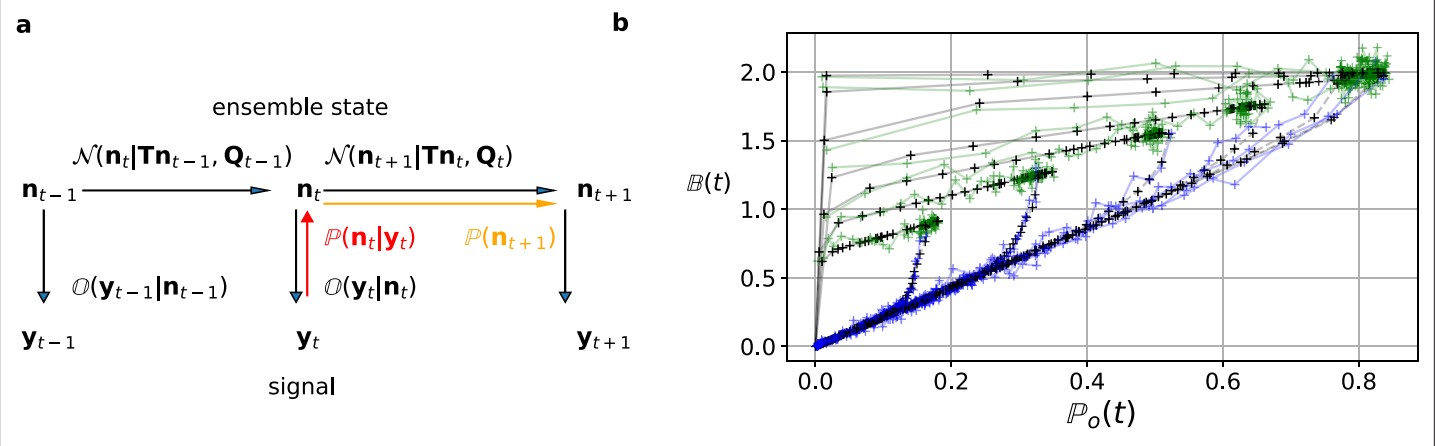

**Figure 2.** The first-order hidden Markov structure is explicitly used by the Bayesian filter. The filter can be seen as continuous state analog of the forward algorithm. (**a**) Graphical model of the conditional dependencies of the stochastic process. Horizontal black arrows represent the conditional multivariate normal transition probability $\mathcal{N}(\mathbf{n}_{t+1}|\mathbf{Tn}_t, \mathbf{Q}_t)$ of a continuous state Markov process. Notably, it is $\mathbf{n}(t)$ which is treated as the Markov state by the KF. The transition matrix $\mathbf{T}$ and the time-dependent covariance $\mathbf{Q}_t = \mathbf{Q}(\mathbf{T}, \mathbf{n}_t)$ characterise the single-channel dynamics. The vertical black arrows represent the conditional observation distribution $\mathbb{O}(\mathbf{y}_t|\mathbf{n}_t)$. The observation distribution summarizes the noise of the experiment, which in the KF is assumed to be multivariate normal. Given a set of model parameters and a data point $\mathbf{y}_t$, the Bayesian theorem allows to calculate in the correction step $\mathbb{P}(\mathbf{n}_t|\mathbf{y}_t)$ (red arrow). The posterior is propagated linearly in time by the model, predicting a state distribution $\mathbb{P}(\mathbf{n}_{t+1})$ (orange arrow). The propagated posterior predicts together with the observation distribution the mean and covariance of the next observation. Thus, it creates a multivariate normal likelihood for each data point in the observation space. (**b**) Observation space trajectories of the predictions and data of the binding per channel vs. open probability for different ligand concentrations. The curves are normalized by the median estimates of $\lambda_b$, and $N_{ch}$ and the ratio of open-channels $\frac{y_{curr}}{N_{ch}i}$ which approximates the open probability $P_o(t)$. The black crosses represent the predicted mean signal $\mathbf{H}\,\mathbb{E}[\mathbf{n}_{t+1}]$, which is calculated by multiplying the observational matrix $\mathbf{H}$ with the mean predicted state $\mathbb{E}[\mathbf{n}_{t+1}]$. For clarity, we used the mean value of the posterior of the KF. The green and blue trajectories represent the part of the time traces with after the jump to non-zero ligand concentration and after jumping backt to zero ligand concentration in the bulk, respectively.

KF is a Bayesian filter (see Materials and methods), that is a continuous state HMM with a multivariate normal transition probability *Ghahramani, 1997* (*Figure 2a*). We define the hidden ensemble state vector

$$\mathbf{n}(t) := (n_1(t), n_2(t), n_3(t), n_4(t))^\top = \sum_{i=1}^{N_{ch}} \mathbf{s}_i(t), \tag{2}$$

which counts the number of channels in each state $\mathbf{s}$ (see Methods). To make use of the KF, we assume the following general form of the dynamic model: The evolution of $\mathbf{n}(t)$ is determined by a linear model that is parametrized by the state evolution matrix $\mathbf{T}$

$$\mathbf{n}_{t+1} = \mathbf{Tn}_t + \omega_t \sim \mathcal{N}(\cdot|\mathbf{Tn}_t, \mathbf{Q}_t), \tag{3}$$

where ~ means *sampled from* and $\mathcal{N}(\cdot|\mu, \Sigma)$ is a shorthand for the multivariate normal distribution, with the mean μ and the variance-covariance matrix $\Sigma$. The state evolution matrix (transition matrix) is related to the rate matrix $\mathbf{K}$ by the matrix exponential $\mathbf{T} = \exp(\mathbf{K}\Delta t)$. The mean of the hidden state evolves according to the equation $\mathbb{E}[\mathbf{n}_{t+1}|\mathbf{n}_t] = \mathbf{Tn}_t$. It is perturbed by normally distributed white process noise $\omega$ with the following properties: The mean value of the noise fulfills $\mathbb{E}[\omega_t] = 0$ and the variance-covariance matrix of the noise is $\text{cov}[\omega_t, \omega_t] = \mathbf{Q}(\mathbf{T}, \mathbf{n}_t)$ (see Materials and methods *Equation 38d*, *Ball, 2016*). In short, *Equation 3* defines a Gaussian Markov process.

The observations $\mathbf{y}_t$ depend linearly on the hidden state $\mathbf{n}_t$. The linear map is determined by an observation matrix $\mathbf{H}$.

$$\mathbf{y}_t = \mathbf{Hn}_t + \nu_t \sim \mathbb{O}(\cdot|\mathbf{Hn}_t) := \mathcal{N}(\cdot|\mathbf{Hn}_t, \Sigma_t) \tag{4}$$

The noise of the measurement setup (Appendix 5 and *Equation 43*) is modeled as a random perturbation of the mean observation vector. The noise fulfills $\mathbb{E}[\nu] = 0$ and $\text{cov}[\nu_t, \nu_t] = \Sigma_t$. *Equation 4* defines the state-conditioned observation distribution $\mathbb{O}$ (*Figure 2a*). The set of all measurements up to time $t$ is defined by $\mathcal{Y}_t = \{y_1, \ldots, y_t\}$. If the system strictly obeys *Equation 3* and *Equation 4*

then the KF is optimal in the sense that it is the minimum variance filter of that system *Anderson and Moore, 2012*. If the distributions of $\nu$ and $\omega$ are not normal, the KF is still the minimum variance filter in the class of all linear filters but there might be better non-linear filters. In case of colored noise $\nu$ and $\omega$ the filtering equations (see Materials and methods) can be reformulated by state augmentation or measurement-time-difference approach techniques *Chang, 2014*. For each element in a sequence of hidden states $\{\mathbf{n}_t : 0 < t < T\}$ and for a fixed set of parameters $\boldsymbol{\theta}$, an algorithm based on a Bayesian filter (*Figure 2a*), explicitly exploits the conditional dependencies of the assumed Markov process. A Bayesian filter recursively predicts prior distributions for the next $\mathbf{n}_t$

$$\mathbb{P}(\mathbf{n}_t|\mathcal{Y}_{t-1}) = \int \mathbb{P}(\mathbf{n}_t|\mathbf{n}_{t-1})\mathbb{P}(\mathbf{n}_{t-1}|\mathcal{Y}_{t-1})\,\mathrm{d}\mathbf{n}_{t-1}, \tag{5}$$

given what is known about $\mathbf{n}_{t-1}$ due to $y_{t-1}$. The KF as a special Bayesian filter assumes that the transition probability is multivariate normal according to *Equation 3*

$$\mathbb{P}(\mathbf{n}_t|\mathcal{Y}_{t-1}) = \int \mathcal{N}(\mathbf{n}_t|\mathbf{T}\mathbf{n}_{t-1},\mathbf{Q}_{t-1})\mathbb{P}(\mathbf{n}_{t-1}|\mathcal{Y}_{t-1})\,\mathrm{d}\mathbf{n}_{t-1} \tag{6}$$

Note, that *Equation 6* is a central approximation of the KF. While the exact transition distribution of an ensemble of ion channels is the generalized-multinomial distribution (Methods *Equation 32*), the quality of normal approximations to multinomial *Milescu et al., 2005* or generalized-multinomial *Moffatt, 2007* distributions depends on the number of ion channels $N_{\mathrm{ch}}$ in the patch and on the position of the probability vector in the simplex space. The difference between the log-likelihoods of the true generalized-multinomial dynamics and *Equation 6* type approximation scales as $1/N_{\mathrm{ch}}$ *Moffatt, 2007*. As a rule of thumb one should be careful with both algorithms for time traces with $N_{\mathrm{ch}} \in \left[10^1, 10^2\right]$. Below or even inside this interval there are more qualified concepts such as the forward algorithm or even particle filters (*Golightly and Wilkinson, 2011*; *Gillespie and Golightly, 2012*) which avoid the normal approximation.

Each prediction of $\mathbf{n}_t$ (*Equation 6*) is followed by a correction step,

$$\mathbb{P}(\mathbf{n}_t|\mathcal{Y}_t) = \frac{\mathbb{O}(\mathbf{y}_t|\mathbf{n}_t)\mathbb{P}(\mathbf{n}_t|\mathcal{Y}_{t-1})}{\int \mathbb{O}(\mathbf{y}_t|\mathbf{n}_t)\mathbb{P}(\mathbf{n}_t|\mathcal{Y}_{t-1})\,\mathrm{d}\mathbf{n}_t}, \tag{7}$$

that allows to incorporate the current data point into the estimate, based on the Bayesian theorem (*Chen, 2003*). Additionally, the KF assumes (*Anderson and Moore, 2012*; *Moffatt, 2007*) a multivariate normal observation distribution

$$\mathbb{P}(\mathbf{n}_t|\mathcal{Y}_t) = \frac{\mathcal{N}(\mathbf{y}_t|\mathbf{H}\mathbf{n}_t,\boldsymbol{\Sigma}_t)\mathbb{P}(\mathbf{n}_t|\mathcal{Y}_{t-1})}{\int \mathcal{N}(\mathbf{y}_t|\mathbf{H}\mathbf{n}_t,\boldsymbol{\Sigma}_t)\mathbb{P}(\mathbf{n}_t|\mathcal{Y}_{t-1})\,\mathrm{d}\mathbf{n}_t}, \tag{8}$$

If the initial prior distribution is multivariate normal then due to the mathematical properties of the normal distributions the prior and posterior $\mathbb{P}(\cdot)$ in *Equation 8* become multivariate normal *Chen, 2003* for each time step. In this case, one can derive algebraic equations for the prediction (Materials and methods *Equation 37*, *Equation 38d*) and correction (Materials and methods *Equation 58* and *Equation 58*) of the mean and covariance. The algebraic equations originate from the fact that a normal prior is the conjugated prior for the mean value of a normal likelihood. Due to the recursiveness of its equations, the KF has a time complexity that is linear in the number of data points, allowing a fast algorithm. The denominator of *Equation 8* is the normal distributed marginal likelihood $\mathbb{L}(\mathbf{y}_t|\mathcal{Y}_{t-1},\boldsymbol{\theta})$ for each data point, which constructs by

$$\mathbb{L}(\mathcal{Y}_T|\boldsymbol{\theta}) = \prod_{t=2}^{N_T} \mathbb{L}(\mathbf{y}_t|\mathcal{Y}_{t-1},\boldsymbol{\theta}) = \prod_{t=2}^{N_T} \int \mathbb{O}(\mathbf{y}_t|\mathbf{n}_t)\mathbb{P}(\mathbf{n}_t|\mathcal{Y}_{t-1},\boldsymbol{\theta})\,\mathrm{d}\mathbf{n}_t = \prod_{t=2}^{N_T} \mathcal{N}(\mathbf{y}_t|\mathbf{H}\mathbb{E}[\mathbf{n}_t],\mathbf{H}\mathbf{P}_t\mathbf{H}^\top + \boldsymbol{\Sigma}_t), \tag{9}$$

a product marginal likelihood of normal distributions of the whole time trace $\mathcal{Y}_T = \{\mathbf{y}_1, \ldots, \mathbf{y}_{N_T}\}$ of length $N_T$ for the KF. For the derivation of $\mathbf{P}_t$ and $\boldsymbol{\Sigma}_t$ see Materials and methods (*Equation 38d*) and *Equation 43*. $\mathbf{P}_t$ is the covariance of the prior distribution over $\mathbf{n}(t)$ before the KF took $\mathbf{y}(t)$ into account. The likelihood for the data allows to ascribe a probability to the parameters $\boldsymbol{\theta}$, given the observed data (Methods *Equation 20*). An illustration for the operation of the KF on the observation space is

given in *Figure 2b*. The predicted mean signal $\mathbf{H}\mathbb{E}[\mathbf{n}(t)]$ corresponds to binding degree $B(t) = \frac{\mathbf{H}\mathbb{E}[\mathbf{n}(t)]_1}{N_{\text{ch}}}$ and open probability $P_O(t) = \frac{\mathbf{H}\mathbb{E}[\mathbf{n}(t)]_2}{N_{\text{ch}}}$. These values are plotted as vector trajectories.

The standard KF (*Moffatt, 2007*; *Anderson and Moore, 2012*; *Chen, 2003*) has additive constant noise $\Sigma_t = const$ in the observation model. Thus, in this case a constant variance term $\Sigma$ is added, in *Equation 9* to the aleatory variance $\mathbf{H}\mathbf{P}_t\mathbf{H}^\top$ which, as mentioned above, originates (*Equation 38d*) from the the fact that we do not know the true system state $\mathbf{n}(t)$. For signals with Poisson-distributed photon counting or open-channel noise, we need to generalize the noise model to account for additional white-noise fluctuations with $\mathbf{n}(t)$-dependent *variance*. For instance, in single-channel currents additional noise is often observed whose variance is referred to by $\sigma_{\text{op}}^2$. In macroscopic currents this additional noise can be modeled by a term $\sigma_{\text{op}}^2 n_4(t)$, causing state-dependency of our noise model.

$$y(t) = \mathbf{H}\mathbf{n}(t) + \nu_{\text{m}}(t) + \nu_{\text{op}}(t) \Leftrightarrow y \sim \text{O}(y|\mathbf{n}) = \mathcal{N}\left(y|\mathbf{H}\mathbf{n}(t), \sigma_{\text{m}}^2 + n_4(t)\sigma_{\text{op}}^2\right) = \mathcal{N}(y|\mathbf{H}\mathbf{n}(t), \Sigma_t) \tag{10}$$

The second noise term $\nu_{\text{op}}$ is defined in terms of the first two moments $\mathbb{E}(\nu_{\text{op}}) = 0$ and $\text{var}(\nu_{\text{op}}) = \mathbb{E}(\nu_{\text{op}}^2) = \sigma_{\text{op}}^2 n_4(t)$. To the best of our knowledge such a state-dependent noise makes the integration of the denominator of *Equation 8* (which is also the incremental likelihood) intractable

$$\mathbb{P}(y(t)) = \int \mathcal{N}(y|\mathbf{H}\mathbf{n}, \sigma_{\text{m}}^2 + n_4\sigma_{\text{op}}^2)\,\mathcal{N}(\mathbf{n}|\bar{\mathbf{n}}(t), \mathbf{P}(t))\,\mathrm{d}n \tag{11a}$$

$$= \frac{1}{const}\int \exp\left(\frac{(y - \mathbf{H}\mathbf{n})^2}{2(\sigma_{\text{m}}^2 + n_4\sigma_{\text{op}}^2)}\right) \exp\left(\frac{1}{2}(\mathbf{n} - \bar{\mathbf{n}}(t))\mathbf{P}^{-1}(\mathbf{n} - \bar{\mathbf{n}}(t))^\top\right)\,\mathrm{d}n \tag{11b}$$

This is because the state distribution $\mathcal{N}(\mathbf{n}|\bar{\mathbf{n}}(t), \mathbf{P}(t))$ as the prior also influences the variance parameter of the likelihood which means that the conjugacy property is lost. While a normal distribution is the conjugated prior of the mean of a normal likelihood, it is not the conjugated prior for the variance. However, by applying the theorem of total variance decomposition *Equation 46a* we deduce a normal approximation to *Equation 8* and to the related problem of Poisson-distributed noise in fluorescence *Equation 57*, *Equation 55a* data. By computing the mean and the variance or covariance matrix of the signal, we can reformulate the noise model to fit the form of the traditional KF framework. Note, that the derived equations for the covariance matrix are still exact for the more general noise model. Mean and covariance just do not form a set of sufficient statistics anymore.

Our derivation is not limited to ligand-gated ion channels. For example, when investigating voltage-gated channels, the corresponding noise model can be easily adapted. This holds also when using the P/n protocol for which the noise model resembles that of the additional variance in the fluorescence signal. The additional variance is induced because the mean signal from the ligands swimming in the bulk (Materials amd methods *Equation 43* Appendix 5) is eliminated by subtracting scaled mean reference signal which itself has an error. This manipulation adds additional variance to the resulting signal comparable to P/n protocol. Other experimental challenges, as for example series resistance compensation promoting oscillatory behavior of the amplifier, deserve certainly advanced treatment. Nevertheless, for voltage-clamp experiments with a rate equation approach it also becomes clear (*Lei et al., 2020*) that modeling of the actual experimental limitations, including series resistance, membrane and pipette capacitance, voltage offsets, imperfect compensations by the amplifier, and leak currents are necessary for consistent kinetic scheme inference.

The Bayesian posterior distribution

$$\mathbb{P}(\boldsymbol{\theta}|\mathcal{Y}_T) = \frac{\mathbb{L}(\mathcal{Y}_T|\boldsymbol{\theta})\mathbb{P}(\boldsymbol{\theta})}{\int \mathbb{L}(\mathcal{Y}_T|\boldsymbol{\theta})\mathbb{P}(\boldsymbol{\theta})d\boldsymbol{\theta}} \tag{12}$$

encodes all information from model assumptions and experimental data used during model training (see Materials and methods). A full Bayesian inference is usually not an optimization (finding the global maximum or mode of the posterior or likelihood) but calculates all sorts of quantities derived from the posterior distribution such as mean values of any function $f$ including the mean value or covariance matrix of the parameters themselves or even the likelihood of the data.

$$\mathbb{E}[f] = \int f(\boldsymbol{\theta})\mathbb{P}(\boldsymbol{\theta}|\mathcal{Y}_T)d\boldsymbol{\theta} \tag{13}$$

Besides the covariance matrix of the parameter to express parameter uncertainty, the posterior allows to calculate a credibility volume. The smallest volume $V_P$ that encloses a probability mass $P$ of

$$P = \int_{V_P} \mathbb{P}(\boldsymbol{\theta}|\mathcal{Y}_T)d\boldsymbol{\theta}. \tag{14}$$

is called the Highest Density Credibility Volume/Interval (HDCV/HDCI). Those credibility volumes should not be confused with confidence volumes although under certain conditions they can become equivalent. Given that our model sufficiently captures the true process, the true values $\boldsymbol{\theta}_{\text{true}}$ will be inside that volume with a probability $P$. Unfortunately, typically there is no analytical solution to *Equation 12*. However, it can be solved numerically with Monte Carlo techniques, enabling to calculate all quantities related to *Equation 13* and *Equation 14*. Our algorithm uses automatic differentiation of the statistical model to sample from the posterior (*Appendix 1—figure 1a*) via Hamiltonian Monte Carlo (HMC) (*Betancourt, 2017*), see Appendix 7, as provided by the Stan software (*Hoffman and Gelman, 2014*; *Gelman et al., 2015*).

## Benchmark for PC data against the gold standard algorithms

We compare the posterior distribution (*Figure 3*) of our algorithm against Bayesian versions of the deterministic (*Milescu et al., 2005*) and stochastic (*Moffatt, 2007*) algorithms, which we consider as the gold standard algorithms for macroscopic patch-clamp data. Simulated currents of a patch with $N_{ch} = 5 \cdot 10^3$ are shown in (*Figure 3d*). The resulting posteriors (*Figure 3a*) show that both former algorithms are further away from the true parameter values with their maxima or mean values (*Figure 3a*). E.g., the relative error of the maximum of the posterior are $\Delta k_{21} \approx 200\%$ for *Milescu et al., 2005* and $\Delta k_{32} \approx 240\%$ for *Moffatt, 2007*. The four other parameters including the three equilibrium constants behave less problematic as judged by their relative error. Additionally, if one does not only judge the performance by the relative distance of maximum (or some other significant point) of the posterior but considers the spread of the posterior as well, it becomes apparent, that the marginal posterior of both former algorithms fail to cover the true values within at least the reasonable parts of their tails. Accordingly, for maximum likelihood inferences the true value would be far outside the estimated confidence interval. For the RE approach only the marginal posterior of $\tilde{K}_{21}$ is nicely centered over the true values and the marginal of $\tilde{K}_{32}$ could be considered to cover within a reasonable part of the distribution the true value. Uncertainty quantification is investigated in more detail further down (*Figures 4–9*). Note that in *Figure 3a*, parameter unidentifiability by heavy tails/ multiple maxima of the posterior distribution or (anti-) correlation is easily visible as non axial symmetric patterns.

To assess the location of the posterior conditioned on $N_{\text{ch}}$, we select the median vector $\boldsymbol{\theta}$ of the marginal posteriors and calculate its Euclidean distance to the true values by:

$$\text{Euclidean Error} = \sqrt{\sum_i [\theta_i/\theta_{i,\text{true}} - 1]^2} \tag{15}$$

This defines a single value to judge the overall accuracy of the posterior. Varying $\sigma_{\text{op}}/i$ reveals the range of the validity (*Figure 3b*) of the algorithm (red) from *Moffatt, 2007*. While both stochastic approaches are nearly equivalent for low open-channel noise, the RE (blue) performs consistently poorer. It may seem surprising that even for $\sigma_{op}/i < 0.01$ the two stochastic algorithms start to produce different results. But considering the scaling (Materials and methods *Equation 46a*) of the total open-channel noise (top axis) from currents of an ensemble patch $\propto (N_{\text{ch}}P_{\text{open,max}}0.5)^{0.5}\sigma_{\text{open}}$ one sees that if $\propto (N_{\text{ch}}P_{\text{open,max}}0.5)^{0.5}\sigma_{\text{open}}$ approaches $\sigma$ the traditional KF suffers from ignoring state dependent noise contributions. The lower scale changes with experiments (e.g. $N_{\text{ch}}$ and $\sigma_{op}$). In contrast, the upper scale is largely independent of the particular measurements. The two different normalizations indicate an experimental intuition: " Why should I consider the extra noise from the open state of the single channel if only $\sigma_{\text{op}}/i = \sigma_{\text{op}}/\sigma \approx 0.01$" is misleading. The small advantage of our algorithm for small $\sigma_{\text{op}}/i$ over *Moffatt, 2007* is due to the fact that we could apply an informative prior in the formulation of the inference problem on $\sigma_{\text{exp}} \sim \text{normal}(\sigma^2_{\text{exp,true}}, \sigma^2_{\text{exp,true}} \cdot 0.01)$ by taking advantage of our generalization (*Equation 46a*) Bayesian filter. Further, *Figure 3b* indicates the importance that the functional form of the likelihood is flexible enough to capture the second order statistics of the noise of the data sufficiently.

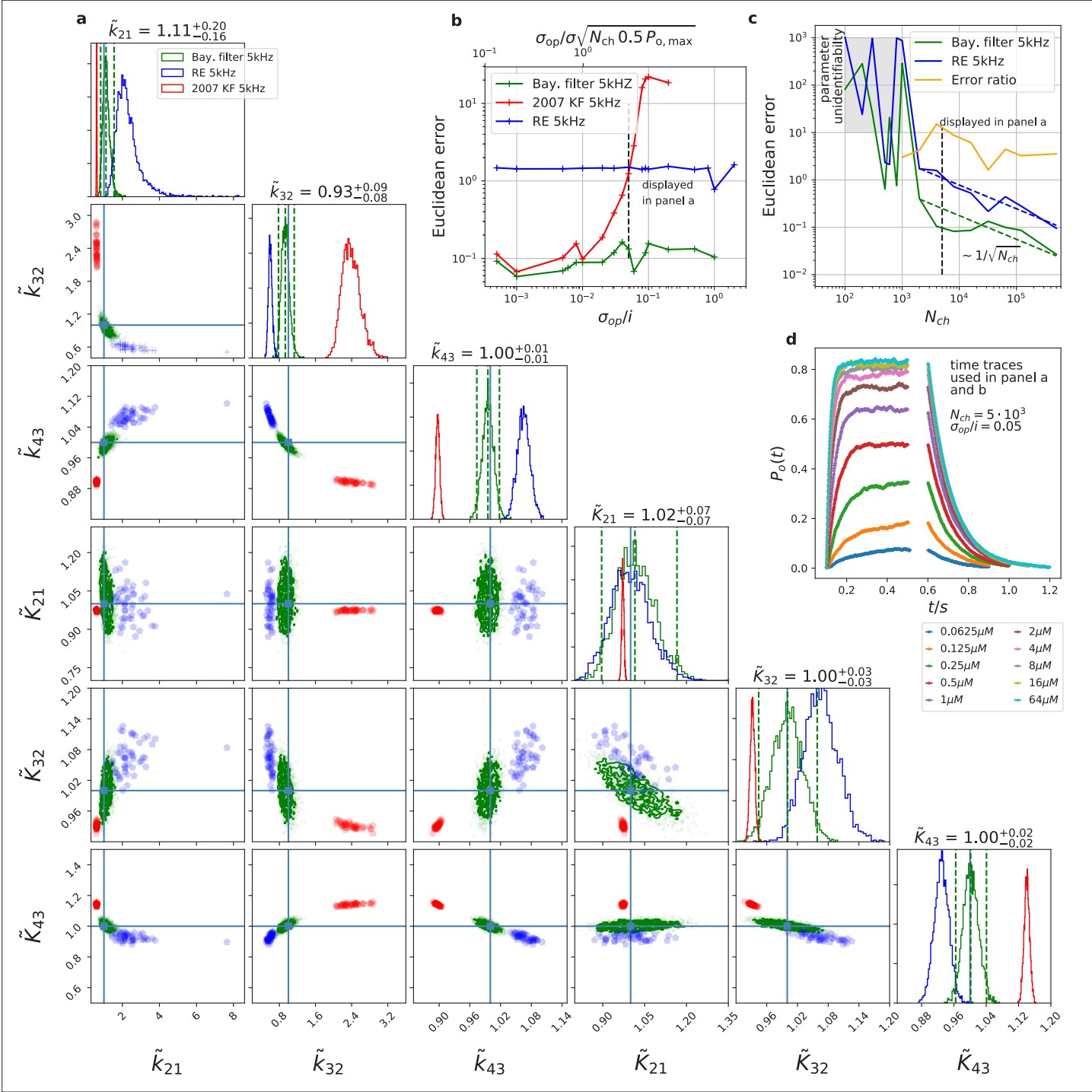

**Figure 3.** The Bayesian filter overcomes the sensitivity to varying (open-channel) noise of the classic Kalman filter and does not show the overconfidence of the RE-approach. Overall it shows the highest accuracy and the posterior covers the true values. The classical deterministic RE (blue), 2007 Kalman filter (red) and our Bayesian filter (green) are implemented as a full Bayesian version and the obtained posterior distributions are compared. For all PC data sets in the figure the analysing frequencies $f_{ana}$ ranges within 2-5. (**a**) Posterior of the parameters for the 3 algorithms for the data set displayed in panel d. The blue crosses indicate the true values. All samples are normalized by their true values which is indicated by the ~ above the parameters. For clarity, we only show a fraction of the samples of the posterior for blue and red. b, Effect of open channel noise: The Euclidean error for all three approaches is plotted vs. $\sigma_{op}/i$ (low axis).The upper axis displays the ratio of the 'typical' standard deviation of the open channel excess noise of the ensemble of channels $\sigma_{op}\sqrt{N0.5P_{o,max}}$ to the standard deviation of instrumental noise. c, Influence of patch size: Scaling of the Euclidean error vs. $N_{ch}$ follows $\sim (N_{ch})^{-0.5}$ indicated by the dashed lines for $N_{ch} > 2 \cdot 10^3$ for the RE and the Bayesian filter approach. The data indicates a constant error ratio (orange) for large $N_{ch}$. For $N_{ch} < 2 \cdot 10^3$ samples of the posteriors for many data sets suggest an improper posterior. An instrumental noise of $\sigma_{ex}/i = 1$

*Figure 3 continued on next page*

*Figure 3 continued*

and $\sigma_{\mathrm{op}}/i = 0.01$ was used. (**d**) The time traces on which the posteriors of panel a are based (for the ligand concentrations see *Figure 1*). Panel b used the same data too, but σ and $\sigma_{op}$ were varied.

The online version of this article includes the following source data for figure 3:

**Source data 1.** The data folder includes all 15 sets of time traces.

---

For an increasing data quality, which in our benchmark is an increasing $N_{\mathrm{ch}}$ per trace, we show (*Figure 3c*) that the deterministic RE and our Bayesian filter are consistent estimators, that is they converge in distribution to the true parameter values with their posterior maxima or median for increasing data quality. The scaling of the RE approach (blue) and our Bayesian filter (green) vs. $N_{\mathrm{ch}}$ shows that for large $N_{\mathrm{ch}}$ both algorithms seem to have a constant error ratio relative to each other. They are both well described by $\mathrm{error}(N_{\mathrm{ch}}) \propto a/\sqrt{N_{\mathrm{ch}}}$ with an error ratio computed from the fit of 4.4. Thus, although our statistical model is singular (meaning that the fisher information matrix is singular *Watanabe, 2007*), its asymptotic learning behaviour is similar to a regular model (*Figure 4c*) which, however, means that the euclidean error from both algorithms stays different also for large $N_{\mathrm{ch}}$. For data with $N_{\mathrm{ch}} < 2 \cdot 10^3$ the samples from the posterior typically indicate that the posterior is improper which is defined as

$$\int \mathbb{P}(\boldsymbol{\theta}|\mathbf{y})d\boldsymbol{\theta} = \infty \tag{16}$$

We consider this as the case of unidentified parameters. This data-driven definition is in so far different from structural and practical identifiability definitions (*Middendorf and Aldrich, 2017a*; *Middendorf and Aldrich, 2017b*) as the two latter cases are not distinguished. Still the practical consequence of structural or practical unidentifiability, which is usually an improper posterior, is captured. Cases of structural or practical unidentifiability which lead to a confined region of constant posterior density will be considered identified as the posterior is still normalizable thus the uncertainty quantification will still be correct, even when this finding is not sufficient to answer the research question at hand.

## Benchmarking for cPCF data against the gold standard algorithm

For the simulated time traces with an optimistically high signal-to-noise assumption, the posterior of the KF (from hereon KF denotes our Bayesian Filter) and a RE (*Milescu et al., 2005*) approach are compared for cPCF data (*Figure 4a–d*). For a brief introduction of the RE approach, see Appendix 8 . The failure to analyze PC data with moderate open-channel noise (*Moffatt, 2007*; *Figure 3a*) disqualifies the classical KF with its constant noise variance also as a useful algorithm for fluorescence data, because here the Poisson distribution of the signal generates an even stronger state dependency of the signal variance.

By "high signal-to-noise assumption" , we refer to an experimental situation with a standard deviation of the current recordings $\sigma_{\mathrm{ex}}/i = 0.5$, a low additional $\sigma_{\mathrm{op}}/i = 0.05$, and a high mean photon rate per bound ligand and frame $\lambda_{\mathrm{b}} = 5$. Additionally, we assume vanishing fluorescence background noise generated by the ligands in the bulk. The benefit of the high signal-to-noise is that the limitations of the two different approximations to the stochastic process of binding and gating can be investigated without running into the risk of being compensated or obscured by the noise from the experimental setup. For these experimental settings (*Figure 4a*), we calculate the Euclidean distance of the median (*Equation 15*) for different $N_{\mathrm{ch}}$. For $N_{\mathrm{ch}} < 500$ (gray shaded area in *Figure 4a*), the Euclidean error of both algorithms is roughly the same. On the single parameter level (*Figure 4b*), this can be seen as an onset of correlated deviations from the true value for both algorithms. Each marginal posterior has for each $N_{\mathrm{ch}}$ a similar deviation in magnitude and direction. That is in particular true for $\tilde{k}_{32}$ and $\tilde{K}_{32}$ which dominate *Equation 15* . In spite of the correlation in direction of the errors of $\tilde{k}_{21}$ and $\tilde{K}_{21}$ their magnitude is still smaller for the KF. In summary, this indicates that in this regime the approximations to the involved multinomial distributions fail in a similar manner for both algorithms. That implies that treating the autocorrelation of the gating and binding becomes similar important compared to the error induced by normal approximations (which are used by the KF and the RE approach). For larger $N_{\mathrm{ch}}$, the Euclidean error of the RE is on average 1.6 times larger than the corresponding error of the

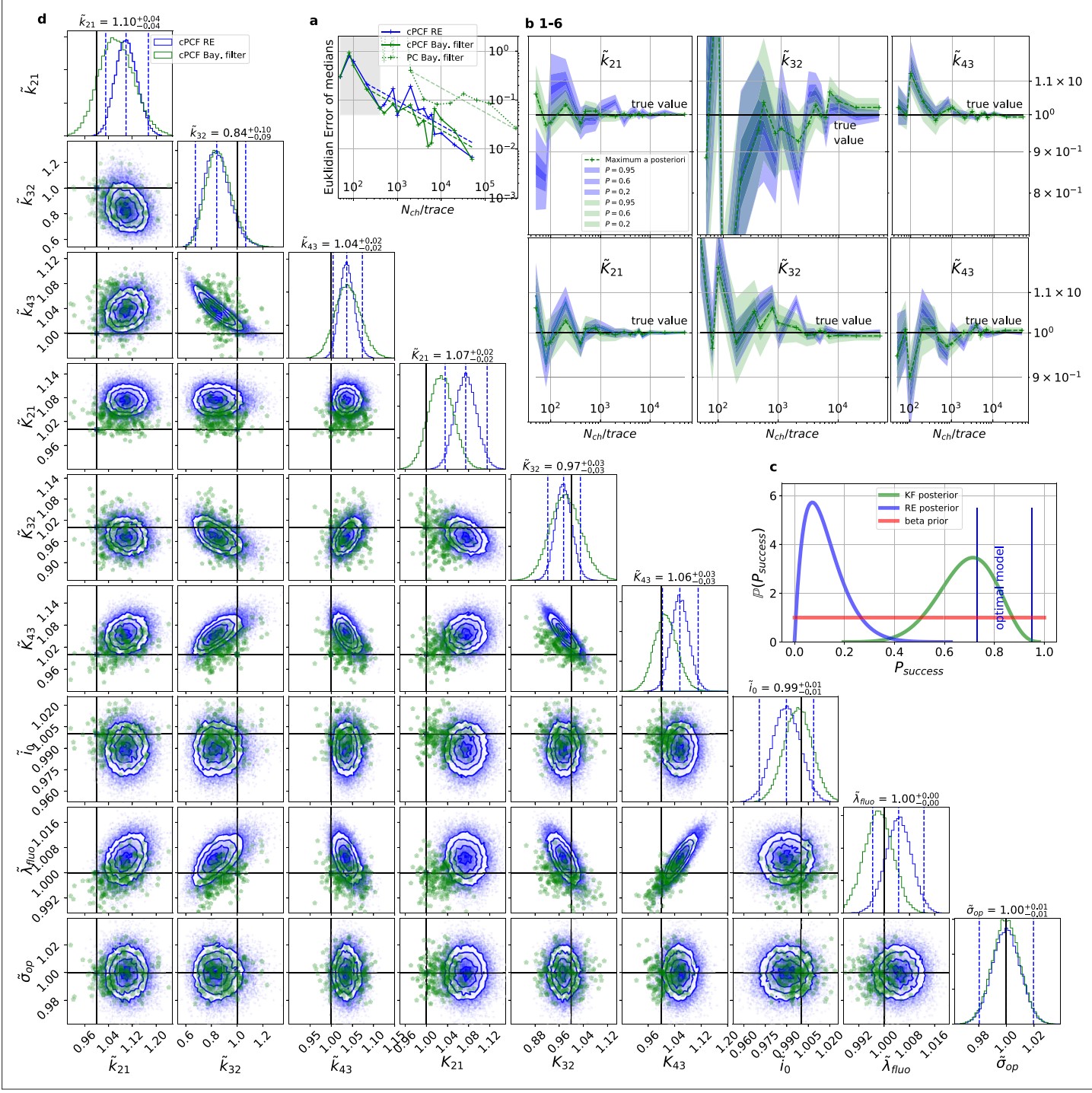

**Figure 4.** For multidimensional data (cPCF) the RE approach almost approaches the accuracy (Euclidean error) of the Bayesian Filter. However, only the Bayesian filter covers the true value in a reasonable HDCV while RE based posteriors are too narrow. All samples are normalized by their true values which is indicated by the ~ above the parameters. (**a**) Euclidean errors of the maximum for the rate $k_{ij}$ and equilibrium constants $K_{ij}$ obtained by the KF (green) and from the REs (blue) are plotted against $N_{ch}$ for $\sigma_{ex}/i = 0.5$, $\sigma_{op}/i = 0.05$ and $\lambda_b = 5$. Both algorithms scale like $1/\sqrt{N_{ch}}$ (dashed lines) for larger $N_{ch}$ which is the expected scaling For smaller $N_{ch} < 500$ (gray range) the error is roughly the same indicating that limitations of the normal approximation to the multinomial distribution dominate the overall error in this regime. The combination of fluorescence and current data(cPCF) decreases the eucleadian error for both approaches compared to current data alone(PC). (**b**), HDCI and the mode of the 3 $k_{ij}$ and 3 $K_{ij}$ plotted vs. $N_{ch}$ revealing that the maximum is a consistent estimator (converges in distribution to the true value with increasing data quality). While the KF (green) 0.95-HDCI includes usually the true value, the RE HDCI (blue) is too narrow and, thus, the real values are frequently not included. (**c**) Bayesian estimation of true success probability for the event that all 6 0.95-HDCI include the respective true values at the same time by a binomial likelihood. Since the data sets have

*Figure 4 continued on next page*

Figure 4 continued

different $N_{ch}$ and the model approximations become better with increasing $N_{ch}$, we use a cut-off for $N_{ch} = 200$. d, Comparison of 1-D and combinations of 2-D marginal posteriors of the parameters of interest for both algorithms calculated from a $N_{ch} = 10^3$ simulation. Blue lines indicate the true value. We depict that in two dimensions the disproportion of the deviation of the mode and the spread of RE (blue) approach is worsened while KF (green) posterior includes the true values with more reasonable probability mass.

The online version of this article includes the following source data for figure 4:

**Source data 1.** 6μMol.

**Source data 2.** 64μMol.

**Source data 3.** 32μMol.

**Source data 4.** 8μMol.

**Source data 5.** 4μMol.

**Source data 6.** 1μMol.

**Source data 7.** 2μMol.

**Source data 8.** 025μMol.

**Source data 9.** 05μMol.

**Source data 10.** 00625μMol.

**Source data 11.** 003125μMol.

**Source data 12.** 0125μMol.

posterior mode of the KF, which we deduce by fitting the function $\text{error}(N_{ch}) = \frac{a}{\sqrt{N_{ch}}}$. On the one hand, both algorithms are better in approaching the true values than with patch-clamp data alone. On the other hand, the smaller error ratio means, that adding a second observable constrains the posterior, such that much of the overfitting is prevented for the RE approach. By overfitting, we define the adaptation of any inference algorithm to the specific details of the used data set due to experimental and intrinsic noise which is aggravated if too complex kinetic schemes are used. Similarly, (**Milescu et al., 2005**) showed that the over fitting tendency of the RE can be reduced if the autocorrelation of the

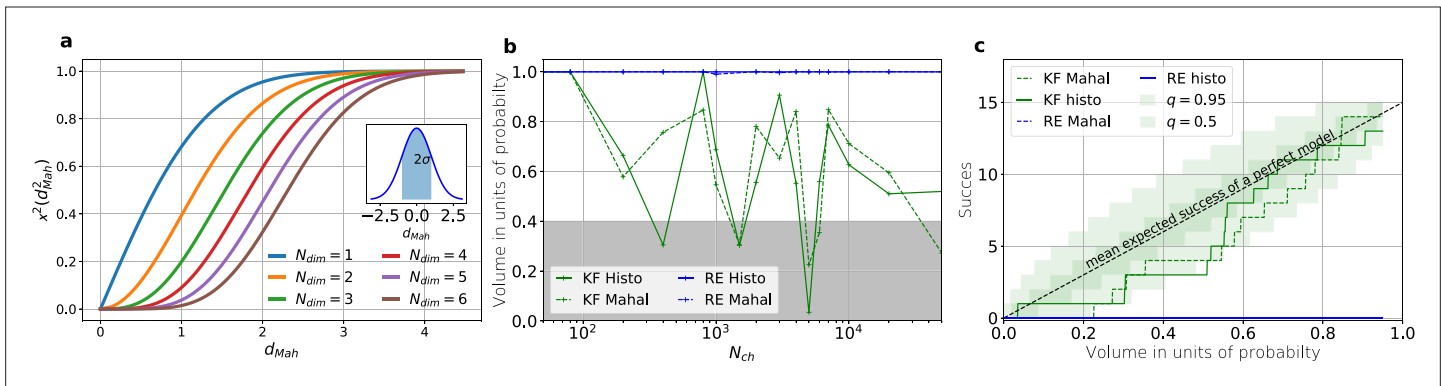

**Figure 5.** The HDCV of the posterior of the KF follows the requested binomial statistics while the HDCVs of the RE approach are too narrow. (**a**) Cumulative $\chi$-square distribution vs. the Mahalanobis distance $d_{\text{Mah}}$. The y axis denotes the probability mass which is counted by moving away from the maximum before an ellipsoid with distance $d_{\text{Mah}}$ is reached. The different colours represent the changes of the cdf with an increasing number of rate parameters. The blue cdf at $d_{\text{Mah}} = 1$ represents how much probability mass can be found from $\int_{-\sigma}^{\sigma} \text{normal}(\theta, 0, \sigma)d\theta$, see inset. In one dimension, we can expect to find the true value within $2\sigma$ around the mean with the usual probability of $P = 0.682$ for univariate normally distributed random variables. The six parameters (brown) of the full rate matrix will almost certainly be beyond $d_{\text{Mah}} = 1.0$. The higher the dimensions of the space the less important becomes the maximum of the probability density distribution for the typical set which is by definition the region where the probability mass resides. The mathematical reason for this is that the probability mass $P = \int_V \mathbb{P}(\boldsymbol{\theta})dV$ is the integrated product of volume and probability density. b, The two methods to count volume in units of probability mass for the KF (green) and the RE (blue). The gray area indicates which data sets are considered a success if one chooses to evaluate a proabability mass of 0.4 of each posterior around its mode. All data sets in the white area are considered a failure. For the optimistic noise assumptions $\sigma_{\text{ex}} = 0.5 \cdot i$, $\sigma_{\text{op}} = 0.05 \cdot i$ and a mean photon count per bound ligand per frame $\lambda_{\text{b}} = 5$ the RE approach (blue) distributes the probability mass such that the HDCV never includes the true rate matrix. From $N_{ch} > 100$ both HDCV estimates of the KF posterior (green curves) include the true value within a reasonable volume and show a similar behaviour. c, Binomial success statistics of HDCV to cover the true value vs. the expected probability constructed from the data of (**b**). Calculated for $i = 0.25\sigma$ and $\sigma_{\text{op}} = 0.025i$ and $\lambda_{\text{b}} = 5$ and minimal background noise.

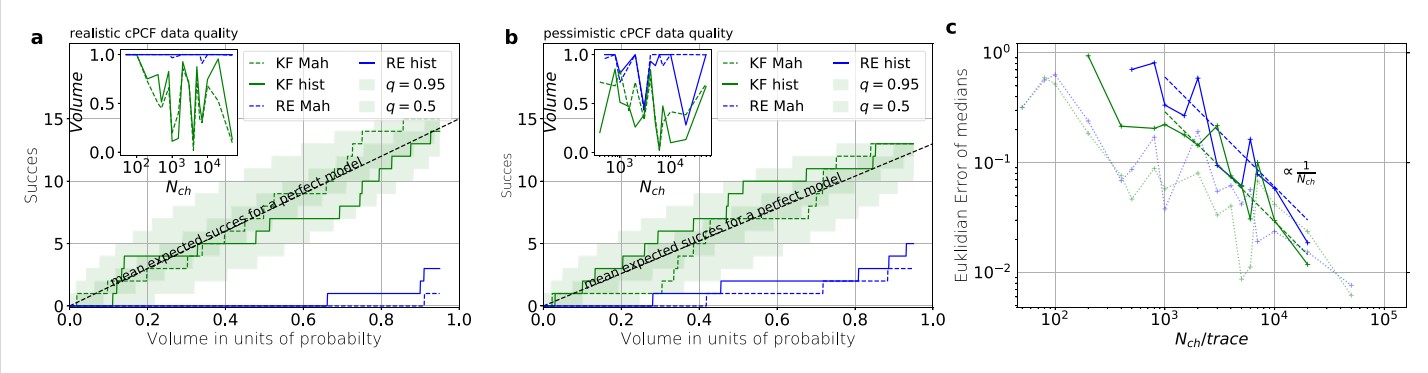

**Figure 6.** Even for the highest tested experimental noise the RE approach does not follow the required binomial statistics, generating an underestimated uncertainty. (**a**) Binomial success statistics of HDCV to cover the true value vs. the expected probability. Calculated for $i = \sigma$ and $\sigma_{op} = 0.1i$ and $\lambda_b = 0.375$ and a strong background noise. (**b**) Binomial success statistics of HDCV to cover the true value vs. the expected probability. For $10 \cdot i = \sigma$ and $\sigma_{op} = 1i$ and $\lambda_b = 0.375$ and a strong background noise. For both algorithms, the adaptation of the sampler of the posterior was more fragile for small $N_{ch}$, leading to differences in the posterior if the posterior is constructed from different independent sampling chains. Those data sets were then excluded. We assume that these instabilities are induced in both algorithms by the shortcomings of the multivariate normal assumptions. (**c**) Comparison of the Euclidean error vs. $N_{ch}$ for the pessimistic noise case (solid lines) with Euclidean error for the optimistic noise case (dotted lines).

data is eliminated. The dotted green curve derives from PC data. The Euclidean error is roughly an order of magnitude larger for $N_{ch} > 2000$. Thus, in this regime the cPCF data set is equivalent to $10^2$ fold more time traces or $10^2$ more $N_{ch}$ in a similar PC data set. For $N_{ch} < 2000$ only cPCF establishes parameter identifiability (given a data set of 10 ligand concentrations and no other prior information). In *Figure 4b(1-6)*, we demonstrate the 0.95-HDCI (*Equation 14*) of all parameters and their modes vs. $N_{ch}$. Even though the Bayesian filter and the RE approach are both consistent estimators, the RE approach covers the true values with its 0.95-HDCI only occasionally. The modeling assumption of the RE approach of treating each data point as if it does not come from a Markov process but from an individual draw from a multinomial distribution with deterministically evolving mean and variance makes the parameter estimates overly confident (*Figure 4b(1-6)*) . A likely explanation can be found by analyzing the extreme case where data points are sampled at high frequency relative to the time scales of the channel dynamics. The RE approach treats each data point as a new draw from *Equation 67* while in reality the ion channel ensemble had no time to evolve into a new state. In contrast, the KF updates its information about the ensemble state after incorporating the current data point and then predicts from this updated information the generalised multinomial distribution of the next data point. For $N_{ch} > 200$, the marginal posterior of the KF usually contains the true value. Nevertheless, one might depict a bias in both algorithms, in particular (*Figure 4b* 2,4) for $\tilde{k}_{32}$ and $\tilde{K}_{32}$ for $N_{ch} < 2 \cdot 10^3$, similar to the findings of *Moffatt, 2007* . A proper investigation of bias can be found in Figure 11 and 12 and in the Appendix. Notably, with the more realistic higher experimental noise level, in those tests the bias is hardly or not all detectable (consider the unfiltered or infinitely fast integrated data). A plausible explanation is that the bias only occurs (*Figure 4* 2,4) because the data are that perfect that the discrete nature of the ensemble dynamics is almost visually detectable, thus deviating from to the modeling assumption of multi-variate normal distributions.

To investigate the six one-dimensional 0.95-HDCIs simultaneously, we declare the analysis of a data set as successful if all 0.95-HDCIs include the true values. Otherwise we define it as a failure. This enables to determine the true probability at which the probability mass of the KF and the RE approach covers the true values in a binomial setting. The left blue vertical line in *Figure 4c* indicates $p = 0.95^6 \approx 0.735$ which is the lower limit and which would be the true success probability for an ideal model whose six 0.95-HDCIs are drawn from $y \sim \text{binomial}(0.95, 6)$. This is the probability of getting 6 successes in 6 trials. The right blue vertical line equals $p = 0.95$, signifying the upper limit obtained by treating the six $0.95-$ HDCIs as being drawn from $y \sim \text{binomial}(0.95, 1)$ each, which is a rather loose approximation. All marginal distributions are computed from the same high-dimensional posterior which is formed by one data set for each trial. Thus, the six $0.95-$ HDCIs $y \sim \text{binomial}(0.95, 1)$ must have success rates between those two extremes if the algorithm creates an accurate posterior. We next combine the binomial likelihood with the conjugated beta prior (*Hines et al., 2014*) for

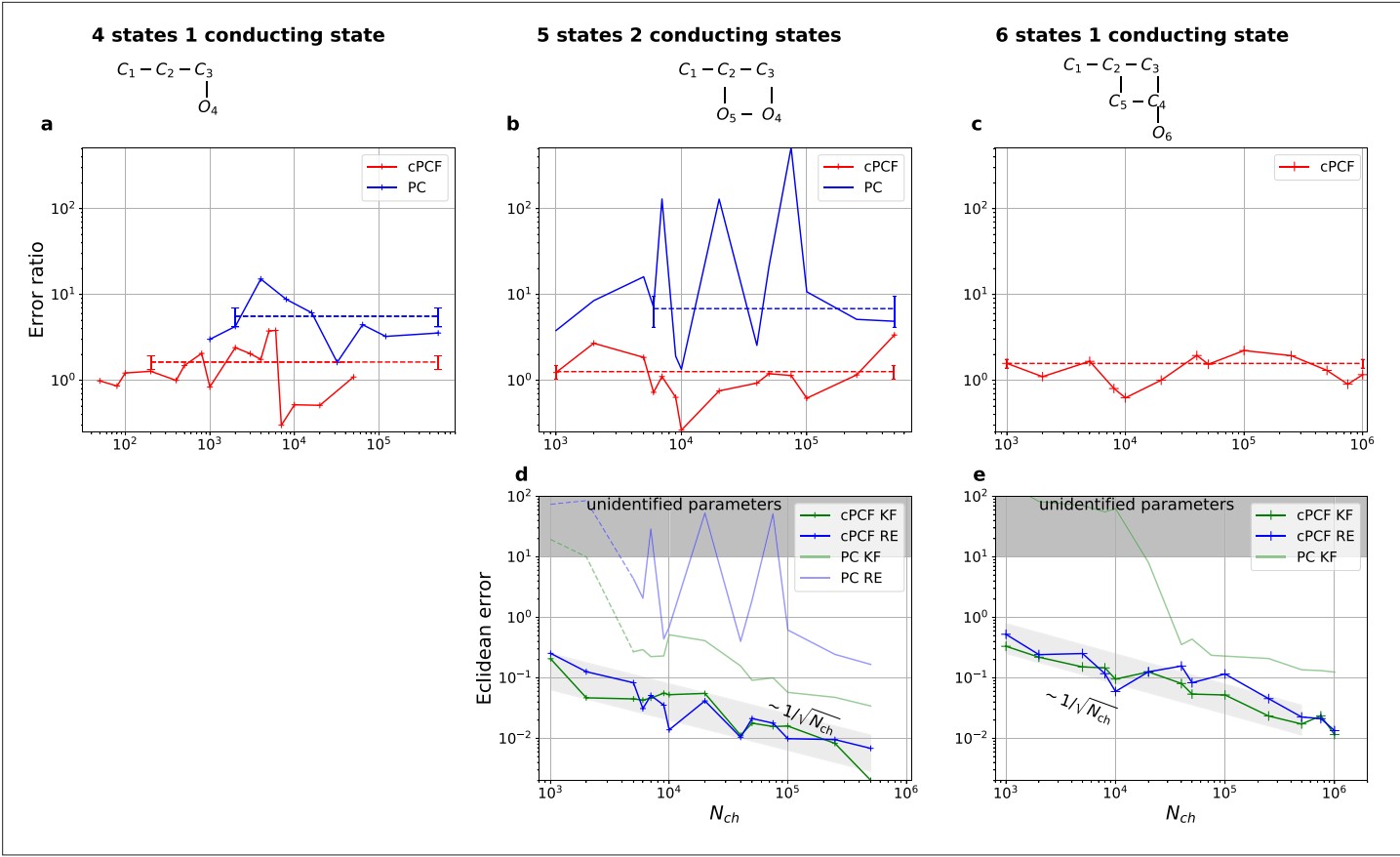

**Figure 7.** Higher model complexity drastically increases the minimal requirements of the data. With PC data the RE approach is frequently incapable to identify all parameters while the Bayesian filter is more robust. cPCF data alleviate the parameter unidentifiabilities for patch sizes for which PC data are insufficient. Each panel column corresponds to a particular true process with increasing complexity from left to right, as indicated by the model schemes on top. Within all kinetic schemes, each transition to the right adds one bound ligand. Each transition to left is an unbinding step. Vertical transitions are either conformational or opening transitions. Plots in each row share the same y-axis respectively. Each column shares the same abscissa. (**a-c**) Error ratio for PC data (blue) and cPCF data (red). The dashed lines indicate the mean error ratio under the simplifying assumption that the error ratio does not depend on $N_{ch}$. The vertical bars are the standard deviations of the mean values. Theses values were calculated from the Euclidean errors shown in *Figures 3c and 4a* for a, and panels (**d-e**), for (**b-c**), respectively. Results from the KF algorithm (green) and the RE algorithm (blue) are compared for PC (lighter shades) and cPCF (strong lines). The diagonal gray areas indicate a $\sim (N_{ch})^{-0.5}$ proportionality. For simulating the underlying PC data, we used standard deviations of $\sigma_{op} = 0.1$ and $\sigma = 1$ and for the cPCF data additionally a ligand of brightness $\lambda_b = 5$. To facilitate the inference for the two more complex models, we assumed that the experimental noise and the single channel current are well characterized, meaning $i \sim \mathcal{N}(i|1, 0.01)$, $\sigma \sim \mathcal{N}(\sigma|1, 0.01)$ and $\sigma_{op} \sim \text{gamma}(\sigma_{op}|1, 100)$. In the models containing loops (last 2 columns), a prior was used to enforce microscopic-reversibility and set to $k_{25}^\star \sim \text{beta}(100, 100)$ multiplied by $k_1 = k_5 k_6 k_7 k_8 (k_2 k_3 k_4)^{-1} \cdot 0.995 + 0.01 \cdot k_1^\star$.

The online version of this article includes the following source data for figure 7:

**Source data 1.** The folder of the five-state model includes 15 sets of time traces in the interval $N_{ch} \in [10^3, 5 \cdot 10^5]$.

**Source data 2.** The folder of the six-state model includes 14 sets of time traces in the interval $N_{ch} \in [10^3, 10^6]$.

mathematical convenience. On this occasion, for the sake of the argument, $\text{beta}(1, 1)$ seems sufficient. A $\text{beta}(1, 1)$ prior is a uniform prior on the open interval $(0, 1)$. The estimated true success rate of the RE approach (blue) is $\approx 0.15$ and therefore far away from the success probability an algorithm should have when it is based on an exact likelihood of the data. In contrast, the posterior (green) of the true success probability of the KF resides with a large probability mass between the lower and upper limit of the success probability of an optimal algorithm (given the correct kinetic scheme). As both algorithms use the same prior distribution, the different performance is not induced by the prior.

Exploiting six one-dimensional posterior distributions does not necessarily answer whether the posterior is accurate in 6 dimensions but we can refine the used binomial setting. In *Figure 4d* $\mathbb{P}(\tilde{k}_{32}, \tilde{K}_{43})$, we see that 2-D marginal distributions can, due to their additional degree of freedom, twist

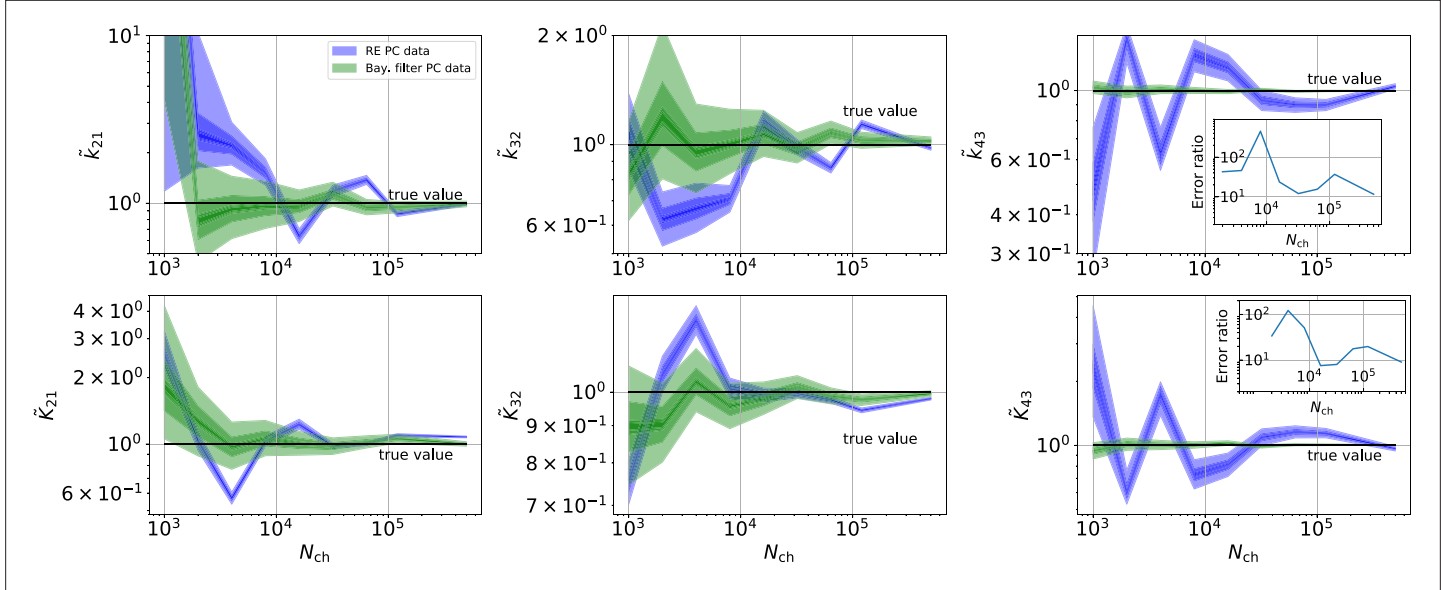

**Figure 8.** Revisiting the PC data obtained by the 4-state-1-open-state model shows that the KF succeeds to produce realistic uncertainty quantification, while the overconfidence problem (unreliable uncertainty quantification) of the RE approach remains. Comparison of a series of HDCIs shown as functions of $N_{ch}$ for each parameter of the rate matrix obtained by the KF (green) and the RE algorithm (blue). The differing shades of green and blue indicate the set of $(0.95, 0.6, 0.2, 0.1)$-HDCIs. Only the interval $N_{ch} > 2 \cdot 10^3$ in which all parameters are identified is displayed. The data are taken from the KF vs. RE benchmark of **Figures 3c and 7a** . The first row corresponds to three rates $k_{ij}$ the second row to the equilibrium constants $K_{ij}$. All parameters are normalized by their true value. The insets show the error ratios of the respective single parameter estimates. Note that the error ratios on the single-parameter level can be even of the order of magnitude of $10^2$. Thus, they can be much larger than the error ratios calculated from the Euclidean error if the errors of the respective parameters are small compared to other error terms in the Euclidean error **Equation 15** .The lowest Euclidean error for this kinetic scheme has cPCF data analyzed with the KF. (**Figure 7d**). A 6-state-1-open-states model with cPCF data has again an error ratio of the the usual scale (**Figure 7c**). As expected, the Euclidean error continuously increases with model complexity (**Figure 7d and e**). For PC data of the 6-state-1-open-states model even the likelihood of the KF is that weak (**Figure 7e**) that it delivers unidentified parameters even for $N_{ch} = 10^4$ and we can detect heavy tailed distributions up until $N_{ch} = 10^5$. Using RE on PC data alone does not lead to parameter identification, thus no error ratio can be calculated.

around the true value without covering it with HDCV (**Equation 14**) of reasonable size while simultaneously the two 1–D marginal distribution do cover it with a reasonable HDCI. In general, the KF posterior distribution has its mode much closer to the true value for various parameter combinations and it seems that the posterior is approximately multivariate normal. Further, we recognize that the probability mass of the reasonably sized HDCV of the KF posterior includes the true values whereas the HDCV from the RE does not. In 6 dimensions we lack visual representations of the posterior. Since we showed that both algorithms are consistent for a given identifiable model, we are looking for a way to ask whether the posterior is accurate (has the posterior distribution the right shape). We can answer that question by asking, how much probability mass around the mode (or around multiple modes) needs to be counted to construct a HDCV **Equation 14** which includes the true values. Then we can ask for $N_{set}$ data sets how often did we find the true values inside a volume $V(P)$ of a specific probability mass $P$ of the posterior distribution

$$\text{success} \sim \text{binomial}(N_{set}, P(V)). \tag{17}$$

An algorithm which estimates the parameters of the true process should fulfill this property simultaneously to being consistent. Otherwise credibility volumes or confidence volumes are meaningless. Noteworthy, that this is a empirical test of how sufficient the Bayesian filter and the RE approach hold frequentist coverage property of their HDCVs (**Rubin and Schenker, 1986**). We explain (Appendix 8) in detail how to quantify the overall shape and $n$-dimensional posterior and comment on its geometrical meaning. One way is to use an analytical approximation via the cumulative Chi-squared distribution (**Figure 5a and b**), The other way is to count the probability mass of $n$-dimensional histogram bins starting with the highest value until the first bin includes the true values (**Figure 5b**).

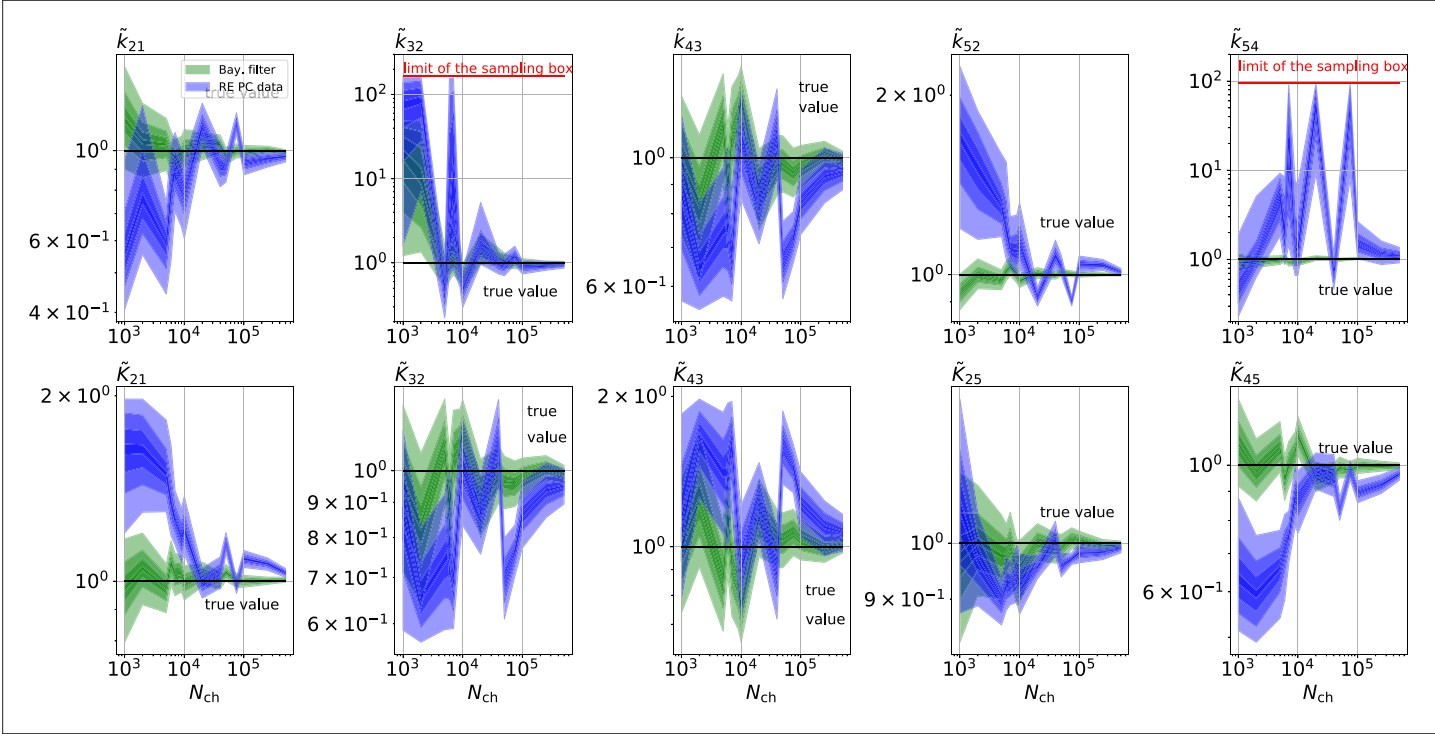

**Figure 9.** The HDCIs for PC data for a 5-state-2-open-states model show negligible bias for the KF with the true value being included. In contrast, the HDCE for RE approach frequently does not include the true value and in general appears biased and frequently leaves certain parameters unidentified. Comparison of a series of $(0.95, 0.6, 0.2, 0.1)$-HDCIs as functions of $N_{ch}$ for each parameter of the rate matrix obtained by the KF (green) and the RE algorithm (blue). The HDCIs correspond to the PC data displayed in *Figure 7b and d* . The first row corresponds to three rates $k_{ij}$ the second row to the equilibrium constants $K_{ij}$. All parameters are normalized by their true value. $\tilde{K}_{25}$ is because of the microscopic-reversibility prior a parameter which is strongly dependent on the other rates and ratios. Refer to the caption of *Figure 7* for details about the prior that enforces microscopic-reversibility. Thus, the deviations of $\tilde{K}_{25}$ are inherited from the other parameters. The rate $\tilde{k}_{54}$ is frequently not identified by the RE approach and only the limits of the sampling box confines he posterior.

Knowing how much volume/probability mass is needed to include the true rate matrix allows us to test whether all HDCVs constructed from the two probability distributions match the binomial distributions of the ideal model. For each data set and for each HDCV of a fixed probability mass, there are two possible outcomes: The true rate matrix is inside or outside of that volume. For a chosen HDCV with a fixed probability volume, as indicated by a gray space in *Figure 5b* , we count how many times the true matrix is included in the volume of that probability mass for each trail in a fixed amount of trials. Since the success is binomially distributed, we plot the expected mean of a perfect model $E[y] = N_{trials}P_{true}$ and binomial quantiles and compare them with the success rate found in our test runs (*Figure 5c*) for both algorithms with both methods to determine the posterior shape. The posterior of the KF distributes the probability mass in a consistent manner such that each volume includes the true rate matrix within the quantile range. In contrast, the RE approach fails for all data sets for all HDCVs (from 0 – 0.95 probability mass) and does not include the true values in one single case. Note, that all the binomial trials for each HDCV are made from the same set of data sets which explains the correlated deviation from the mean. For lower but realistic signal to noise ratios, where the fit quality decreases, for example by producing larger errors/wider posterior distributions (*Figure 6a*), the statistics of the HDCV from the RE approach improve but are still outperformed by the KF. In particular, in our tested case of realistic experimental noise we never find the true values within a 0.65-HDCV if the data are analyzed with a RE approach. Even for the highest noise level (*Figure 6b*), the probability mass of the KF posterior needed to include the true rate matrix remains almost always smaller then the posterior mass of the RE approach. That means that the posterior mass of the KF is much closer to the true value distributed than the posterior mass of the RE. With the KF we find the true rate matrix for one data set in small volume $P < 0.05$ around the mode. To achieve the same with the RE approach we need at least a probability mass of 0.3.

In the inset of *Figure 6a and b* we do not observe a trend, thus no indication that the RE approach has a better performance for large values $N_{ch}$ in this regard. This challenges the common argument that the RE approach should be equivalent to the KF for large $N_{ch}$ because the ratio of mean signal vs. the intrinsic binding and gating noise is so large. Thus, including the autocorrelation into the analysis is important even for unrealistic large $N_{ch}$. One possible explanation is model a signal-to-noise ratio which scales $\propto N_{ch}$. From the multinomial distribution both algorithms inherit mean signals which scale $\propto N_{ch}$ and variances which scale in the terms dominating for large $N_{ch}$ similarly with $\propto N_{ch}$. Thus, identical to the real signal, both algorithms model the scaling of the signal-to-noise ratio $\propto \sqrt{N}$. It is plausible, that both algorithms remain sensitive for the occurrence of autocorrelation of the noise even for largest signal-to-noise ratios. In *Figure 5c* we compare the Euclidean error of the pessimistic high white noise case with an over-optimistic low noise case. We see, that when increasing $N_{ch}$ there is a regime where the Euclidean error increases faster than $\sqrt{N_{ch}}^{-1}$ which we indicate with a coarse approximate fit $\propto N_{ch}^{-1}$. In that regime two effects happen simultaneously. First, the mean and the intrinsic fluctuations of the signal become more and more dominant over the experimental noise. Second, the standard deviation of intrinsic fluctuations becomes smaller relative to the mean signal. We speculate, that this produces together a learning rate which is faster than the usual asymptotic learning rate $\sqrt{N_{ch}}^{-1}$ of a regular model but relaxes asymptotically towards $\sqrt{N_{ch}}^{-1}$.

## Statistical properties of both algorithms for more complex models

We have seen in *Figure 3c* and *Figure 4a* that the RE and the KF algorithm are consistent estimators, while their error ratio (*Figure 7a*) seems to have no trend to approach 1 with increasing $N_{ch}$. Adding a second observable increases parameter accuracy and adds identifiability for both algorithms since less aspects of the dynamics need to be statistically inferred (*Figure 4a*). Furthermore, the second observable takes away much of the tendency (compare *Figure 4b* 1 – 6 with 8) of the RE approach to overinterpret (overfit) which leads to a shrinking of the error ratio $5.6 \pm 1.4$ for PC data to smaller values for cPCF data (*Figure 7a*) (red) which are on average still bigger than one, while the Euclidean error is reduced (*Figure 4a*). If we then keep the amount and quality of the PC/cPCF data but increase the complexity of the model which produced the data (*Figure 7b and d*) from a four-state to a five-state model (see kinetic schemes above *Figure 7a–c*), we see that for cPCF data the error ratio stays roughly the same (difference between *Figure 7a and b*). For PC data instead both algorithms deliver an unidentified $k_{21}$ for $N_{ch} \leqq 2 \cdot 10^3$ (defined as an improper posterior). For larger $N_{ch}$ the KF always identifies all parameters while the RE fails at $N_{ch} \in \{7000, 2000, 75000\}$ to identify $k_{54}$. Thus, the KF reduces the risk of unidentified parameters. To calculate the mean error ratio, we exclude the values were some of the parameters are unidentified in total that still amounts to $6.8 \pm 2.7$ thus the advantage of the KF (given all parameters are identified) might increase with model complexity for PC data. The lowest Euclidean error for this kinetic scheme has cPCF data analyzed with the KF. (*Figure 7d*). A 6-state-1-open-states model with cPCF data has again an error ratio of the the usual scale (*Figure 7c*). As expected, the Euclidean error continuously increases with model complexity (*Figure 7d and e*). For PC data of the 6-state-1-open-states model even the likelihood of the KF is that weak (*Figure 7e*) that it delivers unidentified parameters even for and we can detect heavy tailed distributions up until . Using RE on PC data alone does not lead to parameter identification, thus no error ratio can be calculated.

Consistent with our findings, fluorescence data itself, should lower the advantage of the KF compared to PC data simply by signal-to-noise arguments. The stochastic aspect of the ligand binding is usually more dominated by the noise of Photon counting and background noise than the stochastic gating is dominated in current data by experimental noise. In terms of uncertainty quantification the advantage of the KF with cPCF varies with the model complexity (see, Appendix 9).

Besides analyzing what causes the changes in the Euclidean error (*Figure 7a and b*) at the single parameter, we now investigate whether the posterior is a proper representation of uncertainty. Thus, we look back at the HDCIs. The HDCIs of the 4-state-1-open-state (*Figure 8*) of the PC data from *Figure 3* reveal an exacerbated over-confidence problem of the RE approach (blue) compared to cPCF-data (*Figures 4b1–6*). This, underlines our conclusion of *Figures 5 and 6* that the Bayesian posterior sampled by the RE approach is misshaped. As a consequence a confidence volume derived from the curvature at the ML estimate of the RE algorithm understates parameter uncertainty. A possible way for ML methods to derive correct uncertainty quantification is by using bootstrapping

data methods (*Joshi et al., 2006*). Furthermore, the error ratios of each single parameter from its true value in the last column $\tilde{k}_{43}\tilde{K}_{43}$ strongly increased their magnitudes (insets *Figure 8*). Even error ratios of $5 \cdot 10^2$ are possible. Note, that the way we defined *Equation 15* suppresses the influence of the smaller parameter errors in the overall error ratio. Thus the advantage (error ratio) of the KF over RE approach for a single parameter can be much larger or lower compared to the error ratio derived from the Euclidean error if the respective parameter is contributing less to the Euclidean error. The posterior of the KF (green) seems to be unbiased after the transition into the regime $N_{ch} > 2 \cdot 10^3$ where all parameters are identified. Similarly, for the RE algorithm there is no obvious bias in the inference. If we use the RE algorithm and change from the four-state to the five-state model (PC data from *Figure 7b*), bias occurs (*Figure 9*) in many inferred parameters, even for the highest $N_{ch}$ investigated. *Milescu et al., 2005* showed that one or the reason of the biased inference of the RE approach is its ignorance of autocorrelation of the intrinsic noise. We add here that the bias problem clearly aggravates with an increased model complexity. It is even present in unrealistically large patches which in principle could be generated by summing up $10^2$ time traces with $N_{ch} = 10^3$. In contrast, the KF algorithm reveals that its parameter inference is either unbiased or at least much less biased in the displayed $N_{ch}$ regime. Furthermore, for both algorithms the position of the HDCI relative to the true value is for some parameters highly correlated, which corresponds to the correlation between optima of the ML method of *Milescu et al., 2005* ; *Moffatt, 2007*.

As a side note, unbiased parameter estimates are a highly desirable feature of an inference algorithm. For example, with a bias in the inference, repeated experiments do not lead to the true value if the arithmetic mean of the parameter inferences is taken. With bias even bootstrapping methods fail to produce reliable uncertainty quantification. Due to the variation of the data the $k_{54}$ parameter is either identified in some neighbourhood of the true value or complete unidentified (*Figure 9*), if the RE algorithm is used. The unidentified $k_{54}$ occurs even at high-quality data such as $N_{ch} = 7.5 \cdot 10^4$. Only because of the nonphysical prior (*Figure 9*) of $k_{54}$ induced by the limits of the sampling box of the sampling algorithm the posterior appears to be proper but is in fact either unidentified or or more than two orders of magnitude away from the true value. For the same data using the KF did not result in any unidentified parameters. Note, that comparable inference pathologies such as multimodal distributions of inferred parameter were also reported for the maximum likelihood RE algorithm for low quality PC data or too simple stimulation protocols (*Milescu et al., 2005*).

In conclusion, the two different perspectives on parameter uncertainty: On the one hand distributions of ML estimates due to the random data (*Milescu et al., 2005* ; *Moffatt, 2007*) and the Bayesian posterior distribution loose their tightly linked (and necessary) connection if the RE algorithm is used. Thus, KF robustifies also ML inferences of the rate matrix. Our findings are consistent with the findings for gene regulatory networks (*Gillespie and Golightly, 2012*) which show that RE approaches deliver a too narrow posterior in contrast to stochastic approximations which deliver an acceptable posterior compared to the true posterior (defined by a particle filter algorithm). On the data side of the inference problem adding cPCF data eliminates the bias, reduces the variance of the position of the HDCI and eliminates unidentified parameters (*Appendix 9—figures 1 and 2*) for both investigated algorithms. This advantage increases with model-complexity.

For the five-state and six-state model, we applied microscopic-reversibility (*Colquhoun et al., 2004*). We enforced it by hierarchical prior distribution (Materials and methods *Equation 60*) whose parameters can be chosen such that they allow only arbitrarily small violations of microscopic-reversibility. But the prior distribution can also be used to enforce some softer regularization around microscopic-reversibility. Thus, we can transfer the usually strictly applied algebraic constraint (*Salari et al., 2018*) of microscopic-reversibility to a constraint with scalable softness. In that way we can model the lack of information if microscopic-reversibility is exactly fulfilled (*Colquhoun et al., 2004*) by the given ion channel instead of enforcing the strict constraint upon the model.

## Prior critique and model complexity

In the Bayesian framework, the likelihood of the data and the prior generate the posterior. Thus, the performance of both algorithms can be influenced by appropriate prior distributions. We used a uniform prior over the rate matrix which is not optimal. Note, that uniform priors are widely used by several reasons. They appear to be unbiased, and are assumed to be a 'no prior' option (which they are not). This is true for location parameters like mean values. In contrast, for other parameters, such

as scaling parameters like rates or variances, a uniform prior adds bias to the inference towards faster rates (*Zwickl and Holder, 2004*). We suspect, that for the PC data even in the simplest model discussed here the lower data quality limit below which we detected unidentified parameters (improper posteriors) is caused by the uniform prior. This lower limit for the KF also increases with the complexity of the model from $N_{ch} < 2 \cdot 10^3$ for the foue-state model till $N_{ch} \leq 2 \cdot 10^4$ for 6-state-1-open-state model. Note, that it is hardly possible to fit the 6-state-1-open-state model with the RE approach for the same amount of PC data. We observe cPCF data eases this problem because the likelihood becomes more concentrated for all parameters. The likelihood dominates the uniform prior. Nevertheless, for most parts of the paper we used a uniform prior over the rates and equilibrium constants to be comparable with the usual default method: a plain ML which influences our results in data regimes in which the data is not strong enough to dominate the bias from the uniform prior. Thus, both algorithms perform better with smarter informative or at least unbiased prior choices for the rate matrix.

In principle, to rule out an influence of the prior, unbiased priors should be used for the rates. The standard concept for unbiased least informative priors is to construct a Jeffreys prior *Jeffreys, 1946* for the rate matrix which is, however, beyond the scope of the paper.

## The influence of the brightness of the ligands of cPCF data on the inference

To evaluate the advantage of cPCF data *Biskup et al., 2007* with respect to PC data only (*Figure 10*), we compare different types of ligands: Idealized ligands with brightness $\lambda_b$, emitting light only when bound to the channels, 'real' ligands which also produce background fluorescence when diffusing in the bath solution (Appendix 5) and current data alone. For datasets including fluorescence, the increased precision for the dissociation rate of the first ligand, $k_{2,1}$, is that strong that the variance of the posterior $\mathbb{P}(k_{2,1}, k_{3,2})$ nearly vanishes in the combined plot with the current data (nearly all probability mass is concentrated in a single point in *Figure 10a*). The effect on the error of the equilibrium constants $K_i$ is less strong. Additionally, the bias is reduced and even the estimation of $N_{ch}$ is improved. The brighter the ligands are, the more the posterior of the rates decorrelates, in particular $\mathbb{P}(k_{2,1}, k_{3,2})$ (*Figure 10a*). All median estimates of nine different cPCF data sets (*Figure 10b*) differ by less than a factor 1.1 from the true parameter except $k_{3,2}$, which does not profit as much from the fluorescence data as $k_{2,1}$ (*Figure 10c*). The 95th percentiles, $l_{95}$ of $\mathbb{P}(k_{2,1})$ and $\mathbb{P}(K_1)$ follow $l_{95} \sim 1/\sqrt{\lambda_b}$. Thus, with increasing magnitude of ligand brightness $\lambda$, the estimation of $k_{2,1}$ becomes increasingly better compared to that of $k_{3,2}$ (*Figure 10c*). The posterior of the binding and unbinding rates of the first ligand contracts with increasing $\lambda_b$. The $l_{95}$ percentiles of other parameters exhibit a weaker dependency on the brightness ($l_{95} \sim \lambda^{-0.1}$). For $\lambda_b = 0.01$ photons per bound ligand and frame, which corresponds to a maximum mean signal of 20 photons per frame, the normal approximation to the Poisson noise hardly captures the asymmetry of photon counting noise included in the time traces. Nevertheless, $l_{95}$ decreases about ten times when cPCF data are used (*Figure 10c*). The estimated variance of

$$r(t_i) := \frac{y(t_i) - \left( \mathbf{H}\mathbb{E}[\mathbf{n}(t_i)] \right)}{\sqrt{\text{var}[y(t_i)]}} \tag{18}$$

with the mean predicted signal $\mathbf{H}\mathbb{E}[\mathbf{n}(t_i)]$, for PC or cPCF data is $\sigma^2(r_i) \approx 1$ (*Figure 10d*) which means that the modeling predicts the stochastic process correctly up to the variance of the signal. Note that the mean value and covariance of the signal and the state form sufficient statistics of the process, since all involved distributions are approximately multivariate normal. The fat tails and skewness of $\mathbb{P}(k_{21})$ and $\mathbb{P}(k_{12})$ arises because the true model is too flexible for current data without further prior information. The KF allows to determine the variance (*Figure 10e*) of the open-channel current noise for $\sigma_{op} = 0.1i$. Adding fluorescence data has roughly the same effect on the estimation of $\sigma_{op}$ like using five times more ion channels to estimate $\sigma_{op}^2$.

## Sensitivity towards filtering before the analog-to-digital conversion of the signal

On the one side, every analog signal to be digitized needs analog filtering for antialiasing according to the Nyquist theorem. On the other side, every analog filter does not only suppress unwanted white noise but also distorts the dynamics (*Figure 11a*) of the signal of interest (*Silberberg and Magleby,*

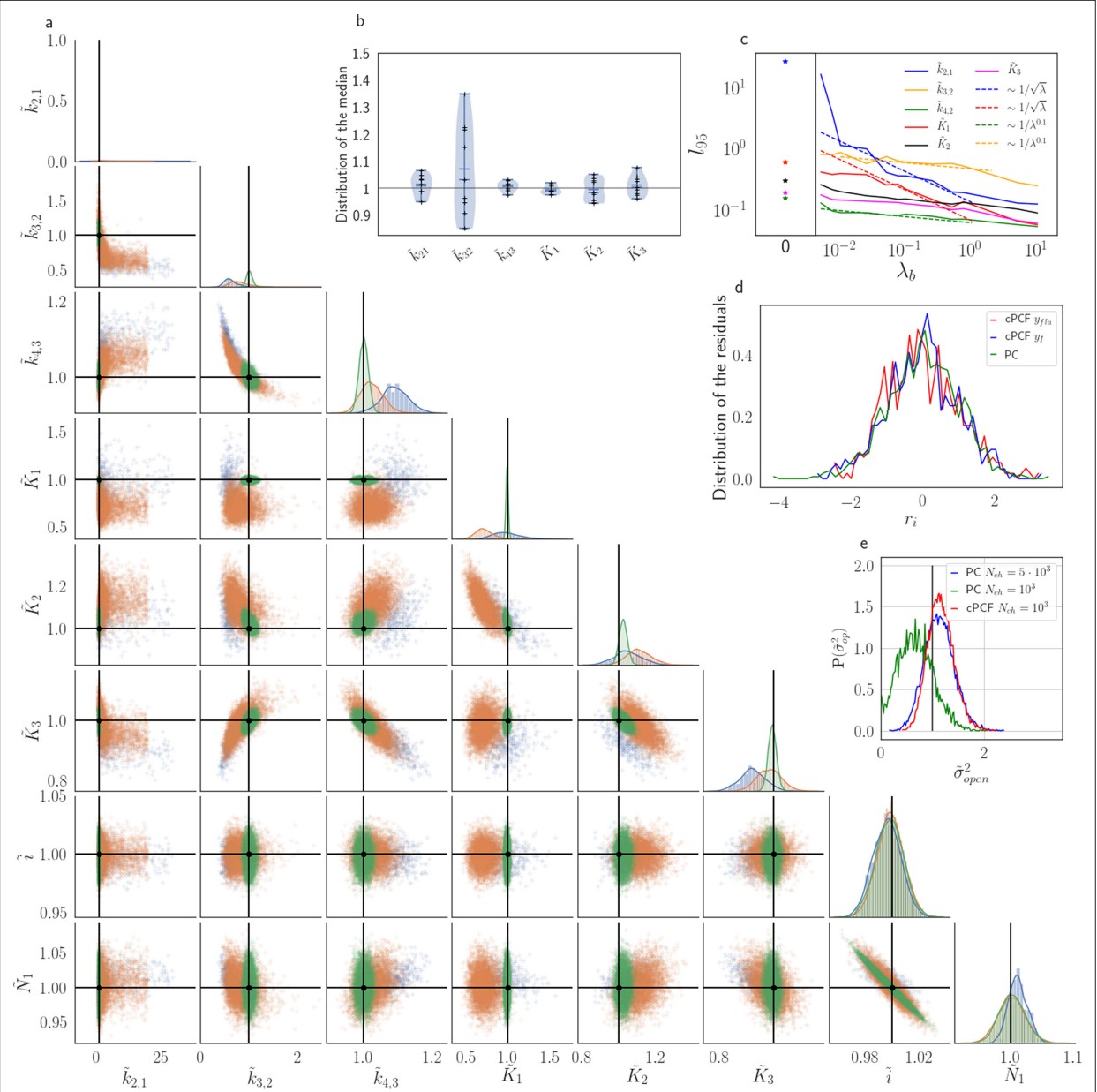

**Figure 10.** The Benchmark of the KF for PC versus cPCF data with different bright ligands shows that even adding a weak fluorescence binding signal can add enough information to identify before unidentified parameters. (**a**) Posteriors of PC data (blue), cPCF data with $\lambda_b = 0.00375$ (orange) and cPCF data with $\lambda_b = 0.375$ (green). For the data set with $\lambda_b = 0.375$, we additionally accounted for the superimposing fluorescence of unbound ligands in solution. In all cases $N_{ch} = 10^3$. The black lines represent the true values of the simulated data. The posteriors for cPCF $\mathbb{P}(k_{2,1}, k_{3,2})$ are centered around the true values that are hardly visible on the scale of the posterior for the PC data. The solid lines on the diagonal are kernel estimates of the probability density. (**b**) Accuracy and precision of the median estimates visualized by a violin plot for the parameters of the rate matrix for 5 different data sets. Four of the five data sets are used a second time with different instrumental noise, with $\lambda_b = 0.375$ and superimposing bulk signal. The blue lines represent the median, mean and the maximal and minimal extreme value. (**c**) The 95th percentile of the marginalized posteriors vs. $\lambda_b$ normalized by the true value of each parameter. A regime with $l_{95} \sim 1/\sqrt{\lambda}$ is shown for $k_{2,1}$ and $K_1$, while other parameters show a weaker dependency on the ligand brightness. (**d**) Histograms of the residuals $r$ of cPCF with $\lambda_b = 2.5 \cdot 10^{-3}$ data and PC data. The randomness of the normalized residuals of the cPCF or PC data is well described by $r_i \sim \text{normal}(0, \sigma_{\text{res}}^2 = 1)$. The estimated variance is $\sigma_{\text{res}}^2 = 0.98 + 0.26$. Note that the fluorescence signal per frame is very low such that the normal approximation to Poisson counting statistics does not hold. e, Posterior of the open-channel noise $\mathbb{P}(\sigma_{\text{op}}^2/\sigma_{\text{op,true}}^2)$ for PC data with $N_{ch} = 10^3$ (green) and $N_{ch} = 10^5$ (blue) as well as for cPCF data with $N_{ch} = 10^3$ (red) with $\lambda_b = 0.375$. We assumed as prior for the instrumental variance $\mathbb{P}(\sigma^2) = \mathcal{N}(1, 0.01)$.

The online version of this article includes the following source data for figure 10:

*Figure 10 continued on next page*

*Figure 10 continued*

**Source data 1.** Five different sets of time traces of panel b.

**Source data 2.** Eighteen sets of time traces of panel c.

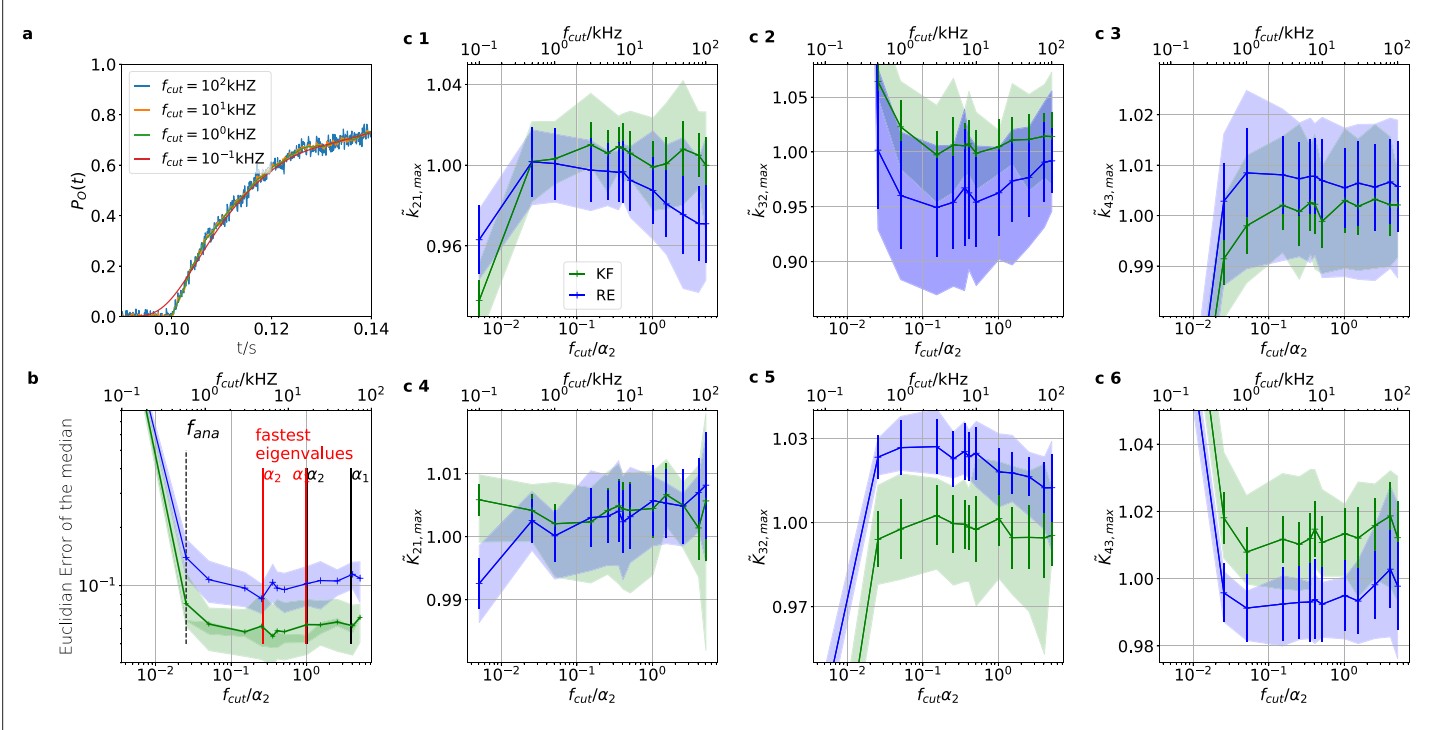

**Figure 11.** The KF is robust against moderate analog filtering of the current signal. High (Bayesian) sampling frequencies and minimal analog filtering does minimize bias which otherwise deteriorates parameter identification. In order to mimic an analog signal before the analog-to-digital conversion we simulated seven different 100 kHz signals which were then filtered by a digital fourth-order (4 pole) Bessel filter. The activation curves were then analyzed with the Bayesian filter at 125 Hz and the deactivation curves at sampling rates between 166-500 Hz. We chose for the analog signal $\sigma_{\exp}/i = 10$, $\sigma_{\mathrm{op}}/i = 0.1$, thus a stronger background noise, and we set the mean photon count per bound ligand as $\lambda_{\mathrm{b}} = 5$. For the ensemble size we choose $N_{\mathrm{ch}} = 10^3$. (**a**) Current time trace filtered with different $f_{cut}$. Except for 100 Hz (red) the signal distortion is visually undetectable. Nevertheless, the invisible signal distortions from analog filtering are problematic for both algorithms. (**b**) Estimate of the distribution mean Euclidean error of the median of the posterior vs. the cut-off frequency of a 4 pole Bessel filter (upper scale is in units of kHz) or scaled to the channel time scale (lower scale, see text). The fastest two eigenvalues $\alpha_{1,2}/\alpha_2$ for the highest ligand concentration are indicated by the black vertical lines. The fastest ratios $\alpha_{1,2}/\alpha_2$ for the next smaller ligand concentration are indicated by the red vertical lines. The slowest eigenvalue ratio $\alpha_3/\alpha_2$ at the highest ligand concentration is beyond the left limit of the x-axis. The solid line is the mean median of five data sets of the respective RE posterior (blue) and KF posterior (green). The green shaded area indicates the 0.6 quantile (ranging from the 20th percentile till the 80th percentile), demonstrating the distribution of the error of the posterior median due to the randomness of the data. (**c**) 1–3, Accuracy (bias) and precision of the maxima of the posterior $\tilde{k}_{\max,ij}$ of the posterior maxima of the rates vs. the cut-off frequency of a Bessel filter. The shaded areas indicate the 0.6 quantiles (ranging from the 20th percentile till the 80th percentile) due the variability among data sets while the error bars show the standard error of the mean. The deviation of the mean from the true value is an estimate of the accuracy of the algorithm while the quantile indicates the precision. (**c**) 4–6, Accuracy and precision of the maxima of the posterior $\tilde{K}_{\max,ij}$ of the posterior maxima of the corresponding equilibria vs. the cut-off frequency of a Bessel filter.

The online version of this article includes the following source data for figure 11:

**Source data 1.** Representative data set of cPCF data whose current has been filtered with decreasing $f_{cut}$.

**Source data 2.** Representative data set of cPCF data whose current has been filtered with decreasing $f_{cut}$.

**Source data 3.** Representative data set of cPCF data whose current has been filtered with decreasing $f_{cut}$.

**Source data 4.** Representative data set of cPCF data whose current has been filtered with decreasing $f_{cut}$.

**Source data 5.** Representative data set of cPCF data whose current has been filtered with decreasing $f_{cut}$.

**Source data 6.** Representative data set of cPCF data whose current has been filtered with decreasing $f_{cut}$.

*1993*). Therefore, (*Qin et al., 2000*) recommend to avoid analog filtering as much as possible in single-channel analysis and let the HMM analyze the data in the rawest available form, even with simultaneous drift correction (*Sgouralis and Pressé, 2017a*). One can also expect that analog filtering of a macroscopic signal is harmful for the inference of the KF and the RE approach. For the CCCO model considered herein we investigated the mean behavior (accuracy and precision) of the posterior of both algorithms with seven data sets (simulated at 100 kHz to mimic an analog signal). A digital fourth-order Bessel filter (*Virtanen et al., 2020*) was then applied. The maximum analysing frequency $f_{ana}$ of the KF used is $100 - 400$ Hz to be comparable to cPCF setups. The slower frequency at which the Bayesian filter analyzes the data is necessary because the applied Bessel filter has caused additional time correlations in the originally white noise of the signal. Thus, an all-data-points fit would immediately violate the white noise assumption of *Equation 4* which we restore by analyzing at a much lower frequency. We then let the time scales of the induced time correlations become larger and larger by decreasing $f_{cut}$. Physically, the absolute cut-off frequency $f_{cut}$ is irrelevant; what matters is the magnitude of $f_{cut}$ relative to $f_{ana}$ and to the eigenvalues $\alpha_i$ of the ensemble (see, Appendix 3), since the eigenvalues determine the time evolution of the mean ensemble state, the autocorrelation, and Fourier spectrum of the fluctuations around the equilibrium distribution (*Colquhoun et al., 1997b*). The eigenvalues depend on the ligand concentration such that for a four-state model for each ligand concentration there are three relevant time scales $-1/\alpha_i$ (where $i = 2, 3, 4$) plus the equilibrium solution which satisfies $\alpha_1 = 0$. For 10 different time series $3 \cdot 10 + 3$ the outcome is to have different values of $\alpha_i$. Each eigenvalue is the inverse of the time constant of an exponential decay (see, Appendix 3). For this reason, we normalize in the following (*Figure 11*) the cut-off frequencies by $\alpha_2$ at the highest ligand concentration. We analyze the arithmetic mean from 7 different data sets of the median of the posterior of the rate matrix. The mean Euclidean error of the median (*Figure 11b*) and a series of quantiles demonstrate that overall the error of the mean median of the posterior KF (green) is smaller than that obtained by the RE. For unfiltered data, the accuracy of the mean median of the KF is increased by $\approx 1.6$. Based on the Euclidean error both algorithms benefit slightly from careful analog filtering for $f_{cut}/\alpha_2 \geq 1$ while the offset remains rather constant. A strong negative effect of analog filtering starts for both algorithms around $f_{cut} \approx 1$ kHZ. This is induced by $f_{cut} \to f_{ana}$ (see, Appendix 10). In contrast, based on the level of each individual parameter of the rate matrix (*Figure 11c* 1–6) the bias induced by analog filtering immediately starts with $f_{cut} = 70$ kHz (*Figure 11c* 1–3). Note, that visual inspection of the signal (*Figure 11a*) does not reveal signal distortions $f_{cut} \geq 10$ kHz though they are detected by both algorithms. For unfiltered data, the maximum of the posterior for the RE approach is a biased estimate $E[\theta_{ME}] \neq \theta_{true}$ for at least the parameters $\tilde{k}_{21}, \tilde{K}_{21}, \tilde{K}_{32}$ of the true value $\theta_{true}$, which is explained (*Milescu et al., 2005*) by the fact that RE approaches ignore the autocorrelation of the intrinsic noise. Additionally, the data indicate that for $\tilde{K}_{43}$ the maximum of the posterior is even for the KF a biased estimate which we interpret as limitations induced by the fact that the mean vector and covariance-matrix do not constitute sufficient statistics as soon as Poisson distributed photon counting or open-channel noise blurs the signal. For the RE approach, the additional bias induced by the analog filter on the mean maximum of all parameters of the posterior starts with $f_{cut} \approx 70$ kHz or, in other words, at the fastest time scale in the whole data set. The total bias in the estimate is reduced for $k_{21}$ with the additional bias from the analog filtering but increased for $k_{32}$ which for the Euclidean error leads at first to a small increase in accuracy. The KF is more robust towards analog filtering, as the results alter less with $f_{cut}$ (given a reasonable $f_{cut}$), and less biased for unfiltered data in the estimates of these parameters. On the one hand, the Euclidean error shrinks for $f_{cut} > 10$ kHz (*Figure 11b*). On the other hand, on the single-parameter level (*Figure 11c* 1–6), the parameter estimates pick up bias due the analog filtering even for high filter frequencies, in particular for the RE approach. Only for $k_{43}$ the KF is more biased than the RE approach.

The KF is the unique minimal variance Bayesian filter for a linear Gaussian process (*Anderson and Moore, 2012*) which means, given that the assumptions of the KF are fulfilled by the true process of interest, the KF is mathematically proven the best model-based filter to apply. Consequently, analog filtering does not provide an advantage unless it removes specific high frequency external noise sources (colored noise). We demonstrate (Appendix 10) this for PC data and varied $f_{cut}$ and $f_{ana}$. On the downside, increasing $f_{ana}$ makes the results of both algorithms more fragile if $f_{cut} \gg f_{ana}$ does not hold. Thus, the critical edge in *Figure 11b* is indeed induced by $f_{cut}$ approaching $f_{ana}$. This suggests that the white noise assumption of both algorithms is violated. On the upside, if $f_{cut} \gg f_{ana}$ is given, the

KF with an order of magnitude higher $f_{ana}$ has a reduced bias of up to 20% for $f_{cut} \to \infty$ for individual parameters compared to the KF with lower $f_{ana}$. Additionally, a higher $f_{ana}$ reduces the variance. To reduce the bias of parameter estimates to a minimum, the experimental design offers two remedies, either doing cPCF experiments with additional discussed advantages or using the KF at a high $f_{ana}$ with even much higher $f_{cut}$.

By theoretical grounds a further argument for doing less analog filtering is that this benchmark analyzes data of a finite state Markov process, which is a coarse proxy for the true process. In reality, relaxation of a protein is a high-dimensional continuous-state Markov process with infinitely many relaxation time scales (eigenvalues) (*Frauenfelder et al., 1991*) which, however, might be grouped in slower experimentally accessible and non-accessible faster time scales (*Noé et al., 2013*). With larger data sets of higher quality from better experiments, the faster time scales might become accessible if they are not distorted by analog filtering. In conclusion, deciding on a specific kinetic scheme and inferring its parameters means finding a model which accommodates in the best way to the set of observed eigenvalues. Analog filtering hampers the RE, KF or HMM forward-backward algorithm (*Qin et al., 2000*) to correctly describe the faster time scales.

## Error due to finite integration time of fluorescence data

So far, we idealized the fluorescence data integration time as being instantaneously relative to the time scales of ensemble dynamics. In real experiments, the fluorescence signal of cPCF data has orders of magnitude longer minimal integration time $T_{int}$ (time to record all voxels of a frame) or maximal integration frequency $f_{int} = 1/T_{int}$, than the possible sampling frequency of current recordings. We mimic the finite integration time

$$y_{digital}(t_i) = \int_{t_{start}}^{t_i = t_{start} + T_{int}} y_{analog}(t)dt \approx \sum_{j \in [t_{start}, t_{start} + T_{int}]} y(t_j)\Delta t \tag{19}$$

by summing with a sliding window the 100 kHz signal including the white noise to obtain data at an effectively lower sampling frequency (*Figure 12a*). Additionally we set the Bessel filter for the current data to $f_{cut}/\alpha_2 = 4.59$ or $f_{cut} = 90$ kHz. The fastest used analysing frequency is $f_{ana} = 500$ Hz. We scale mean photo brightness $\lambda_b$ and background noise down such that the signal-to-noise ratio of the lower integration frequency data is the same as of the high-frequency data $\lambda_b/T_{int} = const$. We do that in order to separate the bias from the finite integration time from other effects such as a better signal to noise ratios for each integrated point. Note that we only analyzed the plot until $f_{int} = f_{ana}$. Both algorithms incur very similar bias due to the finite integration time (*Figure 12b*). The KF (green) is more precise for high integration frequencies $f_{cut}/\alpha_2$ until $f_{cut}/\alpha_2 \approx 0.08$ then the RE approach becomes more robust. Similar to Bessel-filtered current data (*Figure 11b*) on the single parameter level the systematic deviations start early for example $f_{int} = 10$ kHz for $K_{21}$ (*Figure 12c4*). Possibly the systematic deviations start (*Figure 12c2*) already at $f_{int} = 50$ kHz for $k_{32}$. The sudden increase of the Euclidean error (*Figure 12b*) of the mean median at $f_{cut}/\alpha_2 \approx 0.2$ occurs in this case not due to $f_{int}$ approaching $f_{cut}$ but due to $f_{int} \lesssim \alpha_{1,2}$ for many ligand concentrations. To show this we plot the results of the fitting of five different data sets without including the highest 4 ligand concentrations (red) which means the largest eigenvalues are much smaller (*Figure 12b,C1-6*). Additionally, we keep $f_{int} = const$. Although fluctuations of the posterior medians are higher, the KF becomes robust. Note, that the fastest eigenvalues of these reduced data sets are indicated by the blue bars (*Figure 12b and c4*). Based on the Euclidean error (*Figures 11a and 12a*) the robustness of both algorithms against the cut-off frequency is compared with the robustness against the integration frequency found to be about an order of magnitude higher. That is related to a specific detail of the model used: the binding reaction, corresponds to the fastest time scales of the overall dynamic (difference between *Figure 1b and c*), which is exposed by the fluorescence signal. Thus, kinetic analysis of any data should make sure that the corresponding frequency of the most dominant timescales of the time series are much slower than the respective $f_{int}f_{cut}$ independently of the investigated algorithms.

## Conclusions

We generalized the filter equations (Methods *Equation 37*, 38d, 57, 58 and 59) of the KF for analyzing the gating and binding dynamics of ligand-gated ion channels with a realistic signal-generating model for isolated patch-clamp (PC) and confocal patch-clamp fluorometry (cPCF) data including open-channel noise, photon-counting noise and background noise. Any other type of linear kinetic scheme

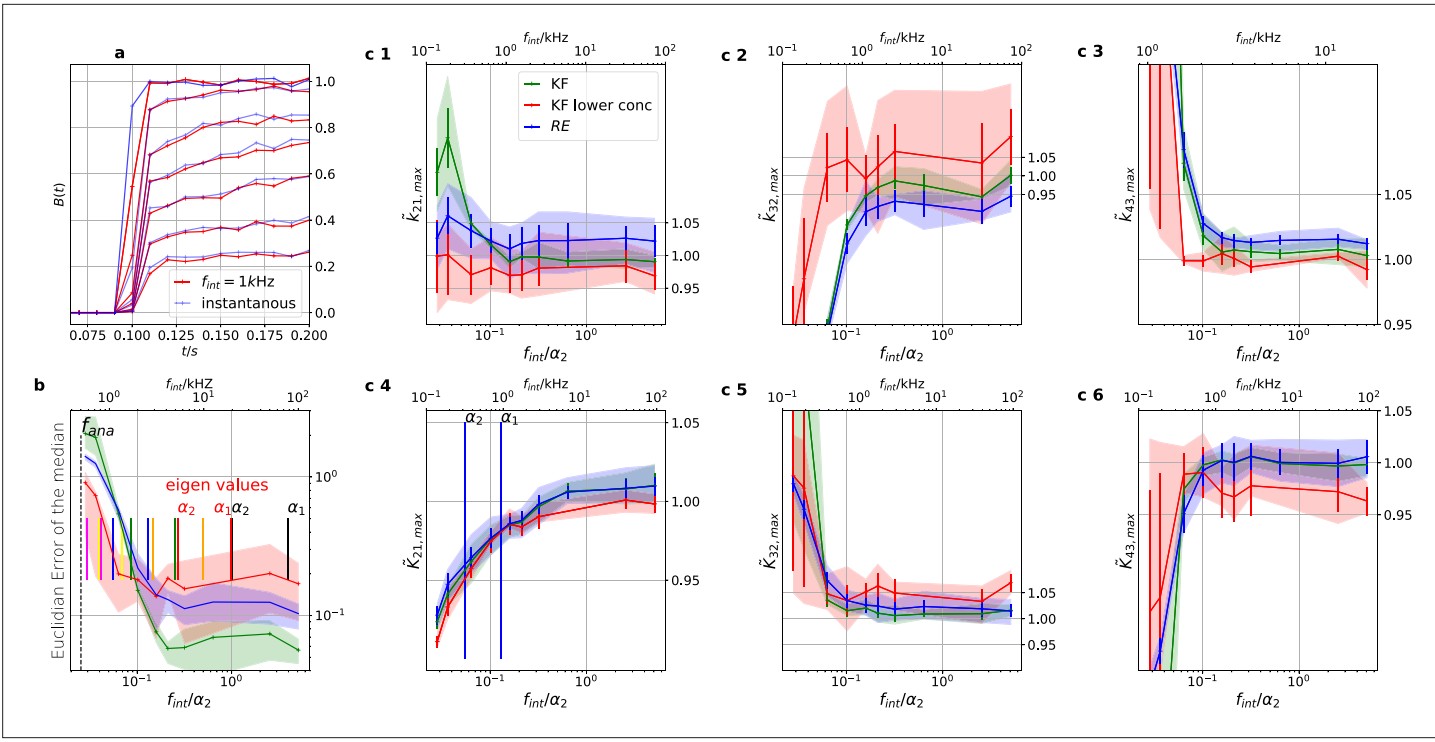

**Figure 12.** Finite integration time of fluorescence recordings acts also as a filter. Thus the sampling should be faster than the fastest eigenvalues to avoid biased results. We simulated five different 100 kHz cPCF signals. All forms of noise were added and then the fluorescence signal was summed up using a sliding window to account for the integration time to produce one digital data point. The activation curves were then analyzed with the Bayesian filter at 125 Hz and the deactivation curves at $166 - 500$ kHz, see caption of *Figure 8*. We plot the 0.6-quantile (interval between the 20th and the 80th percentile) to mimic ±one standard deviation from the mean as well as the mean of the distribution of the maxima of the posterior for different data sets. (Note, this is not equivalent to the mean and quantiles of the posterior of a single data set.). The quantiles represent the randomness of the data while the error bars indicate the standard error of the mean maximum of the posterior. Blue (RE) and green (KF) indicate the two algorithms with the standard data set while red (KF) shows examples that use only the six smallest ligand concentrations for the analysis in order to limit the highest eigenvalues. a, Instantaneous probing of the ligand binding (blue) compared with a probing signal which runs at $f_{\text{int}} = 1$ kHz. The integrated brightness of the bound ligand is $\lambda_b = 5$ photons per frame. Although the red curves seem like decent measurements of the process except for the highest two shown ligand concentrations, the mean error is roughly an order of magnitude worse than for $f_{\text{int}} = 10$ kHz. Note, that for visualization we plot at a higher frequency than the Kalman filter analyzed the data. b, Estimate of the distribution of the (Euclidean error of the mean median of the posterior) vs. the scaled integration frequency $f_{\text{int}}/\alpha_2 = 1/(\alpha_2 \cdot T_{\text{integration}})$. We use integration frequency instead of the integration time to make the plot comparable to the Bessel filter plot. The solid line is the mean median of five data sets of the respective KF posterior (green, red) and RE posterior (blue). The shaded areas indicate the 0.6-quantile which visualizes the spread of the distribution of point estimates. The two fastest time scales (eigenvalues) at the highest ligand concentration are indicated by the vertical black lines, the time scales of the next lower ligand concentrations with the red vertical lines. c 1–3, Accuracy (bias) and precision of the maxima of the posterior $k_{ij,max}$ rates vs. the integration frequency. c 4–6, Accuracy and precision of the maxima of the posterior $K_{ij,max}$ of the corresponding equilibria vs. the cut-off frequency of a Bessel filter.

The online version of this article includes the following source data for figure 12:

**Source data 1.** The data folder contains one data set of cPCF data.

**Source data 2.** The data folder contains one data set of cPCF data.

**Source data 3.** The data folder contains one data set of cPCF data.

**Source data 4.** The data folder contains one data set of cPCF data.

**Source data 5.** The data folder contains one data set of cPCF data.

**Source data 6.** The data folder contains one data set of cPCF data.

**Source data 7.** The data folder contains one data set of cPCF data.

(e.g. for voltage-dependent channels) and signal can be applied as long as the characteristics of the signal are sufficiently described by normal distributions. Our approach is derived by approximating the chemical master equation of a first order chemical reaction network (which ion channel experiments usually are) which is exact up to the second statistical moment. For first-order chemical reaction

networks, the linear noise approximation (*Wallace et al., 2012*) are exact up to the second moment too (*Grima, 2015*). Thus, we can conclude that our Bayesian filter uses a time integrated version of the linear noise approximation. To our understanding of *Wallace et al., 2012* our approach is thus equivalent to approaches based on the chemical Langevin or Fokker-Planck equations (*Gillespie, 2002*). Consequently, this also makes the considerations of the quality of the chemical Langevin equation as an approximation (*Gillespie, 2000*) of the chemical master equation valid for our approach. Compared to previous attempts *Moffatt, 2007*, this mathematical generalization is necessary (*Figure 3b*) in order to use Bayesian filters on macroscopic PC or cPCF data. With our algorithm, we demonstrate (*Figures 3c and 7*) that the common assumption that for large ensembles of ion channels simpler deterministic modeling by RE approaches is on par with stochastic modeling, such as a KF, is wrong in terms of Euclidean error and uncertainty quantification (*Figures 5a–c , and 6a–b*).

Enriching the data by fluorescence-based ligand binding reveals two regimes. In one regime, the two-dimensional data increase the accuracy of the parameter estimates up to $\approx$10-fold (*Figure 4a and c*). In the other regime of lower channel expression, enriching the data by the second observable, makes non-identified parameters to identified parameters. The second observable in cPCF data decreases the overfitting tendency (*Figure 4a, b and d*) of the RE approach on the true process. Thus, in this regard the advantage of the KF becomes smaller. However, by exploiting Bayesian HDCV we gain a second perspective: We show for various signal-to-noise ratios (*Figures 5a–c , and 6a–b*) that the posterior sampled by a RE approach never covers the true values within a reasonable HDCV. Thus, the central feature of Bayesian statistics, exact uncertainty quantification by having the full posterior, is meaningless in combination with an RE approach (considering the type of data and set of signal-to-noise ratios that we tested). This even holds true for very pessimistic signal-to-noise assumptions *Figure 6b*. If HDCVs based on an RE approach cannot be trusted, the same applies to confidence volumes based on the curvature of the likelihood. This is not the case for the KF which delivers properly shaped posteriors (*Figures 6a–c , and 5a–c*). Increasing the model complexity, at unchanged PC data quality (*Figure 7*) shows that the RE approach displays unidentified rates even for large ion channel ensembles while our approach identified all parameters for the same data. We also investigated the robustness of both algorithms against the cut-off frequency of a Bessel filter (*Figure 11*) and showed the overall superior robustness of the KF against errors of analog filtering compared to the RE approach. Analog filtering has its limitations due to distorting the higher frequencies of the Fourier spectrum of the signal. Thus, one should let the KF sample as fast as possible, with a cut-off frequency of at least one order of magnitude higher than the sampling frequency of the KF.

Similar to the Bessel filter, the KF is more robust than the RE approach against errors due to the finite integration time. Nevertheless, it is crucial for both algorithms (*Figure 12*), that the intrinsic time scales (1/eigenvalues) of the process to be analyzed are slower than the integration time of the data points. Otherwise the accuracy of the inference deteriorates.

Altogether, we demonstrated the performance of the generalized Kalman filter on ion channel data for inference of kinetic schemes. Nevertheless, our approach can approximate any other stochastic system and signal distributions of linear (pseudo-first-order) kinetics (*Sorenson and Alspach, 1971*). Prospective extensions of the Bayesian filter, for example by Bayesian Gaussian sum filters or similar numerically brute force concepts such as particle filters (*Golightly and Wilkinson, 2011*; *Gillespie and Golightly, 2012*), can overcome modeling errors at low ion channel numbers or low photon fluxes.

## Materials and methods

We simulated state evolution $s(t)$ with either the software QuB (*Nicolai and Sachs, 2014*) for PC data or an inhouse Matlab routine (The code will be shared on request.) for cPCF data. The inhouse Matlab routine is an implementation of the Gillespie algorithm Gillespie Daniel T. (1977). Traces were summed up, defining the ensemble state vector $\mathbf{n}(t) := (n_1, n_2, n_3, n_4)^\top$, which counts the number of channels in each state. At first we used a 10 kHz sampling frequency for the Gillespie algorithm but for investigating the errors induced by analog filtering the current signal and the finite integration time for each fluorescence data point the Gillespie algorithm sampled at 100 kHz. The KF, RE, and Bayesian filter routines were implemented in Stan (*Carpenter et al., 2017*) with the interface package PyStan and ran on a high performance computing cluster with O(100) Broadwell and SkyLake nodes. A Tutorial for Patch clamp data can be found on the git hub page https://github.com/JanMuench/Tutorial_Patch-clamp_data and for cPCF data, https://github.com/JanMuench/Tutorial_Bayesian_Filter_cPCF_data.

The cPCF data simulation code can be found here: https://cloudhsm.it-dlz.de/s/QB2pQQ7ycMXEitE (**Source code 1**).

## Methods

Hereinafter, we derive the equations for our Bayesian filter for time series analysis of hidden linear chemical reaction networks (kinetic schemes). A detailed description of the experimental noise is provided in the Appendix 5.

### The relation of Bayesian statistics to the Kalman filter

The following conventions are generally used: Bold symbols are used for multi-dimensional objects such as vectors or matrices. Calligraphic letters are used for (some) vectorial time series and double-strike letters are used for probabilities and probability densities. Within the Bayesian paradigm (**Hines, 2015**; **Ball, 2016**), each unknown quantity, including model parameters $\boldsymbol{\theta}$ and time series of occupancies of hidden states $\mathfrak{N}_T = \{\mathbf{n}(t_i)\}_{i=1}^T$, are treated as random variables conditioned on observed time series data $\mathcal{Y}_T = \mathbf{y}(t_i)_{i=1}^T$. The prior $\mathbb{P}(\boldsymbol{\theta}) = \prod_j^{N_{\text{par}}} \mathbb{P}(\theta_j)$ or posterior distribution $\mathbb{P}(\boldsymbol{\theta}|\mathcal{Y}_T)$ encodes the available information about the parameter values before and after analysing the data, respectively. According to the Bayesian theorem, the posterior distribution

$$\mathbb{P}(\boldsymbol{\theta}|\mathcal{Y}_T) = \frac{1}{Z(\mathcal{Y}_T)} \mathbb{L}(\mathcal{Y}_T|\boldsymbol{\theta}) \prod_j^{N_{\text{par}}} \mathbb{P}(\theta_j) \tag{20}$$

is a probability distribution of a parameter set $\boldsymbol{\theta}$ conditioned on $\mathcal{Y}_T$. The likelihood $\mathbb{L}(\mathcal{Y}_T|\boldsymbol{\theta})$ encodes the distribution of the data by modelling the intrinsic fluctuations of the protein as well as noise coming from the experimental devices. The prior provides either assumptions before measuring data or what has been learnt from previous experiments about $\boldsymbol{\theta}$. The normalization constant

$$Z(\mathcal{Y}_T) = \int \mathbb{L}(\mathcal{Y}_T|\boldsymbol{\theta}) \mathbb{P}(\boldsymbol{\theta}) d\boldsymbol{\theta} \tag{21}$$

ensures that the posterior is a normalized distribution. The KF is a special class of models in the family of Bayesian filters (**Ghahramani, 1997**), which is a generalisation of the classical KF. Due to its linear time evolution (**Equation 1**), the KF is particularly useful for modeling time series data of ensembles dynamics of first order chemical networks. It delivers a set of recursive algebraic equations (Materials and methods **Equation 28** and **Equation 32**) for each time point, which allows to express the prior $\mathbb{P}(\mathbf{n}(t)|\mathcal{Y}_{t-1})$ and (after incorporating $\mathbf{y}(t)$) the posterior $\mathbb{P}(\mathbf{n}(t)|\mathcal{Y}_t)$ occupancies of hidden states $\mathbf{n}(t)$ for all $t$ given a set of parameters $\boldsymbol{\theta}$. This means the KF solves the filtering problem (inference of $\mathfrak{N}_T$) by explicitly modeling the time evolution of $\mathbf{n}(t)$ by multivariate normal distributions. This allows us to replace $\mathbb{L}(\mathcal{Y}_T|\boldsymbol{\theta})$ of **Equation 20** by the expression of **Equation 9**.

The Bayesian framework (as demonstrated in this article) has various properties which makes it superior to ML estimation (MLE) (**McElreath, 2018**). Those properties are in particular useful for the analysis of biophysical data since very often the dynamics of interest are hidden or latent in the data. Models with a hidden structure are called singular. For regular (non-singular) statistical models, maxima $\boldsymbol{\theta}_{\text{ML}}$ of the posterior or likelihood converge in distribution

$$\lim_{n\to\infty} \sqrt{n}(\boldsymbol{\theta}_{\text{ML}} - \boldsymbol{\theta}_{\text{true}}) \sim \mathcal{N}(0, \mathbf{F}^{-1}(\boldsymbol{\theta}_{\text{true}})) \tag{22}$$

to the true value $\boldsymbol{\theta}_{\text{true}}$, where $\mathbf{F}^{-1}(\boldsymbol{\theta}_{\text{true}})$ is the inverse Fisher information matrix. Under those conditions it is justified to derive from the curvature of the likelihood at $\boldsymbol{\theta}_{\text{ML}}$ via the Cramer-Rao-bound theorem

$$\text{covar}[\boldsymbol{\theta}_{\text{ML}}] = \mathbf{F}^{-1}(\boldsymbol{\theta}_{\text{ML}}) \tag{23}$$

a confidence volume for the inferred parameters. In contrast, consider for example the type of data investigated in this study which probes the protein dynamics by current and light. Singularity means that the Fisher information matrix of a model is not invertible leading to the breakdown of the Cramer-Rao Bound theorem. Due to the breakdown, it cannot be guaranteed that even in the asymptotic limit the log-likelihood function can be approximated by a quadratic form *Watanabe, 2007*. Thus, usually the MLE does not obey **Equation 22**. Consequently, the posterior distribution is usually not

a normal distribution either (*Watanabe, 2007*). Using the full posterior distribution without further approximations detects the resulting problems such as deviation from normality or non-identifiability of parameters, related to the singularity. In conclusion, the posterior is still a valid representation of parameter plausibility while ML fails.

## Time evolution of a Markov Model for a single channel

In the following, we write the time $t$ as function argument rather than a subscript. Following standard approaches, we attribute to each state of the Markov model an element of a vector space with dimension $M$. At a time, a channel can only be in a single state. This implies that the set of possible states is S:=$\{(1, 0, 0, \ldots), (0, 1, 0, \ldots), \ldots, (\ldots, 0, 1)\} \subset \{0, 1\}^M$. In the following, Greek subscripts refer to different states while Latin subscripts refer to different channels. By $\mathbf{s}(t) = \mathbf{e}_\alpha$ we specify that the channel is in state $\alpha$ at time $t$. Mathematically, $\mathbf{e}_\alpha$ stands for the $\alpha$-th canonical unit Cartesian vector (*Table 1*).

Assuming that the state transitions can be modeled by a first order Markov process, the path probability can be decomposed as the product of conditional probabilities as follows:

$$\mathbb{P}(\text{path}) = \mathbb{P}(\mathbf{s}(0), \mathbf{s}(1), \ldots, \mathbf{s}(T)) = \mathbb{P}(\mathbf{s}(0)) \cdot \mathbb{P}(\mathbf{s}(1) \mid \mathbf{s}(0)) \cdot \mathbb{P}(\mathbf{s}(2) \mid \mathbf{s}(1)) \cdots \mathbb{P}(\mathbf{s}(T) \mid \mathbf{s}(T-1)). \quad (24)$$

Markov models (MMs) and rate models are widely used for modeling molecular kinetics (Appendix 2). They provide an interpretation of the data in terms of a set of conformational states and the transition rates between these states. For exactness it remains indispensable to model the dynamics with a HMMs (*Noé et al., 2013*). The core of a hidden Markov model is a conventional Markov model, which is supplemented with a an additional observation model. We will therefore first focus on a conventional Markov model. State-to-state transitions can be equivalently described with a transition matrix $\mathbf{T}$ in discrete time or with a rate matrix $\mathbf{K}$ in continuous time, as follows:

$$\mathbf{T}_{\alpha,\beta} := \mathbb{P}(\mathbf{s}(t+1) = \mathbf{e}_\alpha \mid \mathbf{s}(t) = \mathbf{e}_\beta) = \exp(\mathbf{K} \cdot \Delta t)_{\alpha,\beta}, \quad (25)$$

where exp is the matrix exponential. We aim to infer the elements of the rate matrix $\mathbf{K}$, constituting a kinetic model or reaction network of the channel. Realizations of sequences of states can be produced by the Doob-Gillespie algorithm Gillespie Daniel T. (1977). To derive succinct equations for the stochastic dynamics of a system, it is beneficial to consider the time propagation of an ensemble of virtual system copies. This allows to ascribe a probability vector $\mathbf{p}(t)$ to the system, in which each element $p_\alpha(t)$ is the probability to find the system at $t$ in state $\alpha$. One can interpret the probability vector $\mathbf{p}$ as the instantaneous expectation value of the state vector $\mathbf{s}$.

$$\mathbf{p}(t) = \mathbb{E}[\mathbf{s}(t)] \quad (26)$$

The probability vector obeys the discrete-time Master equation

$$\mathbf{p}(t+1) = \mathbf{T}\mathbf{p}(t) \quad (27\text{a})$$

$$\mathbb{E}[\mathbf{s}(t+1)] = \mathbf{T}\mathbb{E}[\mathbf{s}(t)] \quad (27\text{b})$$

## Time evolution of an ensemble of identical non-interacting channels

We model the experimentally observed system as a collection of non-interacting channels. A single channel can be modeled with a first-order MM. The same applies to the ensemble of non-interacting channels. We focus on modeling the time course of extensive macroscopic observables such as the mean current and fluorescence signals as well as their fluctuations. A central quantity is the vector $\mathbf{n}(t)$ which is the occupancy of the channel states at time $t$:

$$\mathbf{n}(t) = \sum_{i=1}^{N_{\text{ch}}} \mathbf{s}_i(t) \quad (28)$$

This quantity, like $\mathbf{s}(t)$, is a random variate. Unlike $\mathbf{s}(t)$, its domain is not confined to canonical unit vectors but to $\mathbf{n} \in \mathbb{N}^M$. From the linearity of *Equation 28* in the channel dimension and from the single-channel CME *Equation 27b* one can immediately derive the equation for the time evolution of the mean occupancy $\bar{\mathbf{n}}(t) = \mathbb{E}[\mathbf{n}(t)]$:

$$\bar{n}_\alpha(t+1) = \sum_\beta T_{\alpha,\beta} \bar{n}_\beta(t) \quad (29)$$

**Table 1.** Important symbols.

| Symbol | Meaning |
| --- | --- |
| $\boldsymbol{\theta}$ | Set of all unknown model parameters for which the posterior distribution is sampled |
| $\mathbf{n}(t)$ | Hidden ensemble occupancy vector of channel states in a specific patch at time $t$ which is a continuous Markov state vector $\mathbf{n}(t) \in \mathbb{R}^M$ |
| $\mathbf{P}(t)$ | Variance-covariance matrix of a hidden ensemble state $\mathbf{n}(t)$ n a specific patch at time $t$ which contains the dispersion of the ensemble and the lacking knowledge of the algorithm about the true $\mathbf{n}(t)$ |
| $\mathbf{T}$ | Transition matrix of a single channel |
| $\mathbf{K}$ | Rate matrix which is the logarithm of the transition matrix |
| $\mathbf{H}$ | Observation matrix which projects the hidden ensemble state vector onto its mean signal. |
| $\mathbf{s}$ | Single-molecule Markov state vector |
| $k_{i,j}$ | Specific transition rate from state $j$ to state $i$, $[\mathbf{K}]_{i,j} = k_{i,j}$ , |
| $K_i$ | Ratio of two transition rates i.e. an equilibrium constant |
| $\mathbf{y}(t)$ | Data point at time |
| $T$ | Number of observations in a time series |
| $\mathcal{Y}_T$ | Time series of $T$ data points, $\mathcal{Y}_T = \mathbf{y}(t_i)_{i=1}^{T}$ |
| $\mathfrak{N}_T$ | Time series of $T$ hidden ensemble states, $\mathfrak{N}_T = \{\mathbf{n}(t_i)\}_{i=1}^{T}$ |
| $N_{\mathrm{ch},j}$ | Number of channels in patch number |
| $i$ | Mean electrical current through a single-channel |
| $\sigma_{\mathrm{m}}^2$ and $\sigma_{\mathrm{ex}}^2$ | Variance of the current including all noise from the patch and the recording system |
| $\sigma_{\mathrm{op}}^2$ | Variance of the current noise generated by a single open-channel |
| $\lambda_{\mathrm{b}}$ | Mean brightness of a bound ligand |
| $\lambda_{\mathrm{Fl}}$ | Mean brightness of the fluorescence signal from bulk and bound ligands |
| $\sigma_{\mathrm{back}}^2$ | Variance of the fluorescence generated by unbound ligands after subtraction of the image obtained for the reference dye |
| $M$ | Number of single-channel states which is the dimension of $\mathbf{n}(t) \in \mathbb{N}^M$ in the KF algorithm |
| $N_{\mathrm{obs}}$ | Dimensions of the observational space |
| $\mathbb{F}(\mathcal{Y})$ | True probability density of $\mathcal{Y}$, i.e. the true data-generating process |
| $\mathbb{L}(\mathcal{Y}|\boldsymbol{\theta})$ | Likelihood function of the model parameters |
| $\mathbb{P}(\boldsymbol{\theta}|\mathcal{Y})$ | Posterior distribution of the model parameters |
| $\mathbb{P}_{\mathrm{pred}}(\widetilde{\mathcal{Y}}|\mathcal{Y})$ | Predictive distribution of the new data points |
| $\mathbb{O}(\mathbf{y}|\mathbf{n})$ | Distribution of observables for a single time step |
| $\mathcal{N}(\cdot|\boldsymbol{\mu}, \boldsymbol{\Sigma})$ | Normal distribution with mean μ and variance-covariance matrix $\sum$ |
| $\mathbb{E}[\cdot]$ | Mean value |

with the transition matrix $\mathbf{T}$. The full distribution $\mathbb{P}(\mathbf{n}(t + 1)|\mathbf{n}(t))$ is a generalized multinomial distribution. To understand the generalized multinomial distribution and how it can be constructed from the (conventional) multinomial distribution, consider the simplified case where all channels are assumed to be in the same state α. Already after one time step, the channels will have spread out over the state space. The channel distribution after one time step is parametrized by the transition probabilities in

row number α of the single-channel transition matrix $\mathbf{T}$. According to the theory of Markov models, the final distribution of channels originating from state α is the multinomial distribution

$$\mathbb{P}(\mathbf{n}^{(\alpha)}(t+1) \mid n_\alpha \mathbf{e}_\alpha) = \mathbb{P}(n_1, \ldots, n_M \mid \mathbf{n}(t) = n_\alpha \mathbf{e}_\alpha) = \frac{n_\alpha!}{n_1! \cdots n_M!} T_{1,\alpha}^{n_1} \cdots T_{M,\alpha}^{n_M} \tag{30}$$

In general, the initial ensemble will not have only one but multiple occupied channel states. Because of the independence of the channels, one can imagine each initial sub-population spreading out over the state space independently. Each sub-population with initial state α gives rise to its own final multinomial distribution that contributes $n_\beta^{(\alpha)}$ transitions into state β to the total final distribution. The total number of channels at $t+1$ in each state can then be simply found by adding the number of channels transitioning out of the different states α.

$$\mathbf{n}(t+1) = \sum_\alpha \mathbf{n}^{(\alpha)}(t+1) \tag{31}$$

Evidently, the total number of channels is conserved during propagation. The distribution of $\mathbf{n}(t+1)$, defined by *Equations 30; 31*, is called the *generalized multinomial distribution*:

$$\mathbf{n}(t+1) \sim \text{general-multinomial}(\mathbf{n}(t), \mathbf{T}) \tag{32}$$

While no simple expression exists for the generalized multinomial distribution, closed form expressions for its moments can be readily derived. For large $N_{\text{ch}}$ each $\mathbb{P}(\mathbf{n}^{(\alpha)}(t+1) \mid n_\alpha \mathbf{e}_\alpha)$ can be approximated by a multivariate-normal distribution such that also general-multinomial$(\mathbf{n}(t), \mathbf{T})$ has a multivariate-normal approximation. In the next section, we combine the kinetics of channel ensembles with the KF by a moment expansion of the governing equations for the ensemble probability evolution.

## Moment expansion of ensemble probability evolution

The multinomial distribution (*Fredkin and Rice, 1992*) has the following mean and covariance matrix

$$\overline{\mathbf{n}}^{(\alpha)}(t+1) = n_\alpha \mathbf{T}_{:,\alpha} \tag{33}$$

$$\mathbf{\Sigma}^{(\alpha)}(t+1) = n_\alpha \text{diag}(\mathbf{T}_{:,\alpha}) - n_\alpha \mathbf{T}_{:,\alpha,:} \mathbf{T}_{:,\alpha}^\top \tag{34}$$

where $\mathbf{T}_{:,\alpha}$ denotes the column number α of the transition matrix and diag$(\mathbf{T}_{:,\alpha})$ describes the diagonal matrix with $\mathbf{T}_{:,\alpha}$ on its diagonal. Combining *Equation 31* with *Equations 33; 34* we deduce the mean and variance of the generalized multinomial distribution:

$$\mathbb{E}\left[\mathbf{n}(t+1) \mid \mathbf{n}(t)\right] = \sum_\alpha n_\alpha(t) \mathbf{T}_{:,\alpha} = \mathbf{T}\mathbf{n}(t) \tag{35}$$

$$\text{cov}\left[\mathbf{n}(t+1), \mathbf{n}(t+1) \mid \mathbf{n}(t)\right] = \sum_\alpha n_\alpha(t)\left(\text{diag}(\mathbf{T}_{:,\alpha}) - \mathbf{T}_{:,\alpha}\mathbf{T}_{:,\alpha}^\top\right) = \text{diag}\left(\mathbf{T}\mathbf{n}(t)\right) - \mathbf{T}\text{diag}\left(\mathbf{n}(t)\right)\mathbf{T}^\top \tag{36}$$

Note that *Equations 35; 36* are conditional expectations that depend on the random state $\mathbf{n}$ at the previous time $t$ and not only on the previous mean $\overline{\mathbf{n}}$. To find the absolute mean, the law of total expectation is applied to *Equation 35*, giving

$$\overline{\mathbf{n}}(t+1) = \mathbb{E}\left[\mathbb{E}\left[\mathbf{n}(t+1) \mid \mathbf{n}(t)\right]\right] = \mathbf{T}\overline{\mathbf{n}}(t), \tag{37}$$

in agreement with the simple derivation of *Equation 29*. We introduce a shorthand $\mathbf{P}(t) := \text{cov}(\mathbf{n}(t), \mathbf{n}(t))$ for the absolute covariance matrix of $\mathbf{n}(t+1)$. Similarly, $\mathbf{P}(t)$ can be found by applying the law of total variance decomposition (*Weiss, 2005* to *Equations 35; 36*), giving

$$\mathbf{P}(t+1) = \mathbb{E}\left[\text{cov}\left(\mathbf{n}(t+1), \mathbf{n}(t+1) \mid \mathbf{n}(t)\right)\right] + \text{cov}\left[\mathbb{E}(\mathbf{n}(t+1) \mid \mathbf{n}(t)), \mathbb{E}(\mathbf{n}(t+1) \mid \mathbf{n}(t))\right] \tag{38a}$$

$$= \text{diag}\left(\mathbf{T}\overline{\mathbf{n}}(t)\right) - \mathbf{T}\text{diag}\left(\overline{\mathbf{n}}(t)\right)\mathbf{T}^\top + \text{cov}(\mathbf{T}\mathbf{n}(t), \mathbf{T}\mathbf{n}(t)) \tag{38b}$$

$$= \text{diag}\left(\mathbf{T}\overline{\mathbf{n}}(t)\right) - \mathbf{T}\text{diag}\left(\overline{\mathbf{n}}(t)\right)\mathbf{T}^\top + \mathbf{T}\text{cov}(\mathbf{n}(t), \mathbf{n}(t))\mathbf{T}^\top \tag{38c}$$

$$= \text{diag}\left(\mathbf{T}\overline{\mathbf{n}}(t)\right) - \mathbf{T}\text{diag}\left(\overline{\mathbf{n}}(t)\right)\mathbf{T}^\top + \mathbf{T}\mathbf{P}(t)\mathbf{T}^\top \tag{38d}$$

*Equations 37, 38d* dare compact analytical expressions for the mean and the covariance matrix of the occupancy vector $\mathbf{n}$ at $t+1$ that depend on the mean $\overline{\mathbf{n}}$ and covariance matrix $\mathbf{P}$ at the previous

time step $t$. Chaining these equations for different time steps $t = 0, \ldots, T$ allows to model the whole evolution of a channel ensemble. Moreover, these two equations together with the output statistics of $\mathbb{O}(\mathbf{y}|\mathbf{n}(t))$ are sufficient to formulate correction equations *Equation 59* of the KF (*Moffatt, 2007*; *Anderson and Moore, 2012*). These equations will be used in a Bayesian context to sample the posterior distribution of the model parameters. The sampling entails repeated numerical evaluation of the model likelihood. Therefore, analytical equations for the ensemble evolution that can be quickly evaluated on a computer millions of times are indispensable. This was achieved by deriving *Equation 37*, *Equation 38d*. Comparing *Equation 38d* with the KF prediction equation (*Anderson and Moore, 2012*) for $\mathbf{P}(t)$, we obtain the state-dependent covariance matrix of *Equation 3* as

$$\mathbf{Q}(\mathbf{T}, \overline{\mathbf{n}}(t)) = \text{diag}\left(\mathbf{T}\overline{\mathbf{n}}(t)\right) - \mathbf{T}\text{diag}\left(\overline{\mathbf{n}}(t)\right)\mathbf{T}^{\mathsf{T}} \tag{39}$$

In the following section on properties of measured data and the KF, we no longer need to refer to the random variate $\mathbf{n}(t)$. All subsequent equations can be formulated by only using the mean hidden state $\overline{\mathbf{n}}(t)$ and the variance-covariance matrix of the hidden state $\mathbf{P}(t)$. We therefore drop the overbar in $\overline{\mathbf{n}}(t)$ so that the symbol $\mathbf{n}(t)$ refers from now on to the mean hidden state.

## Modeling simultaneous measurement of current and fluorescence

In the following, we develop a model for the conditional observation distribution $\mathbb{O}(\mathbf{y}|\mathbf{n}(t))$ (Appendix 5 for experimental details). Together with the hidden ensemble dynamics this will enable us to derive the output statistics of the KF (see, below). Let $\mathbf{y}(t)$ be the vector of all observations at $t$. Components of the vector are the ion current and fluorescence intensity.

$$\mathbf{y}(t) = \begin{pmatrix} \text{fluorescence intensity}(t) \\ \text{ion current}(t) \end{pmatrix} = \begin{pmatrix} y_{\text{flu}}(t) \\ y_{\text{curr}}(t) \end{pmatrix} \tag{40}$$

As outlined in the introduction part, in *Equation 4* we model the observation by using a conditional probability distribution $\mathbb{O}(\mathbf{y}(t)|\mathbf{n}(t))$ that only depends on the mean hidden state $\mathbf{n}(t)$, as well as on fixed channel and other measurement parameters. $\mathbb{O}(\mathbf{y}(t)|\mathbf{n}(t))$ is modeled as a multivariate normal distribution with mean $\mathbf{H}\mathbf{n}(t)$ and variance-covariance matrix $\boldsymbol{\Sigma}(t)$, that can in general depend on the mean state vector $\mathbf{n}(t)$ (much like the covariance matrix of the kinetics in (*Equation 38d*) ). The observation matrix $\mathbf{H} \in \mathbb{R}^{N_{\text{obs}} \times M}$ projects the hidden state vector $\mathbf{n}(t)$ onto $\mathbf{H}\mathbf{n}(t) \in \mathbb{R}^{N_{\text{obs}}}$, the observation space. The observation distribution is

$$\mathbb{O}(\mathbf{y}(t)|\mathbf{n}(t)) = \mathcal{N}\left(\mathbf{y}(t)|\mathbf{H}\mathbf{n}(t), \boldsymbol{\Sigma}(\mathbf{n}(t))\right) \Leftrightarrow \mathbf{y}(t) = \mathbf{H}\mathbf{n}(t) + \boldsymbol{\nu}(t). \tag{41}$$

This measurement model is very flexible and allows to include different types of signals and error sources arising from both the molecules and the instruments. A summary of the signals and sources

---

**Table 2.** Summary of signals and noise sources for the exemplary CCCO model with the closed states $\alpha = 1, 2, 3$ and the open state $\alpha = 4$.

The observed space is two-dimensional with $y_{Fl}$ = fluorescence and $y_I$ = ion current. The fluorescence signal is assumed to be derived from the difference of two spectrally different Poisson distributed fluorescent signals. That procedure results in a scaled Skellam distribution of the noise.

| | ion current | | fluorescence | |
| --- | --- | --- | --- | --- |
| | current signal | measurement noise | fluorescence signal | background fluorescence |
| Signaling states | Open state | - | Ligand-bound states | - |
| Error term | Open-channel noise | Measurement noise | Photon counts | Bulk noise |
| Affected signal | Current | Current | Fluorescence | Fluorescence |
| Distribution | Normal $(in_4, \sigma_{\text{op}}^2 n_4)$ | Normal $(0, \sigma_{\text{m}}^2)$ | Poisson $(\lambda_b n_i(t))$ | Scaled Skellam |
| Contribution to $\mathbf{H}$ | $H_{2,4} = i$ | - | $\mathbf{H}_{1,:} = (0, \lambda_b, 2\lambda_b, 2\lambda_b)$ | - |
| Contribution to $\boldsymbol{\Sigma}$ | $\Sigma_{2,2} = \sigma_{\text{op}}^2 n_4(t)$ | $\Sigma_{2,2} = \sigma_{\text{m}}^2$ | $\Sigma_{1,1} = (0, \lambda_b, 2\lambda_b, 2\lambda_b)\mathbf{n}(t)$ | $\Sigma_{1,1} = \sigma_{\text{back}}^2$ |

---

of measurement error and their contributions to the parameters of $\mathbb{O}(\mathbf{y}(t)|\mathbf{n}(t))$ is provided by **Table 2**. Below we address the two types of signals and four noise sources one by one. For this, we decompose the observation matrix and the observation noise covariance matrix into the individual terms:

$$\mathbf{H} = \mathbf{H}_{\mathrm{I}} + \mathbf{H}_{\mathrm{binding}} \tag{42}$$

$$\mathbf{\Sigma}(t) = \mathbf{\Sigma}_{\mathrm{open}}(t) + \mathbf{\Sigma}_{\mathrm{meas.}} + \mathbf{\Sigma}_{\mathrm{binding}}(t) + \mathbf{\Sigma}_{\mathrm{back}} \tag{43}$$

In the following, we report the individual matrices for the exemplary CCCO model with one open state $\alpha = 4$ and three closed states $\alpha = 1, 2, 3$. Matrices can be constructed analogously for the other models. For the definition of $\mathbf{\Sigma}_{\mathrm{back}}$ refer to (Appendix 5).

## Macroscopic current and open-channel noise

We model the current and the intrinsic fluctuations of the open-channel state $\mathbf{s} = \mathbf{e}_4$ (the *open channel noise*) by a state-dependent normal distribution with mean $in_4(t)$ where $n_4(t)$ is the number of channels in the open state at $t$ and   is the single-channel current. The additional variance of the single-channel current is described by $\sigma_{\mathrm{open}}^2$. The sum of the instrumental noise of the experimental setup and the *open channel noise* is modeled as uncorrelated (white) normally distributed noise with the mean $\mathbb{E}[\nu_{\mathrm{I}}(t)] = 0$ and variance $\mathbb{E}[\nu_{\mathrm{I}}^2(t)] = \sigma_{\mathrm{op}}^2 n_4(t) + \sigma_{\mathrm{m}}^2$. By making the open-channel noise dependent on the hidden state population $n_4(t)$, we fully take advantage of the flexibility of Bayesian filters which admits an (explicitly or implicitly) time-dependent observation model. By tabulating the parameters of the two normal distributions into $\mathbf{H}$ and $\mathbf{\Sigma}$, we obtain

$$\mathrm{H}_{\mathrm{I}} := \begin{pmatrix} 0 & 0 & 0 & 0 \\ 0 & 0 & 0 & i \end{pmatrix} \tag{44}$$

$$\mathbf{\Sigma}_{\mathrm{open}}(t) + \mathbf{\Sigma}_{\mathrm{meas.}} := \begin{pmatrix} 0 & 0 \\ 0 & \sigma_{\mathrm{op}}^2 n_4(t) + \sigma_{\mathrm{m}}^2 \end{pmatrix} \tag{45}$$

One can now ask for the variance of a data point $y(t)$ given the epistemic and aleatory uncertainty of $\mathbf{n}(t)$ encoded by $\mathbf{P}(t)$ in **Equation 38d**. By using the law of total variance the signal variance follows as:

$$\mathrm{var}(\mathbf{y}(t)) = \mathbb{E}[\mathrm{var}[\mathbf{y}(t)|\mathbf{n}(t)]] + \mathrm{var}[\mathbb{E}[\mathbf{y}(t)|\mathbf{n}(t)]] \tag{46a}$$

$$= \mathbb{E}[\sigma_{\mathrm{op}}^2 n_4(t) + \sigma_{\mathrm{m}}^2] + \mathrm{var}[\mathbf{H}_{\mathrm{I}}\mathbf{n}(t)] \tag{46b}$$

$$= \sigma_{\mathrm{op}}^2 \mathbb{E}[n_4(t)] + \sigma_{\mathrm{m}}^2 + (\mathbf{H}_{\mathrm{I}}\mathbf{P}(t)\mathbf{H}_{\mathrm{I}}^{\top})_{2,2} \tag{46c}$$

See, Appendix 6 for further details.

## Fluorescence and photon-counting noise

The statistics of photon counts in the fluorescence signal are described by a Poisson distribution with emission rate $\lambda_{\mathrm{Fl}}$

$$y_{\mathrm{Fl}}(t) \sim \mathrm{Pois}(\lambda_{\mathrm{Fl}}(t)). \tag{47}$$

The total emission rate $\lambda_{\mathrm{Fl}}$ can be modeled as a weighted sum of the specific emission rates $\lambda_b$ of each ligand class $\{0, 1, 2\}$. The weights are given by the stoichiometric factors which reflect the number of bound ligands. In order to cast the Poisson distribution into the functional form of the observation model (**Equation 41**), we invoke the central limit theorem to approximate

$$y_{\mathrm{Fl}} \sim \mathrm{Pois}(\lambda_{\mathrm{Fl}}) \approx \mathcal{N}(\lambda_{\mathrm{Fl}}(t), \lambda_{\mathrm{Fl}}(t)) \tag{48}$$

The larger $\lambda_{\mathrm{Fl}}$ the better is the approximation. We assume, that the confocal volume is equally illuminated. For our model of ligand fluorescence, we assume for a moment that there is no signal coming from ligands in the bulk. We will drop this assumption in the next section. With these assumptions, we arrive at the following observation matrix

$$H_{\text{binding}} := \begin{pmatrix} 0 & \lambda_b & 2\lambda_b & 2\lambda_b \\ 0 & 0 & 0 & 0 \end{pmatrix} \tag{49}$$

The matrix $\mathbf{H}$ aggregates the states into two conductivity classes: non-conducting and conducting and three different fluorescence classes. The first element $(\mathbf{Hn})_1$ is the mean fluorescence $\lambda_{\text{Fl}}(t) = \lambda_b[n_2(t) + 2(n_3(t) + n_4(t))]$. The variance-covariance matrix $\Sigma_{\text{binding}}$ can be derived along the same lines using *Equation 48*. We find

$$\Sigma_{\text{binding}}(t) := \begin{pmatrix} (\mathbf{Hn}(t))_1 & 0 \\ 0 & 0 \end{pmatrix} \tag{50}$$

Under these assumptions, the observation matrix can be written as follows

$$H := \begin{pmatrix} 0 & \lambda_b & 2\lambda_b & 2\lambda_b \\ 0 & 0 & 0 & i \end{pmatrix} \tag{51}$$

## Output statistics of a Kalman Filter

with two-dimensional state-dependent noiseNow simultaneously measured current and fluorescence data $\mathbf{y} \in \mathbb{R}^2$, obtained by cPCF, are modeled. Thus, the observation matrix fulfills $\mathbf{H} \in \mathbb{R}^{2 \times M}$. One can formulate the observation distribution as

$$y(t) = Hn(t) + \nu_m(t) + \begin{pmatrix} \nu_{\text{pois}}(t) \\ \nu_{\text{op}}(t) \end{pmatrix} \Leftrightarrow \mathbf{y} \sim \mathcal{N}(\mathbf{Hn}(t), \Sigma(t)). \tag{52}$$

The vector $\nu_m$ denotes the experimental noise, with $\mathbb{E}[\nu_m] = 0$ and variance given by the diagonal matrix $\Sigma_{\text{meas}} + \Sigma_{\text{back}}$. The second noise term arises from Poisson-distributed photon counting statistics and the open-channel noise. It has the properties

$$\mathbb{E}\left[\begin{pmatrix} \nu_{\text{pois}}(t) \\ \nu_{\text{op}}(t) \end{pmatrix}\right] = 0 \tag{53}$$

and

$$\text{cov}\left(\begin{pmatrix} \nu_{\text{pois}}(t) \\ \nu_{\text{op}}(t) \end{pmatrix}, \begin{pmatrix} \nu_{\text{pois}}(t) \\ \nu_{\text{op}}(t) \end{pmatrix}\right) = \Sigma_{\text{open}}(t) + \Sigma_{\text{binding}}(t). \tag{54}$$

The matrix $\Sigma$ is a diagonal matrix. To derive the covariance matrix $\text{cov}(\mathbf{y}(t))$ we need to additionally calculate $\text{var}(y_{\text{fluo}}(t))$ and $\text{cov}(y_{\text{fluo}}(t), y_{\text{patch}}(t))$. By the same arguments as above we get

$$\text{var}[y_{\text{fluo}}(t)] = \mathbb{E}[\text{var}(y(t)|\mathbf{n}(t))] + \text{var}[\mathbb{E}(y(t)|\mathbf{n}(t)] \tag{55a}$$

$$= \mathbb{E}\left[\sigma_{\text{back}}^2 + (\mathbf{Hn}(t))_1\right] + \text{var}(\mathbf{Hn}(t)) \tag{55b}$$

$$= \sigma_{\text{back}}^2 + (\mathbf{Hn}(t))_1 + (\mathbf{Hn}(t))\mathbf{H}^{\mathsf{T}})_{1,1} \tag{55c}$$

The cross terms can be calculated by using the law of total covariance

$$\text{cov}(y_{\text{patch}}, y_{\text{fluo}}) = \mathbb{E}[\text{cov}(y_{\text{patch}}, y_{\text{fluo}}|\mathbf{n})] + \text{cov}(\mathbb{E}(y_{\text{patch}}|\mathbf{n}), \mathbb{E}(y_{\text{fluo}}|\mathbf{n})) \tag{56a}$$

$$= 0 + \text{cov}(\mathbf{H}_{2,:}\mathbf{n}, \mathbf{H}_{1,:}\mathbf{n}) \tag{56b}$$

$$= \mathbf{H}_{2,:}\text{cov}(\mathbf{n}, \mathbf{n})\mathbf{H}_{1,:}^{\mathsf{T}} = \mathbf{H}_{2,:}\mathbf{P}(t)\mathbf{H}_{1,:}^{\mathsf{T}} \tag{56c}$$

yielding the matrix

$$\text{cov}(\mathbf{y}, \mathbf{y}) = \mathbf{HP}(t)\mathbf{H}^{\mathsf{T}} + \Sigma(t) \tag{57}$$

We assumed that the Poisson distribution is well captured by the normal approximation. In cPCF data, the ligand binding to only a sub-ensemble of the channels is monitored, which we assume to represent the conducting ensemble such that $N_{ch,FL} = N_{ch,I}$. For real data, further refinement might be necessary to model the randomness of the sub-ensemble in the summed voxels. With the time evolution equations for the mean (*Equation 35*) and for the covariance matrix *Equation 38d* as well as with the expressions for the signal variance, we possess all parameters that are needed in the correction equation of the (*Kalman, 1960*; *Anderson and Moore, 2012*).

## The correction step

For completeness we write down the correction step (Bayesian update) of the KF, although its derivation can be found in *Chen, 2003*; *Anderson and Moore, 2012*; *Moffatt, 2007*. The mean ensemble state $\mathbf{n}(t)$ is corrected by the current data point

$$\mathbf{n}_{\text{posterior}}(t) = +\mathbf{n}_{\text{prior}}(t) + \mathbf{K}_{\text{Kal}}\left(\mathbf{y}(t) - \mathbf{H}\mathbf{n}_{\text{prior}}(t)\right) \tag{58}$$

where Kalman gain matrix $\mathbf{K}_{\text{Kal}} := \mathbf{P}(t)_{\text{prior}}\mathbf{H}^{\top}\mathbf{\Sigma}^{-1}$ evaluates the intrinsic noise against the experimental noise. How precise are my model predictions about $\mathbf{n}(t)$ compared with the information gained about $\mathbf{n}(t)$ by measuring $\mathbf{y}(t)$. The covariance $\mathbf{P}(t)$ of the ensemble state $\mathbf{n}(t)$ is corrected by

$$\mathbf{P}_{\text{posterior}}(t) = \mathbf{P}_{\text{prior}}(t) - \mathbf{K}_{\text{Kal}}\left(\mathbf{H}\mathbf{P}_{\text{prior}}(t)\mathbf{H} + \mathbf{\Sigma}(t)\right)\mathbf{K}^{\top} \tag{59}$$

*Equation 58,59, 37 and 38d* form the filtering equations which summarize the algorithm. One initialises the first $\mathbf{n}(0)$ and $\mathbf{P}(0)$ and with an equilibrium assumption.

## Microscopic-reversibility as a hierarchical prior

We applied microscopic-reversibility (*Colquhoun et al., 2004*) by a hierarchical prior distribution. Usually, micro-reversibility is strictly enforced by setting the product of the rates of the clockwise loop $k_1, k_2, k_3 k_4$ equal to the anti-clockwise loop $k_5, k_6, k_7, k_8$ and then solving for the desired rate parameter to be replaced. This means that the classical approach can be described by drawing the resulting rate from a Dirac delta distribution prior with

$$k_1 \sim \delta(k_1 - \tfrac{k_5 k_6 k_7 k_8}{k_2 k_3 k_4}) \tag{60}$$

Following *Equation 60*, we can model microscopic-reversibility with any hierarchical prior distribution whose limit for a vanishing variance is *Equation 60*. For mathematical convenience, we defined the hierarchical prior by a sharply peaking beta distribution

$$k_1^{\star} \sim \text{beta}(100.01, 100.01) \tag{61}$$

and by rescaling and adding an offset

$$k_1 = \tfrac{k_5 k_6 k_7 k_8}{k_2 k_3 k_4} \cdot 0.995 + 0.01 \cdot k_1^{\star} \tag{62}$$

we derived a conditional prior which allows at maximum a ±0.005 relative deviation from the strict microscopic-reversibility. The ±0.005 micro-reversibility constraint is applied in (*Figure 7b–d*). In this way, one could model or even test possible small violation of microscopic-reversibility if smaller beta parameters such as $\text{beta}(1, 1)$ would be chosen.

## Acknowledgements

The authors are grateful to E Schulz for designing a software to simulate channel activity in ensemble patches and to Th Eick for performing the simulations. FP acknowledges funding from the Yen Post-Doctoral Fellowship in Interdisciplinary Research and from the National Cancer Institute of the National Institutes of Health (NIH) through Grant CAO93577. The authors are also indebted to M Habeck and I Schroeder for comments on the manuscript, to M Bücker for help with the computer cluster

at the Friedrich Schiller University Jena, and to F Noé, R Blunck, G Mirams and S Pressé for helpful discussions. This work was supported by the Research Unit 2,518 DynIon (Project P2) and the TRR 166 ReceptorLight (Project A5) of the Deutsche Forschungsgemeinschaft to KB. Model parameterization, prior distributions, and the general time-reversible model in bayesian phylogenetics. *Systematic Biology*, 53(6):877 – 888.

## Additional information

### Funding

| Funder | Grant reference number | Author |
|---|---|---|
| Deutsche Forschungsgemeinschaft | TRR 166 ReceptorLight (Project A5) of the | Klaus Benndorf |
| Deutsche Forschungsgemeinschaft | Research Unit 2518 DynIon (Project P2) | Klaus Benndorf |

The funders had no role in study design, data collection and interpretation, or the decision to submit the work for publication.

### Author contributions

Jan L Münch, Conceptualization, Formal analysis, Investigation, Methodology, Software, Validation, Visualization, Writing – original draft; Fabian Paul, Methodology, Writing – original draft; Ralf Schmauder, Data curation, Methodology, Supervision, Writing – original draft; Klaus Benndorf, Conceptualization, Funding acquisition, Project administration, Supervision, Writing – original draft

### Author ORCIDs

Jan L Münch  http://orcid.org/0000-0002-9177-6466
Klaus Benndorf  http://orcid.org/0000-0002-0707-4083

### Decision letter and Author response

Decision letter https://doi.org/10.7554/eLife.62714.sa1
Author response https://doi.org/10.7554/eLife.62714.sa2

## Additional files

### Supplementary files

- Transparent reporting form
- Source code 1. cPCF data simulation code.

### Data availability

We included the simulated data time traces into supporting files and we uploaded the source code on  https://github.com/JanMuench/Tutorial_Patch-clamp_data  and  https://github.com/JanMuench/Tutorial_Bayesian_Filter_cPCF_data.

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

# Appendix 1

## Efficiency of the Hamiltonian Monte Carlo sampler

Bayesian posterior sampling requires a fast and efficient sampler which requires in turn a fast evaluation of the likelihood and prior distribution. Optimally, the calculation time should increase as little as possible with the amount of data. This is why we chose the KF framework and parallelized the processing of a time trace across many CPUs. Compared to the maximization of the likelihood, sampling from the posterior requires approximately one to multiple orders of magnitude times more evaluations of the likelihood function to have a decent representation of the posterior. For example, in *Moffatt, 2007* both algorithms need roughly $10^2$ to $10^3$ evaluations of the likelihood function to converge. This is only a small fraction of the number of evaluations needed here to obtain a converged estimate of the posterior (in fact the typical "warm-up" phase of the sampler which we used is barely half completed with such a small number of evaluations).

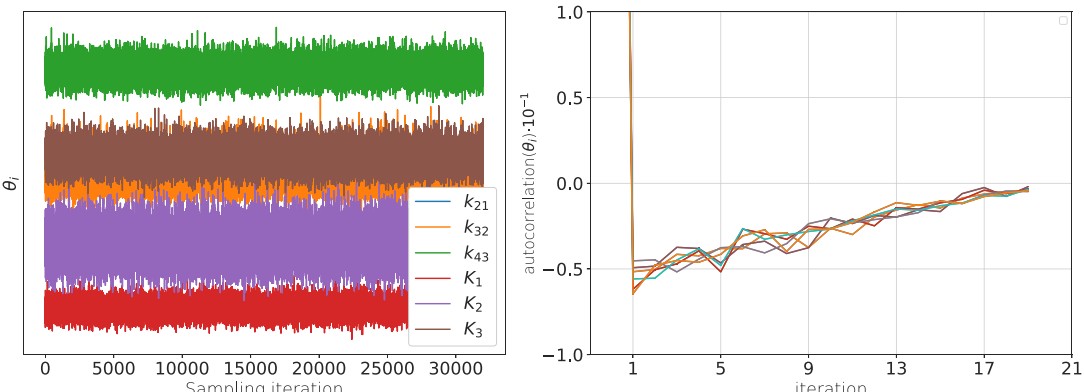

**Appendix 1—figure 1.** The posterior samples display autocorrelation which lowers the effective sample size. (**a**) Sample traces from the 6-dimensional posterior of the rate matrix recorded after the warm-up of the sampler. (**b**) These traces are anti-correlated (in time) such that the effective sample size with which all quantities of relevance are estimated is smaller than the actual number of samples. Note that even at one iteration the absolute value of the auto-correlation is never larger than 0.05. The standard error of the bootstrapped mean auto-correlation is in the order of $10^{-4}$.

The ratio of the number of used samples to the number of likelihood evaluations can serve as a simple efficiency measure. However, this ratio is a too optimistic measure as revealed by a look into the internal dynamics of Hamiltonian Monte Carlo sampler (HMC). HMC treats the current parameter sample as part of a phase space point of a (dynamic) Hamiltonian system. The force acting on the system is the derivative of the negative logarithm of the posterior density *Betancourt, 2017*. Each suggested sample is derived from a ballistic movement with random kinetic energy in the augmented parameter space. Calculating this movement is done by integration of the corresponding Hamiltonian equations. Therefore, one suggested sample requires many evaluations of the gradient of the log posterior distribution. How many evaluations were needed in turn depends on the shape of the posterior and the length of the ballistic movement.

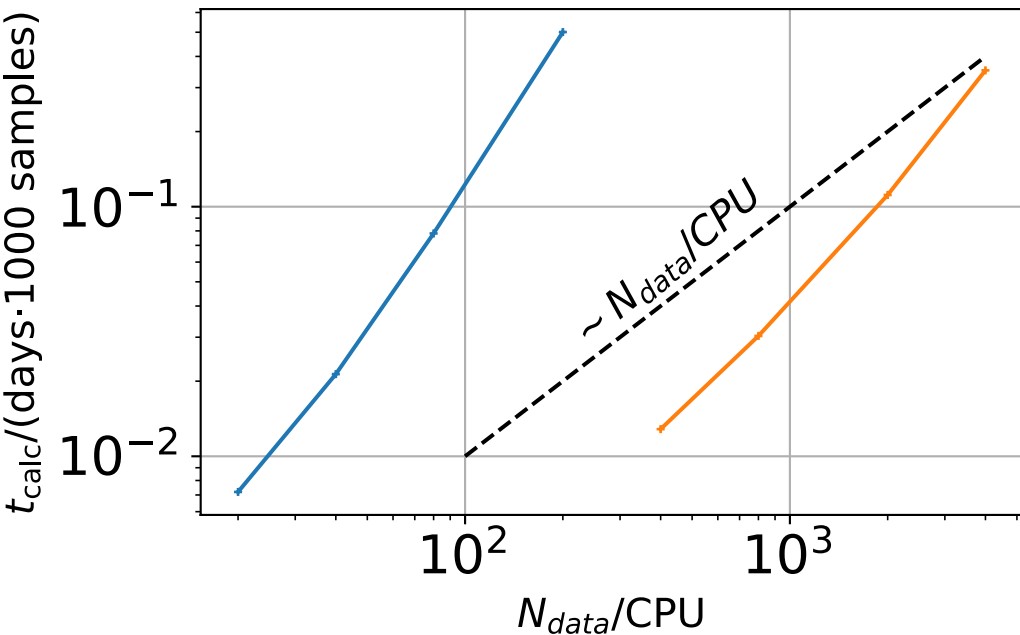

**Appendix 1—figure 2.** Computational time in units of 1000 samples per day vs $N_{\mathrm{data}}/\mathrm{CPU}\,10^{-4}$. The two colors represent two different implementations. The difference between the two implementations is that the derived quantities which are the predicted mean signal and the predicted signal variance for each data point in the less efficient sampling algorithm are saved. In the more efficient algorithm these derived quantities are discarded.

Strong changes of the gradient of the posterior (high curvature of the posterior) require more integration steps. Setting all those sampler parameters requires either expert knowledge about the statistical model and sampler or a sampler which automatically learns optimal sampler parameter settings (warm-up) before it starts the actual sampling of the parameters of interest. Nevertheless, HMC generates samples (*Appendix 1—figure 1a*) from high-dimensional, correlated posteriors more efficiently than many other classical Markov chain Monte Carlo methods (*Betancourt, 2017*). By more efficient we mean the product of the speed of drawing one correlated sample with the amount of iterations needed for the autocorrelation function of those samples to vanish sufficiently. By construction, the samples from any Markov chain Monte Carlo method are correlated (*Appendix 1—figure 1b*). The quality of the adaptive NUTS HMC sampler provided by Stan can by seen in *Appendix 1—figure 1b*. The sample traces show a small anticorrelation whose absolute value is hardly larger than 0.05. This is achieved by letting the sampler adapt to the geometry of the posterior in the warm-up phase. The samples collected during the warm-up phase are discarded.

Hereinafter, we show that likelihood and sampler considerations are one aspect of computational efficiency. The other is the implementation of a parallelized algorithm. The blue curve (*Appendix 1—figure 2*) represents the scaling of the KF with an implementation of the algorithm which saves besides the sampled parameters the predicted mean and the covariance of the signal. This means for cPCF data that for each parameter sample $5 \cdot N_{\mathrm{traces}} \cdot N_{\mathrm{data}}/\mathrm{CPU}$ more memory operations are needed on the cluster node. Note, since we assigned each ligand concentration jump and the following data points to its own CPU the following relation holds: $N_{\mathrm{data}}/\mathrm{CPU} = N_{\mathrm{data}}/\mathrm{trace}$. Skipping saving those derived quantities (orange curve) creates a speedup of roughly 2 orders of magnitude. The derived quantities are redundant quantities which we suggest to calculate by feeding a small subsample of the posterior samples to the KF filter which then reanalyzes the traces by the given draws but does not redraw from the posterior. See Git-Hub: https://github.com/JanMuench/Tutorial_Patch-clamp_data.

We compare the computation time for the KF (green) and the RE approach (blue) for the 4-state model (*Appendix 1—figure 3a*) versus the data quality by $N_{\mathrm{ch}}$. The calculation time $t_{\mathrm{calc}}$ rises for both algorithms roughly like $\sim \sqrt{N_{\mathrm{ch}}}$. For each curve the likelihood evaluations stay constant for the whole plot. Thus this scaling relates to the integration time of the HMC sampler. Taking the average of the last 5 calculation times shows that the KF is about $2.7 \pm 0.2$ times slower than the RE approach. The computation time for cPCF data of the 5-state-2-open states model (*Appendix 1—figure 3b*)

shows a similar $\sim \sqrt{N_{\mathrm{ch}}}$-regime until it seems to become independent of $N_{\mathrm{ch}}$. Note, the different $N_{\mathrm{ch}}$ interval which is displayed.

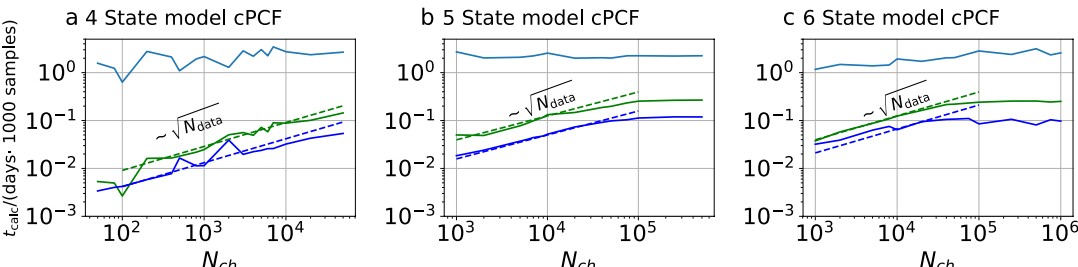

**Appendix 1—figure 3.** Computational time for cPCF data. Computational time in units of 1000 samples per day vs. the amount of ion channels per trace $N_{\mathrm{ch}}$. The green curve belongs to the results from the KF, the blue curve to the results from the RE approach. The brighter blue curve corresponds to the ratio of computational time. a, four-state model with cPCF data. The posterior of the RE approach was sampled from three independent chains. We drew 18,000 samples per chain which included a large fraction of 6000 warm-up samples per chain. The posterior of the KF was sampled from 4 independent chains. We drew 26,000 samples per chain which included a large fraction of 4000 warm-up samples per chain. b Computational time of the 5-state-2-open-states model with cPCF data. The KF algorithm drew 9000 samples which included a large fraction of 6,000 warm-up samples per chain. The RE algorithm drew 10,000 samples which included a large fraction of 5,000 warm-up samples per chain. In total, we used four independent chains. c, Computational time of the 6-state-1-open-state model with cPCF data. The KF algorithm drew 10,000 samples which included a large fraction of 6000 warm-up samples per chain. The RE algorithm drew 12,000 samples which included a large fraction of 7000 warm-up samples per chain. In total, we used four independent sampling chains.

Taking the average of the last 4 computation times shows that the KF is about $2.24 \pm 0.01$ times slower than the RE approach. Surprisingly, with increasing model complexity the ratio of the computation time becomes not necessarily larger. The computation time for cPCF data of the 6-state-1-open state model (*Appendix 1—figure 3c*) shows a similar $\sim \sqrt{N_{\mathrm{ch}}}$-regime until it becomes independent of $N_{\mathrm{ch}}$. Taking the average of the last 4 calculation times reveals that the KF is about $3.3 \pm 0.3$ times slower than the RE approach. With increasing model complexity, the ratio of the computation times becomes larger. Note, that there is little variation of the absolute calculation time of the algorithm across model complexity which is a good prospect for more complex kinetic schemes. Nevertheless, to achieve this computational speed we used in total 72 CPUs on one node, 20 CPUs per chain for 20 time traces. Thus, nodes with more then 80 CPUs would enable a better performance. Overall, the more complex likelihood calculations of the KF require 2–3 times more calculation for the tested models and cPCF data. Surprisingly, for most part of the displayed interval the KF is faster than the RE approach (*Appendix 1—figure 4a*) for PC data. Considering the more expensive likelihood evaluation (prediction and correction) of the KF, this result reminds that this computational time benchmark is a downstream test that integrates all aspects of the sampling. One possible explanation could be the following: We describe in the main text (*Figures 4b1–6*) for the RE approach, that due to the treatment of every data point as an individual draw from a multinomial distribution, HDCVs (*Figures 5–6*) are too narrow. This problem gets exacerbated (*Figure 8*) as we used 10 times more data points which we interpret as the cause of the KF being faster than the RE approach (*Appendix 1—figure 4a*). The narrower the posterior is the more integration steps from the sampler are needed to calculate the proposed trajectory with sufficient precision. Another example displayed (*Appendix 1—figure 4b*), has an increased kinetic scheme complexity. Note that we reduced the analyzing frequency 10 times to be comparable with the analyzing frequency of cPCF data. Under this condition, the intuition from likelihood calculation complexity is confirmed that the KF is slower. Taking the average of the last 4 calculation times shows that the KF is about $1.8 \pm 0.2$ slower than the RE algorithm for PC data.

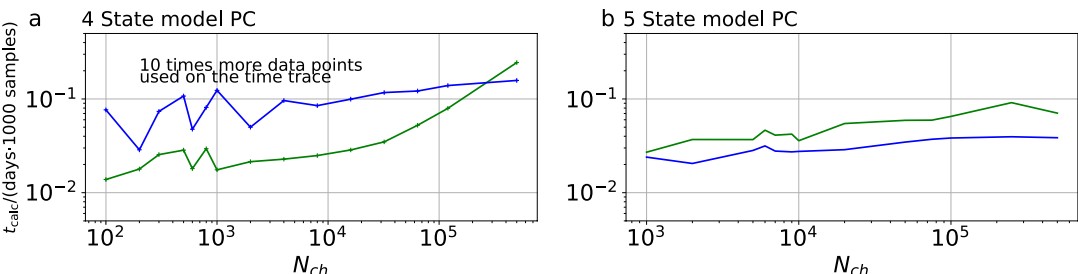

**Appendix 1—figure 4.** Computational time for PC data. The computational time in units of 1,000 samples per day are plotted vs the amount of ion channels per trace $N_{\mathrm{ch}}$. The green curve belongs to the results from the KF, the blue curve to the results from the RE approach. a, Four-state model with PC data. In contrast to the cPCF data (*Appendix 1—figure 3*), $f_{ana}$ is increased by an order of magnitude. Thus, 10 times more data points per time trace and CPU are evaluated. These results were obtained by four independent chains which drew 9000 samples per chain, including a large fraction of 7000 warm-up samples per chain.b, Five-state model with PC data. Note that here $f_{ana}$ equals the analyzing frequency of cPCF data (*Appendix 1—figure 3*).

## Convergence diagnostics

In the limit of infinitely many samples drawn from a posterior by any MCMC method, given that posterior space is simply connected, the sampling is guaranteed to converge to the typical set *Betancourt, 2017*. For a finite amount of samples from the posterior it is not guaranteed that the samples are representative for the typical set of the posterior. To ensure that the trace has converged to the typical set is an active field of statistical research. A discussion of these important consistency checks (*Gelman and Rubin, 1992a*; *Gelman et al., 2013*; *Vats and Knudson, 2018*; *Gabry et al., 2019*; *Vehtari et al., 2021*) ensuring that the sampled traces represent the typical set, is beyond the scope of this paper. In short, we usually worked with 4 independent sampling chains of the posterior (*Gelman and Rubin, 1992b*). On the one hand, we visually checked that all 4 chains do report the same typical set to verify that the sampler is fine. This convergence can be monitored by the $\hat{R}$ statistics (*Gelman et al., 2013*; *Vehtari et al., 2021*). $\hat{R}$ evaluates differences between the within- and between-chain parameter estimates. A bad mixing is indicated by large values of $\hat{R}$. Perfectly converged chains would be reported with $\hat{R} = 1$. Additionally, the effective sample size $N_{\mathrm{eff}}$ of each object of interest needs to be monitored. Both of them are reported by the "summary" method from the "fit" object in PyStan. We show (*Appendix 1—table 1*) an edited version of the output from this method. The last two columns would indicate signs of missing convergence. The second column reports the Markov sampling error telling if we drew enough samples in total. Note that any quantity far out in the tails would have a much larger error. There were cases when the sampler was not fine for all chains, meaning that 3 chains showed the same posterior but one showed something different. In such cases we simply discarded that chain. This occurred in particular more often with the RE approach. As we are benchmarking methods and know the true parameter values, it was rather obvious which chain did not work well. In particular, the successful binomial test of the HDCV is a downstream test that confirms besides the assumptions of the algorithm also whether the sampler worked sufficiently. For the Bayesian filter, sampler problems occurred at very low ion channel counts below $N_{\mathrm{ch}} = 200$.

**Appendix 1—table 1.** Typical summary statistics of the posterior as printed by the "summary" method from the "fit" object in Pystan.

We deleted the 25% and 75% percentiles to decrease the size of the table and changed some of the names with respect to the used name in the algorithm to the symbol used in this article. Note that we assumed to measure 2 ligand concentrations (which includes 4 jumps) from one patch. Thus, we assumed for 10 ligand concentrations that they are coming from 5 patches which leads to a five-dimensional channels per patch vector $\mathbf{N}_{\mathrm{ch}}$. The bias of $N_{\mathrm{ch}}$ discussed in the main article, is also observed here.

|          | mean   | se$_{mean}$ | sd    | 2.5%   | 50%    | 97.5%  | n$_{eff}$ | Rhat |
|----------|--------|-------------|-------|--------|--------|--------|-----------|------|
| $k_{21}$ | 507.65 | 0.06        | 11.91 | 484.74 | 507.46 | 531.62 | 41262     | 1.0  |
| $k_{32}$ | 481.15 | 0.16        | 28.63 | 428.34 | 480.19 | 540.32 | 33610     | 1.0  |

*Appendix 1—table 1 Continued on next page*

*Appendix 1—table 1 Continued*

| | mean | $se_{mean}$ | sd | 2.5% | 50% | 97.5% | $n_{eff}$ | Rhat |
|---|---|---|---|---|---|---|---|---|
| $k_{43}$ | 509.95 | 0.04 | 6.99 | 496.39 | 509.91 | 523.77 | 36006 | 1.0 |
| $K_1$ | 0.5 | 2.6e-5 | 5.2e-3 | 0.49 | 0.5 | 0.51 | 40548 | 1.0 |
| $K_2$ | 0.51 | 6.7e-5 | 0.01 | 0.49 | 0.51 | 0.54 | 33228 | 1.0 |
| $K_3$ | 0.49 | 4.2e-5 | 7.5e-3 | 0.47 | 0.49 | 0.5 | 31445 | 1.0 |
| $N_{ch}[1]$ | 1007.1 | 0.14 | 20.05 | 968.84 | 1006.8 | 1047.2 | 19838 | 1.0 |
| $N_{ch}[2]$ | 1006.1 | 0.14 | 20.04 | 967.83 | 1005.8 | 1046.0 | 19874 | 1.0 |
| $N_{ch}[3]$ | 1003.9 | 0.14 | 20.0 | 965.66 | 1003.6 | 1043.8 | 19877 | 1.0 |
| Nch[4] | 1007.0 | 0.14 | 20.09 | 968.67 | 1006.6 | 1047.1 | 19843 | 1.0 |
| $N_{ch}[5]$ | 1005.9 | 0.14 | 20.09 | 967.61 | 1005.6 | 1045.9 | 19874 | 1.0 |
| i | 1.0 | 7.0e-5 | 9.8e-3 | 0.98 | 1.0 | 1.02 | 19971 | 1.0 |
| $\sigma^2_{exp}$ | 0.25 | 1.2e-5 | 2.5e-3 | 0.25 | 0.25 | 0.26 | 43142 | 1.0 |
| $\sigma^2_{op}$ | 2.5e-3 | 1.2e-7 | 2.5e-5 | 2.5e-3 | 2.5e-3 | 2.5e-3 | 42707 | 1.0 |
| $\sigma^2_{fluo}$ | 1.3e-5 | 2.3e-8 | 4.6e-6 | 1.0e-5 | 1.2e-5 | 2.5e-5 | 41306 | 1.0 |
| $\lambda$ | 9.98 | 1.1e-4 | 0.02 | 9.94 | 9.98 | 10.02 | 35109 | 1.0 |

## HMC parameter settings

The sampling efficiency of an HMC algorithm is sensitive to three parameters (*Neal, 2011*, *Hoffman and Gelman, 2014*). The three parameters, discretization time $\epsilon$, the metric $\mathbf{M}$ and the number of steps taken $L$ can be predefined or/and are adapted in the warm-up phase of the sampler. A typical number of iterations $N_{warm}$ in the warm-up phase for each chain is between $N_{warm} = 3 \cdot 10^3 \ldots 7 \cdot 10^3$. Since a sufficient $N_{warm}$ depends on the correlation structure of the posterior, which depends on the quality and quantity of the data, we did not engage to optimize the number of warm-up steps. We rather checked whether for a set of different data (for example with different $N_{ch}$) all chains are fine.

## Stan

Stan is a probabilistic programming language that provides the research community with easy access to HMC sampling, variational Bayes, and optimization procedures like Maximum posterior/ML approaches. Stan uses automatic analytical differentiation of the statistical model such that no analytical treatment by hand has to be done for any tested model. The only requirement from the user is to define a statistical model of the data. The Stan compiler calculates the analytical derivatives of the model and then translates the statistical model to C++code and after that into an executable program. Furthermore, Stan uses adaptive HMC such that the length of the ballistic trajectory as well as the variance of the random kinetic energy to create the ballistic trajectory and the number of integration steps are optimized automatically. Then the program can be started from any high-level data processing language of the user's choice such as Python, Matlab, Julia, R or from the command line.

## Appendix 2

## Markov models for a single ion channel

Markov models and rate models are widely used for modeling molecular kinetics. They provide an interpretation of the data in terms of a set of functional states and the transition rates between these states. Markov models can be estimated from experimentally recorded data as well as from computer simulation data. The use of Markov models with one-step memory is supported by the concept of the molecular free energy landscape. Molecular energy landscapes are typically characterized by conformationally well-defined free-energy minima that are separated by free-energy barriers. State transitions in molecules are thermally activated barrier-crossing events on this landscape (*Frauenfelder et al., 1991*) leading to a rapid equilibration of the system in the vicinity of this new minimum. Memory of other minima that have been visited in the past is not required. Regarding the wide spectrum of time scales at which processes in a protein take place, one has to be aware that there is typically a small number of relaxation modes with excessively long autocorrelation times and many relaxation modes with much faster autocorrelation times. To model the slow, experimentally accessible processes, it is sufficient to retain the small number of slow modes (*Noé et al., 2011*). It has been shown rigorously that working with the set of slow modes is equivalent to model the state dynamics with a small number of fuzzily defined metastable states in the full conformational space (*Deuflhard and Weber, 2005*) Later it has been shown that the set of slow modes can be well approximated with a hidden Markov model (*Noé et al., 2013*).

## Appendix 3

## Eigenvalues and fitting of sums exponential decays: Mean time evolution of the spectral components of the time evolution

From the experimental perspective we are used to fit sums of exponential decays to time series data of any signal such as currents or fluorescence.

$$I(t) = I_\infty + \sum_{i=1}^{M} I_i \exp(-t/\tau_i) \tag{63}$$

The mathematical background for this procedure is the spectral decomposition of the solution of the corresponding RE equation

$$\frac{d\mathbf{p}(t)}{dt} = \mathbf{K}\mathbf{p}(t), \tag{64}$$

which is a vector differential equation whose evolution is governed by the rate matrix $\mathbf{K}$. The general solution to the mean time evolution of the system is

$$\mathbf{p}(t) = \exp(\mathbf{K} \cdot t)\mathbf{p}(t = 0),$$

A Markov state model has as many real valued eigenvalues as it has states $M$. With those eigenvalues one can decompose the solution into its spectral components

$$\mathbf{p}(t) = C_1 \cdot \exp(\alpha_0 t)\tilde{\mathbf{p}}_0 + \sum_{i=1}^{M-1} C_i \cdot \exp(\alpha_i t)\tilde{\mathbf{p}}_i$$

All of the eigenvalues obey $\alpha \leq 0$. One of them always fulfills $\alpha_0 = 0$ which belongs to the equilibrium probability distribution of the ion channel. With $exp(0) = 1$ we can write the sum as

$$\mathbf{p}(t) = C_1 \cdot \tilde{\mathbf{p}}_0 + \sum_{i=1}^{M-1} C_i \cdot \exp(\alpha_i t)\tilde{\mathbf{p}}_i \tag{65}$$

The eigenvalue $\alpha_0 = 0$ determines the equilibrium solution. Due to $\alpha_i \leq 0$ the solution is stable and converges to

$$\mathbf{p}(t = \infty) = C_1 \cdot \tilde{\mathbf{p}}_0$$

The time evolution of the mean signal in its spectral components can then be derived

$$I(t) = \mathbf{H}\left(C_1 \cdot \tilde{\mathbf{p}}_0 + \sum_{i=1}^{M-1} C_i \cdot \exp(\alpha_i t)\tilde{\mathbf{p}}_i\right) \tag{66}$$

by multiplying with the observation matrix $\mathbf{H}$ and setting $\alpha_i = -\frac{1}{\tau_i}$ we derived *Equation 63*, *Colquhoun and Hawkes, 1995a*. To derive the likelihood based on this deterministic approximation *Milescu et al., 2005* one assumes for each data point that the ensemble state $\mathbf{n}(t)$ is drawn from

$$\mathbf{n}(t) \sim \text{multinomial}(N_{\text{ch}}, \mathbf{p}(t)) \tag{67}$$

with the additional approximation of the multinomial distribution by a multivariate normal distribution. Subsequently, the equation

$$\mathbf{y}(t) \sim \text{normal}(\mathbf{H}n(t), \mathbf{\Sigma}(t)) \tag{68}$$

provides the contribution of the recording system to the signal of the ensemble state. In contrast, the KF approximates a

$$\mathbf{n}(t + \Delta t) \sim \text{generalized-multinomial}(\mathbf{n}(t), \mathbf{T}) \tag{69}$$

distribution with multivariate normal distributions (see Methods) and then the details of the experimental setup are adjoined

## Appendix 4

## A note about Kalman filtering within other frameworks such as variational Bayes, maximum a posteriori and ML

We like to emphasize that the key property of the Kalman filter likelihood for an autocorrelated time series (see *Equation 9*) to factorize over time enables us to treat each data point as if it were independent. The alternative is to treat the whole time series as one high-dimensional data point as done in *Celentano and Hawkes, 2004*. A ML Kalman filter (*Moffatt, 2007*) would try to optimize *Equation 9*, which is usually done in the log space to improve numerical stability. Taking the logarithm of *Equation 9* gives also an insight for the reader, who might be more experienced in using residual sums of squares for fitting.

$$\log \mathbb{L}(\mathcal{Y}_T | \boldsymbol{\theta}) = \log \prod_{t=2}^{N_T} \mathcal{N}(\mathbf{y}_t | \mathbf{H}\mathbb{E}[\mathbf{n}_t], \mathbf{H}\mathbf{P}_t\mathbf{H}^\top + \boldsymbol{\Sigma}_t) = \sum_{t=2}^{N_T} \log \mathcal{N}(\mathbf{y}_t | \mathbf{H}\mathbb{E}[\mathbf{n}_t], \mathbf{H}\mathbf{P}_t\mathbf{H}^\top + \boldsymbol{\Sigma}_t), \quad (70)$$

where we used the logarithmic algebra rule $\log(\prod_i a_i) = \sum_i \log(a_i)$ Using the functional form of the multi-variate normal distribution, we arrive at

$$\log \mathbb{L}(\mathcal{Y}_T | \boldsymbol{\theta}) = \sum_{t=2}^{N_T} \log(\frac{1}{\sqrt{(2\pi)^2 \det(\mathbf{H}\mathbf{P}_t\mathbf{H}^\top + \boldsymbol{\Sigma}_t)}})(-0.5) \cdot (\mathbf{y}_t - \mathbf{H}\mathbb{E}[\mathbf{n}_t])[\mathbf{H}\mathbf{P}_t\mathbf{H}^\top + \boldsymbol{\Sigma}_t]^{-1}(\mathbf{y}_t - \mathbf{H}\mathbb{E}[\mathbf{n}_t])^\top, \quad (71)$$

which is a vectorial residual sum of squares whose summands are weighted by $\mathbf{H}\mathbf{P}_t\mathbf{H}^\top + \boldsymbol{\Sigma}_t$. The past observations of time series are encoded in $\mathbf{P}_t$ and $\mathbb{E}[\mathbf{n}_t]$, which are calculated with the Kalman filtering equations before the likelihood can be evaluated.

Generally, the user of our algorithm is not restricted to the full Bayesian inference. He/she can also stick to optimizing *Equation 71*. This is because in the probabilistic programming language Stan one only formulates the statistical model. The method to evaluate the statistical model and the data is provided by the Stan compiler and the high level programming language interface. Together they allow to switch quickly between HMC sampling, variational Bayesian posterior estimation, maximum a posteriori and ML. In particular, the variational Bayesian posterior might be a way to ease the computational burden due HMC sampling while keeping many of the benefits of knowing the full posterior distribution.

In the previous versions before PyStan 3.0 all three approaches were supported. Currently, with PyStan 3.0 only HMC sampling of the posterior is supported but, if desired, any other interface can be used (Matlab, Julia, R, CmdStan, Pystan 2.9, CmdStanPy,…) to perform optimization or variational Bayesian approaches. No actual changes in the Stan code are required, eventually with the exception that the CPU based parallelization needs to be eliminated. Given that one chooses optimization, the differences between maximum a posteriori/penalized maximum likelihood or only ML are just a matter of prior choice which need to be coded into the statistical model. Apart from those minor points, there is no need of extensive re-implementation of the code.

## Appendix 5

### The fluorescence signal of cPCF experiments

First four moments of a photomultiplier signal

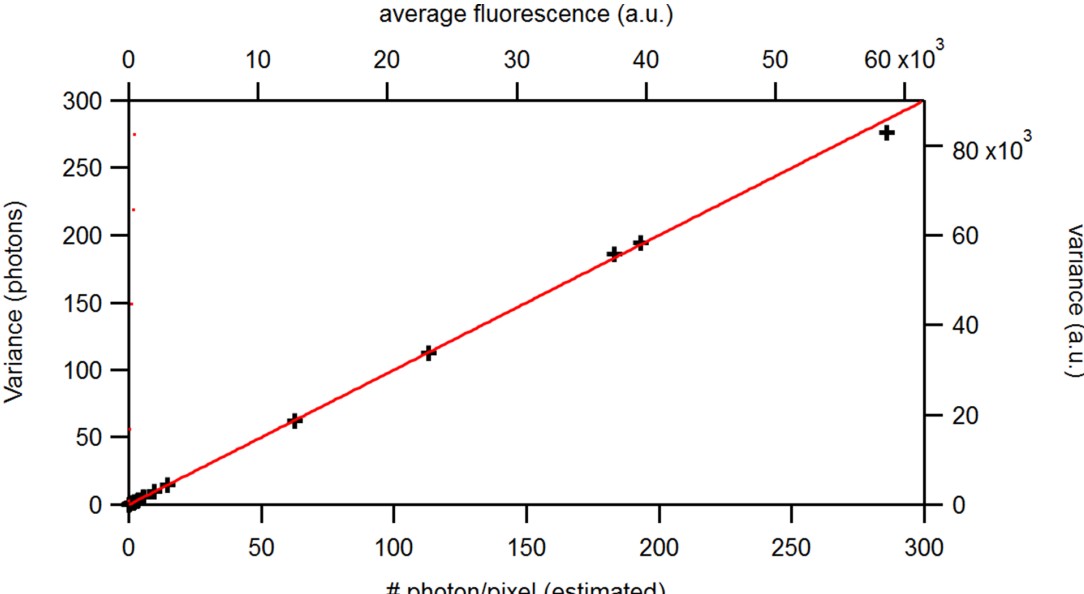

**Appendix 5—figure 1.** Benchmark of the signal variance vs its mean for experimental solution data recorded under cPCF conditions: The concentrations of the fluorescent ligand were 0.25, 3, and 15 $\mu M$ and a reference dye was present. The laser intensities covered 1.6 orders of magnitude at constant detection settings. The data points were obtained from $1.4 \cdot 10^6$ pixel. The red and blue lines indicate the theoretical prediction for a Poisson and Gamma distribution, respectively, assuming $\theta = 1$. The linear relation allows to relate the measured a.u. (top, right axis) to photons (bottom, left axis). The inset provides the corresponding log-log plot. Important for the KF algorithm is that skewness and excess kurtosis is small.

In this work, the KF analysis assumes Poisson statistics for the fluorescence signal in cPCF. Many commercial microscopes are not equipped with photon counting detectors or detectors are not operated in photon-counting mode, often to due to ease of use or limitation in dynamic range. Therefore, it is important to verify that the fluorescence signal follows, at least approximately, Poisson counting statistics. In particular, for the KF it is assumed that higher order statistics, such as skewness and excess kurtosis, vanish. The central assumption of the derivation of our Bayesian filter is $\mathrm{var}[y_{\mathrm{fl}}] = \mathbb{E}[y_{\mathrm{fl}}]$

Here we show that this assumption for the detectors used in our previously used LSM 710 (Carl Zeiss, Jena) system (**Biskup et al., 2007**; **Kusch et al., 2010**) is fulfilled by re-scaling to photon-numbers. The measured variance obeys $\mathrm{var}[y_{\mathrm{fl}}] = a \cdot \mathbb{E}[y_{\mathrm{fl}}]$. It depends linearly on the mean signals (**Appendix 5—figure 1**).

$$\mathrm{var}[y] = a\mathbb{E}[y] \tag{72}$$

$$\mathrm{var}[\mu x] = a\mathbb{E}[\mu x] \tag{73}$$

$$\mu^2 \, \mathrm{var}[x] = a\mu \, \mathrm{E}[x] \tag{74}$$

For the scaled signal $x$ being Poisson distributed follows $\mu = a$. Re-scaling of the signal by $1/a$ provides approximately Poisson distributed values. A linear fit yields $a = 205$ a.u.(16 bit)/photon (for 680 V PMT voltage, 3.26 $\mu s$ pixel dwell time). (**Appendix 5—figures 2 and 3**) show that excess kurtosis and skewness remain small at all levels of photons/pixel but are somewhat higher than theoretically predicted for Poisson-distributed data. The proportionalities are correctly described by the Poisson distribution assumption but the skewness and the kurtosis are too small by a constant factor of $\sqrt{2}$ and 4, respectively. This finding has to be verified for different experimental conditions, because at lower

concentration/particle densities and higher count rates, particle number fluctuations can dominate statistics (**Brown et al., 2008**). For comparison another option would be a Gamma distribution which has the mean and the variance of $\mathbb{E}[y] = k\theta$ and $var[y] = k\theta^2$, respectively. Thus, the applied scaling requires that $\theta = 1$. The Gamma distribution has a higher skewness by factor two (independently of θ) than a Poisson distribution and overscores the skewness and excess kurtosis of the detector. For simplicity only the Poisson distribution is considered in this work. In conclusion: Typical cPCF fluorescence signal detection rates are well approximated by a Gamma or Poisson distribution which in turn have the desired property that can be approximated by a normal distribution.

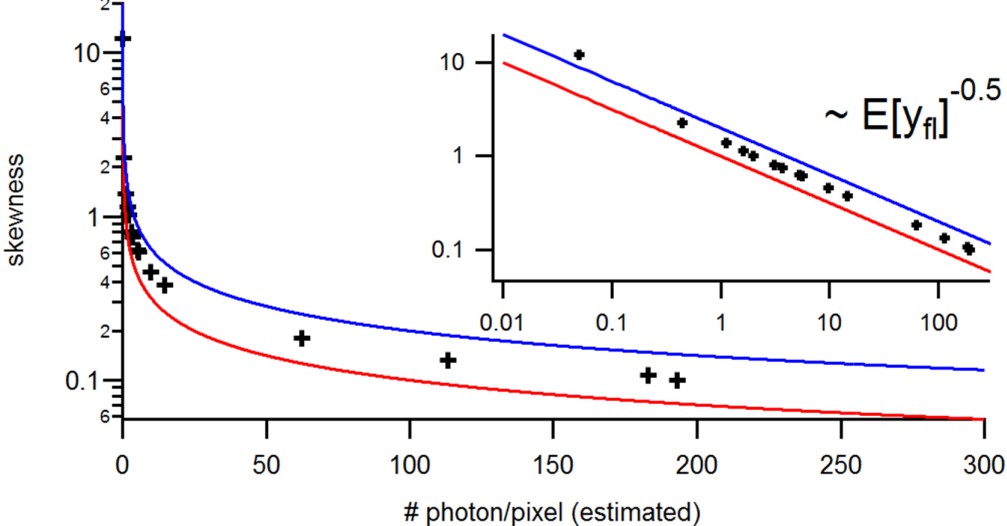

**Appendix 5—figure 2.** Benchmark of the signal Skewness vs its the mean for the identical experimental solution data under cPCF conditions. The Skewness is small but the values are slightly larger than theoretically predicted. The inset provides the corresponding log-log plot. Important for the KF algorithm is that skewness and excess kurtosis is small.

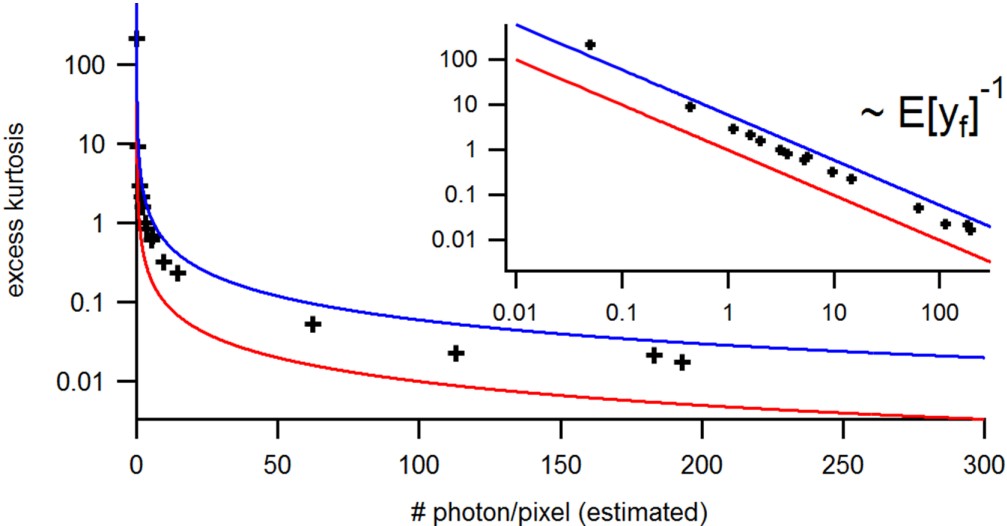

**Appendix 5—figure 3.** Benchmark of the signal Kurtosis vs its mean for the identical experimental solution data under cPCF conditions. The Kurtosis is small but the values are slightly larger than theoretically predicted. The inset provides the corresponding log-log plot. Important for the KF algorithm is that skewness and excess kurtosis is small.

## Background noise statistics

In cPCF measurements with fluorescence-labeled ligands, the signals of the ligands bound to the receptors overlap with the signals from freely diffusing fluorescence-labelled ligands in the bulk. This bulk signal is subtracted from the total signal (*Biskup et al., 2007*). While the mean difference signal $y_{fl,k}(t)$ of the confocal voxel $k$ represents the bound ligands in that voxel, its noise $y_{\zeta,k}(t)$ originates from both bound and bulk ligands. The additional bulk signal, e.g. the fraction of bulk solution inside that voxel, varies from voxel to voxel and can hardly be described theoretically. Nevertheless, it can be determined experimentally (*Biskup et al., 2007*). At low expression levels or at ligand concentrations above low nano-molar levels, this background signal is not negligible. It scales linearly with the ligand concentration, while the signal from bound receptors depends on the affinity, as estimated by the concentration of half maximum binding $BC_{50}$, and the number of ion channels in the membrane of the observed volume. The binding signal saturates at high concentrations (*Appendix 5—figure 4*). Thus, both high affinity (low $BC_{50}$) and high expression reduce the relative contribution of the background to the overall signal, improving the signal to noise ratio.

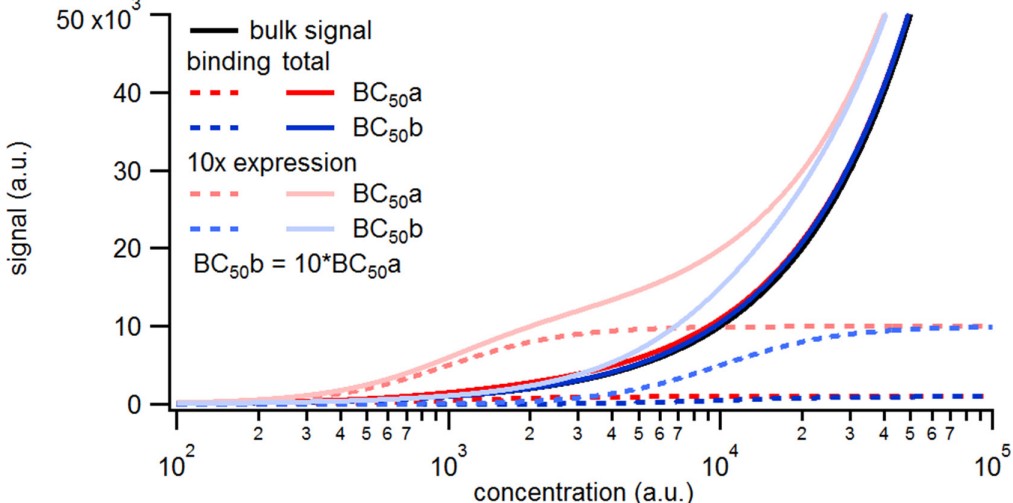

**Appendix 5—figure 4.** Simulated binding signals. a, Comparison of binding of a labeled ligand at two concentrations. A simple two-ligand binding process was simulated with the Hill equation for the two expression levels of 1000 or 10,000 binding sites per patch and a $BC_{50}$ of 1000 ($BC_{50}a$) or 10,000($BC_{50}b$), respectively, given in molecules per observation unit. The observed signal is the sum of the signal from ligands free in solution and bound to the receptors. The solution signal scales linearly with the concentration, while the binding signal saturates.

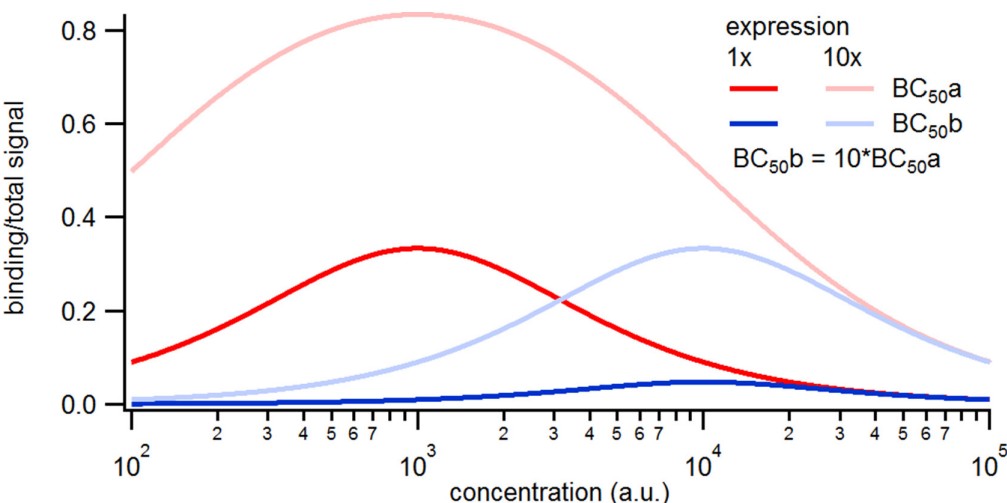

**Appendix 5—figure 5.** Simulated binding signals. Relative contribution of the binding signal to the total signal. Note that the contribution of the binding signal scales linearly with the expression level and inversely with the. $BC_{50}$

Practically, the bulk signal is estimated by counter-staining the solution with a spectrally distinct reference dye (**Biskup et al., 2007**). The spatial distribution of this dye mimics the spatial distribution of the freely diffusing ligands. The bulk absolute concentration as well as the molecular brightness of the reference dye and the labeled ligand differ. Hence, the binding signal is calculated as the average pixel intensity of the scaled difference image between the signal of labeled ligand and reference dye according to

$$y_{\text{fl,k}} = y_{\text{lig,total}} - \hat{\lambda}_{\text{lig,back}} - (y_{\text{fl,ref}} - \hat{\lambda}_{\text{ref,back}}) \frac{\hat{\lambda}_{\text{lig}} - \hat{\lambda}_{\text{lig,back}}}{\hat{\lambda}_{\text{ref}} - \hat{\lambda}_{\text{ref,back}}}, \tag{75}$$

where $\hat{\lambda}_{\text{lig,back}}$ and $\hat{\lambda}_{\text{ref,back}}$ are the arithmetic mean background signals of the ligand and reference dye recorded beyond the membrane where no signal should be recorded. They represent a signal offset which needs to be subtracted. The mean intensities in the bulk, $\hat{\lambda}_{\text{bulk}}$ and $\hat{\lambda}_{\text{ref}}$, are estimated outside the pipette. In order to get the correct scaling, the mean intensities need to be corrected by the respective mean background signals. If $\frac{\hat{\lambda}_{\text{lig}} - \hat{\lambda}_{\text{lig,back}}}{\hat{\lambda}_{\text{ref}} - \hat{\lambda}_{\text{ref,back}}} = 1$ holds then $y_{\text{fl,bin}}$ would be Skellam distributed (**Hwang et al., 2007**). The total signal is then $y_{\text{fl}} = \sum_k y_{\text{fl,k}}$. This procedure creates $\mathbb{E}[y_\zeta] = 0$ but adds an additional noise term $\zeta(t_j)$. For the general case of different intensities, we name the distribution 'scaled Skellam distributed'. The scaling variance of the background noise in each voxel of the difference image

$$\sigma_\zeta^2 = \lambda_{\text{lig}} + \frac{\lambda_{\text{lig}}^2}{\lambda_{\text{ref}}} \tag{76}$$

is derived from simulated data further below. $\lambda_{\text{lig}}$ and $\lambda_{\text{ref}}$ are the fluorescence intensity from the freely diffusing ligands and reference dye molecules per voxel, respectively. $\lambda_{\text{lig}}$ and $\lambda_{\text{ref}}$ are proportional to the volume fraction of the voxel, which is occupied by the bulk, and to the respective concentrations. To achieve a symmetric $\mathbb{P}(\zeta)$, one can set $\lambda_{\text{lig}} = \lambda_{\text{ref}}$. The summed variance of all selected voxels can be tabulated according to

$$\Sigma_{back} = \begin{pmatrix} \sigma_\zeta^2 & 0 \\ 0 & 0 \end{pmatrix} \tag{77}$$

To mimic an experiment which creates time series data $\zeta(t)$, we draw Poisson numbers for the signal from the membrane Poisson($\mathbf{H}\mathbf{n}(t)$) and for the signal from the bulk we draw numbers from the two respective Poisson distributions. Then subtraction of the two background signals is performed according to

$$y_{\text{bulk}} = y_{\text{lig,bulk}} - y_{\text{ref,bulk}} \frac{\lambda_{\text{lig,bulk}}}{\lambda_{\text{ref,bulk}}} \tag{78}$$

assuming that the dark count signal has been correctly subtracted. Then we add the bulk signal to the bound ligand signal. In this way we produce a time trace with colored noise by the Gillespie algorithm and add white noise to time traces as it is observed in real experiments.

## Deriving the moments of the background noise for the difference signal

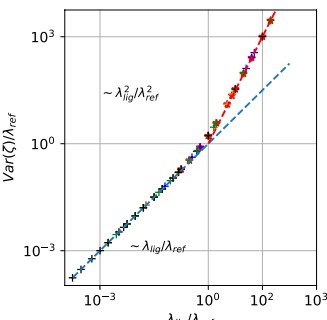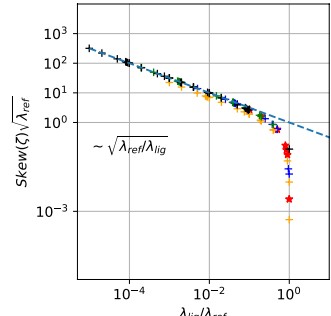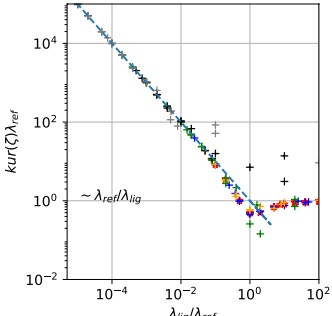

**Appendix 5—figure 6.** Master curves of 2nd till 4th centralized moment of photon counting noise $\zeta$ arising from the difference signal of fluorescent ligands and the dye in the bulk. The curves are created from $4 \cdot 10^5$ draws from Poisson distributions with different combinations of intensities for the reference dye $\lambda_{\text{ref}}$ and of the intensity of the confocal voxel fraction. $\lambda_{\text{lig}}$

For the KF the variance, skewness and kurtosis arising from the background noise has to be calculated. Skewness and excess kurtosis of the distribution have to be small compared to the total variance of the signal including all noise sources because only in this case the KF algorithm can be considered as the optimal solution for the filtering and inference problem (*Anderson and Moore, 2012*). In the following the 2nd to 4th moment of $\zeta$ are derived. The noise intensity parameter of the reference dye $\lambda_{\text{ref}}$ is proportional to $\rho_{\text{ref}}\lambda_{\text{bulk}}$ with $V$ being the confocal volume fraction containing fluorophores and $\rho_{\text{ref}}$ the density of the fluorophores in this volume. In *Appendix 5—figure 6* we deduce master curves for the variance skewness and excess kurtosis of the white noise by drawing $4 \cdot 10^5$ Poisson numbers from the respective Poisson distribution and subtract them from each other according to Appendix 5 *Equation 78*. The variance is derived empirically to be

$$\frac{\sigma_\zeta^2}{\lambda_{\text{ref}}} = \frac{\lambda_{\text{lig}}}{\lambda_{\text{ref}}} + \frac{\lambda_{\text{lig}}^2}{\lambda_{\text{ref}}^2}. \tag{79}$$

In *Appendix 5—figure 6a*, we confirm the intuition $\lambda_{\text{ref}} \to \infty \Rightarrow \text{var}(\zeta) = \lambda_{\text{lig}}$. Optimally, the skewness should be zero to avoid a biased estimate of $\theta$ when the data are analyzed by the KF. Empirically, for $\lambda_{lig} \ll \lambda_{ref}$ the skewness holds

$$\text{skew}(\zeta)\sqrt{\lambda_{\text{ref}}} = \sqrt{\frac{\lambda_{\text{ref}}}{\lambda_{\text{lig}}}} \tag{80}$$

Additionally for $\lambda_{lig} < \lambda_{ref}$ the skewness holds

$$\text{skew}(\zeta)\sqrt{\lambda_{\text{ref}}} \leq \sqrt{\frac{\lambda_{\text{ref}}}{\lambda_{\text{lig}}}} \tag{81}$$

It is zero when $\lambda_{ref} = \lambda_{lig}$. The KF is optimal if the kurtosis excess approaches zero, in other words if $\zeta$ is distributed normally. Empirically the kurtosis holds this

$$\text{kur}(\zeta)\lambda_{\text{ref}} \leq \frac{\lambda_{\text{ref}}}{\lambda_{\text{lig}}} \tag{82}$$

for $\lambda_{ref} \leq \lambda_{lig}$. The relative intensity $\lambda_{lig}$ of the voxel fraction compared to the intensity $\lambda_b$ depends on the affinity of the ligand to the receptor, the number of receptors in the patch, and the density of the fluorophores $p_{lig}$ at the patch. For larger concentrations the ratio should be $\lambda_{lig}/\lambda_{ref}$.

## Appendix 6

### Output statistics of Bayesian filters

#### Classical Kalman Filter without open-channel noise

Assuming that current measurements are only compromised by additive technical white noise $\nu$ but do not contain open-channel noise $\nu_{op}$, then our noise model reduces to

$$y(t) = \mathbf{H}\mathbf{n}(t) + \nu(t) \Leftrightarrow y \sim \mathbb{O}(y|\mathbf{n}) = \text{normal}(\mathbf{H}\mathbf{n}(t), \sigma_m^2) \tag{83}$$

The noise term $\nu_m$ has a mean of $\mathbb{E}[\nu_m] = 0$ and variance $\mathbb{E}[\nu_m^2] = \sigma_m^2 = const$. One has to keep in mind that we have to add an extra variance term originating from the dispersion of channels over the state space, as encoded by $\mathbf{P}(t)$ and $\mathbf{n}(t)$. The uncertainty $\mathbf{P}(t)$ is calculated using Methods *Equation 30*. The variance of the total output is

$$\text{var}(y(t), y(t)) = \mathbb{E}[(y(t) - \mathbb{E}[y(t)])(y(t) - \mathbb{E}[y(t)])^\top] \tag{84a}$$

$$= \mathbb{E}[(y(t) - \mathbf{H}\mathbb{E}[\mathbf{n}(t)])(y(t) - \mathbf{H}\mathbb{E}[\mathbf{n}(t)])^\top] \tag{84b}$$

$$= \mathbb{E}[(\mathbf{H}\mathbf{n}(t) + \nu(t) - \mathbf{H}\mathbb{E}[\mathbf{n}(t)])(\mathbf{H}\mathbf{n}(t) + \nu(t) - \mathbf{H}\mathbb{E}[\mathbf{n}(t)])^\top] \tag{84c}$$

$$= \mathbf{H}\mathbb{E}[(\mathbf{n}(t) - \mathbb{E}[\mathbf{n}(t)])(\mathbf{n}(t) - \mathbb{E}[\mathbf{n}(t)]^T]\mathbf{H}^\top + \mathbb{E}[\nu(t)^2] \tag{84d}$$

$$= \mathbf{H}\mathbf{P}(t)\mathbf{H}^\top + \sigma_m \tag{84e}$$

The two cross terms $\mathbb{E}[\nu(t_1)(\mathbf{n} - \mathbb{E}[\mathbf{n}])^T\mathbf{H}^T]$ and $\mathbb{E}[\mathbf{H}(\mathbf{n} - \mathbb{E}[\mathbf{n}])\nu(t_1)^T]$ are zero since $\nu$ is independent of $\mathbf{n}$ and $\mathbb{E}[\nu_m] = 0$. Our derivation is equivalent to marginalization over the predicted normal prior of the ensemble state $\mathbb{P}(\mathbf{n}(t)|\mathcal{Y}_{t-1})$ at the time of the measurement except that the prior distribution could be any probability distribution with some mean and variance. *Equation 84a* is the classical KF variance prediction of a signal. The first term in *Equation 84a*, describes the variance from stochastic gating and that the ensemble state is hidden. Notably, by Methods *Equation 30* we realize that $\text{var}(y(t))$ contains information about $\mathbf{T}$ and $\mathbf{n}(t-1)$, which we can exploit with the KF framework.

#### A generalized Kalman filter with state-dependent open-channel noise

In addition to the standard KF with only additive noise (*Moffatt, 2007*; *Anderson and Moore, 2012*; *Chen, 2003*), fluctuations arising from the single-channel gating lead to a second white-noise term $\nu_{op}n_4(t)$, causing state-dependency of our noise model. The output model is then

$$y(t) = \mathbf{H}\mathbf{n}(t) + \nu_m(t) + \nu_{op}(t) \Leftrightarrow y \sim p(y|\mathbf{n}) = \text{normal}(y|\mathbf{H}\mathbf{n}(t), \sigma_m^2 + n_4(t)\sigma_{op}^2) \tag{85}$$

The second noise term $\nu_{op}$ is defined in terms of the first two moments $\mathbb{E}(\nu_{op}) = 0$ and therefore $\text{var}(\nu_{op}) = \mathbb{E}(\nu_{op}^2) = \sigma_{op}^2\mathbf{n}_4(t)$. To the best of our knowledge such a state-dependent noise makes the following integration intractable

$$\mathbb{P}(y(t)) = \int \text{normal}(y|\mathbf{H}\mathbf{n}, \sigma_m^2 + n_4\sigma_{op}^2)\,\text{normal}(\mathbf{n}|\bar{\mathbf{n}}(t), \mathbf{P}(t))\,\mathrm{d}n \tag{86a}$$

$$= \frac{1}{const}\int \exp\left(\frac{(y - \mathbf{H}\mathbf{n})^2}{2(\sigma_m^2 + n_4\sigma_{op}^2)}\right) \exp\left(\frac{1}{2}(\mathbf{n} - \bar{\mathbf{n}}(t))\mathbf{P}^{-1}(\mathbf{n} - \bar{\mathbf{n}}(t))^\top\right)\,\mathrm{d}n \tag{86b}$$

When assuming that the relative fluctuations of $\mathbf{n}(t)$ are on average small then $n_4$ in the denominator is close to $\mathbb{E}(n_4)$ of the state. Thus, the incremental likelihood can be written as in the standard KF, with the only difference that the measurement noise is the sum of two components.

$$y(t) \sim \text{normal}(\mathbf{H}\bar{\mathbf{n}}(t), \sigma_m^2 + \sigma_{op}^2\bar{n}_4(t) + \mathbf{H}\mathbf{P}\mathbf{H}^\top) \tag{87}$$

To see that this approximation of the variance is correct, we apply the law of total variance decomposition *Weiss, 2005*.

$$\text{var}(y(t)) = \mathbb{E}[\text{var}[y(t)|\mathbf{n}(t)]] + \text{var}[\mathbb{E}[y(t)|\mathbf{n}(t)]] \tag{88a}$$

$$= \mathbb{E}[\Sigma + \sigma_{\text{op}}^2 n_4(t)] + \text{var}[\mathbf{Hn}(t)] \tag{88b}$$

$$= \sigma_m^2 + \sigma_{\text{op}}^2 \mathbb{E}[n_4(t)] + \mathbf{H}\mathbf{P}(t)\mathbf{H}^\top \tag{88c}$$

The terms $\mathbf{H}\mathbf{P}(t)\mathbf{H}^\top + \sigma_m^2$ are the standard output covariance matrix. Again $\mathbf{P}(t)$ contains information about $\mathbf{T}$, $\mathbf{n}(t-1)$ while the additional variance term includes information about the current $\mathbf{n}(t)$. The information contained in the noise influences likelihood in two ways. By the variance or covariance of the current $\mathbf{y}(t)$ but also for $\mathbf{y}(t+1)$ in correction step by the Kalman gain $\mathbf{K}_{\text{Kal}}$ matrix defined in the next section.

# Appendix 7

## Error induced for the RE or KF approach by analog filtering of PC data

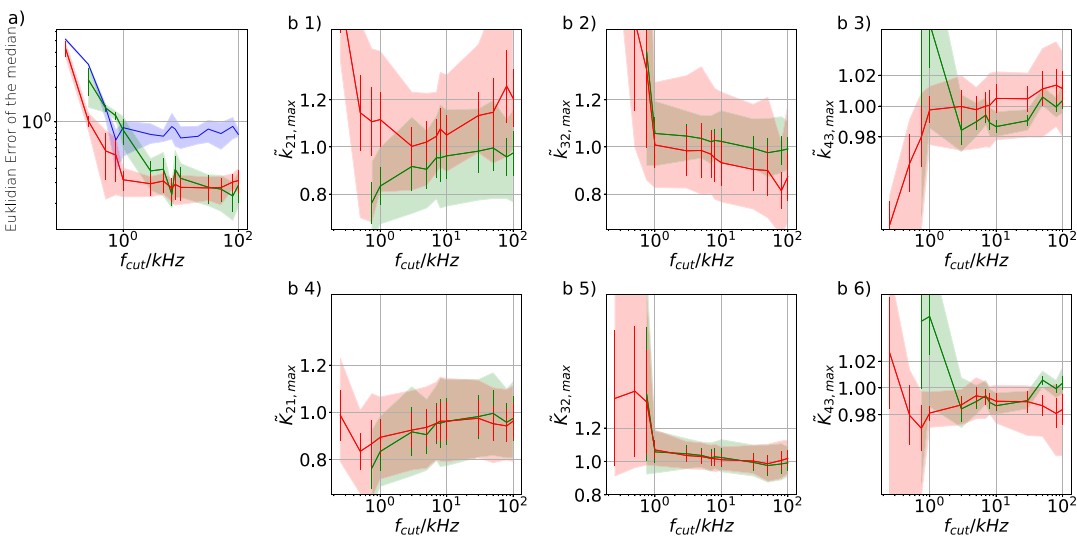

**Appendix 7—figure 1.** Influence of analog filtering on the accuracy and precision of the maximum of the posterior from PC data We chose for the analog signal $\sigma_{\mathrm{exp}}/i = 10$, $\sigma_{\mathrm{op}}/i = 0.1$, thus a strong background noise. For the ensemble size, we chose $N_{\mathrm{ch}} = 4 \cdot 10^3$ such that the likelihood dominates the uniform prior. (**a**) Estimate of the distribution of the accuracy (mean Euclidean error of the median of the posterior) vs. the cut-off frequency of a 4th-order Bessel filter scaled to the channel time scale. The solid line is the mean median of 5 data sets of the respective RE posterior (blue) and KF posterior (green). The green shaded area indicates the 0.6 quantile (ranging from 0.2 till 0.8) demonstrating the distribution of the error of the median of the posterior due to the randomness of data.(**b** 1–3) Accuracy and precision of the maxima of the posterior $\tilde{k}_{ij,max}$ of the maxima of the posterior rates vs. the cut-off frequency of a Bessel filter. The shaded areas indicate the 0.6 quantiles (ranging from 0.2 till 0.8) due to the variability from data set do data set while the error bars show the standard error of the mean. (**b** 4–6) Accuracy and precision of the maxima of the posterior $\tilde{K}_{ij,max}$ of the maxima of the posterior of the corresponding equilibria vs. the cut-off frequency of a Bessel filter.

In order to mimic an analog signal before the analog-to-digital conversion we simulated 5 different data sets of 100 kHz signals which were then filtered by a digital fourth-order Bessel filter. The activation curves were then analyzed with the Bayesian filter at 125 Hz and the deactivation curves at sampling rates between $166 - 500$ Hz. Operating the Bayesian filter at a lower frequency is necessary because due to applying the Bessel filter the former white noise of the signal obtained additional time correlations. Thus, an all data points fit would immediately violate the white noise assumption of *Equation 4* which we restore by analyzing at a much lower frequency. We then let the time scales of the induced time correlations become larger and larger by decreasing $f_{cut}$. We show (*Appendix 7—figure 1a*) the results of the Euclidean Error for 3 different cases for PC data. The RE approach (blue) and the Bayesian filter (red) share the same analyzing frequency. In contrast, an increased $f_{\mathrm{ana}} = 1660 - 5000$ Hz (green) is used for the Bayesian filter. Both algorithms (blue, red) with the same $f_{\mathrm{ana}}$ show a similar rather constant region separated by an offset. Similar, to cPCF data the KF is more robust. The Bayesian filter with $f_{\mathrm{ana}} = 1660 - 5000$ Hz (green) does not show this constant region but outperforms the Bayesian filter with the lower $f_{\mathrm{ana}}$ if the recording system is operated with minimal analog filtering. This becomes even more apparent when we consider the single parameter deviation vs. $f_{cut}$ (*Appendix 7—figure 1b(1-6)*). Note, the strong dependence of the critical $f_{cut}$ at which the performance of all algorithms becomes suddenly worse scales with $f_{\mathrm{ana}}$. Additionally, it is crucial, independently of the algorithm, that $f_{\mathrm{cut}} \gg f_{\mathrm{ana}}$. In (*Appendix 7—figure 1b(1-6)*) we demonstrate the distribution of the errors on the single parameter level and compare only the KF for different $f_{\mathrm{ana}}$. Similar to the findings in the main text, on the one hand, the Euclidean error (red) shows some robustness against $f_{\mathrm{cut}}$. On the other hand, the influence of $f_{\mathrm{cut}}$ on the individual parameter level sets in immediately and is complex. The best results are achieved with minimal analog filtering and a high $f_{\mathrm{ana}}$.

## Appendix 8

### Details of the specified two methods to count the probability mass needed to include the true value

In **Figure 4c** we compare the true against the expected success probabilities of finding the complete true rate matrix within an $n$-dimensional HDCV for different expected success probabilities of a perfect model. By 'expected' and 'perfect' we refer to a fictitious ideal algorithm which exactly models exhaustively all details of the true process, including all measurement details. The first way assumes a multivariate normal posterior distribution with $\mathbb{P}(\boldsymbol{\theta}|y)_{\text{post}} \approx \text{normal}(\boldsymbol{\theta}|E[\boldsymbol{\theta}], \boldsymbol{\Sigma})$. Then the ellipsoid of a constant probability density is exactly the surface of a HDCV of given probability mass $P$. For a two-dimensional representation see **Figure 4d** below the diagonal. In one dimension, the ellipsoid consists of the two points with a distance $d_{\text{Mah}} = \sqrt{(\theta - E[\theta])^2/\sigma}$ from the mean (inset **Figure 5a**). In general, the $n$-dimensional ellipsoids around the mean of a multivariate normal distribution can be described by points which have the same Mahalanobis distance $d_{\text{Mah}}$ from the mean.

$$d_{\text{Mah}}(\boldsymbol{\theta}) = \sqrt{(\boldsymbol{\theta} - E[\boldsymbol{\theta}])\Sigma^{-1}(\boldsymbol{\theta} - E[\boldsymbol{\theta}])} \tag{89}$$

For a multivariate standard normal distribution without correlation, the Mahalanobis distance becomes the Euclidean distance $\sqrt{(\boldsymbol{\theta} - E[\boldsymbol{\theta}])^T(\boldsymbol{\theta} - E[\boldsymbol{\theta}])}$. Rewriting the multivariate normal distribution in terms of the Mahalanobis distance

$$\text{normal}(\boldsymbol{\theta}) = \frac{1}{\sqrt{\det(2\pi\Sigma)}} \exp[-0.5 d(\boldsymbol{\theta})^2_{\text{Mah}}] \tag{90}$$

reveals that all points $\boldsymbol{\theta}$ with $d_{\text{Mah}} = const$ have the same probability density. The random variable $d^2_{\text{Mah}}$ is $\chi$-square distributed. Thus the $\chi$-square distribution (**Figure 5a**) is the probability density of drawing an $\boldsymbol{\theta}$ in $n$ dimensions and finding it on the ellipsoid's $n - 1$-dimensional surface, at a distance $d_{\text{Mah}}$ from the mean. This allows us to use the cumulative $\chi^2$-square distribution function to calculate the probability mass inside the ellipsoid which is the desired HDCV. By evaluating $d_{\text{Mah}} = \sqrt{(\boldsymbol{\theta}_{\text{true}} - E[\boldsymbol{\theta}])\Sigma^{-1}(\boldsymbol{\theta}_{\text{true}} - E[\boldsymbol{\theta}])}$ and $\chi^2_{\text{cdf}}(d^2_{\text{Mah}})$ we specify how much volume, in units of probability mass, has to be counted until the volume includes the true value. Note, that with increasing dimensionality (**Figure 5a**) the probability mass is shifted away from the mode of the probability density. For general probability densities, this is also true in higher dimensions where one only rarely finds the true value within a small distance to the maximum of the probability density. Note, that there is no mathematical theorem for singular models (**Watanabe, 2007**) saying that the posterior approaches asymptotically a multivariate normal with increasing data quality/quantity. Consequently, the underlying assumption that the posterior distribution is multivariate normal, is situation dependent valid or highly questionable. Thus the displayed method should be validated additionally in an independent way. To determine the probability mass needed to include the true value into the HDCV for such a posterior, we can also use a histogram-based method. One starts by constructing an $n$-dimensional histogram from the samples of the posterior and by initializing a global variable with zero. Then we start counting from the bin with most counts and check whether the true value falls inside this bin. If not, we add the probability mass inside that bin to the global variable. Repeating this for next lower bins eventually leads to the detection of the bin with the true rate matrix inside. On detection, the global variable contains the sought-after value $P$. While this procedure does not depend on a multivariate normal assumption, it is prone to errors due to the discrete bins and the finite samples from the posterior.

Nevertheless, both procedures show (**Figure 5b**) good agreement when plotting the volume/probability mass needed to include the true parameter value vs. $N_{\text{ch}}$. Again $N_{\text{ch}} \approx 200$ seems to be a reasonable data quality to trust the statistics of the posterior using the KF. In contrast, the posterior of the RE approach never includes the true values with a reasonable HDCV. Note, that a probability mass of $\approx 1$ to reach the true value means that qualitatively speaking the estimate of the RE approach is infinitely far away from the true value in terms of the Mahalanobis distance. Further, due to the finite sampling the method of histogram counting is not qualified for the largest HDCVs approaching $P \approx 1$. The two developed methods contrast the Euclidean distance from the true values to the maximum mode, median or even the mean of the distribution against all higher moments of the distribution. Thus it tests the overall shape of the posterior.

## Appendix 9

### Uncertainty quantification for the 5-state and 6-state model with cPCF data

The effect of the second observable on the single parameter level for the 5-state model can be seen in *Appendix 9—figure 1*. The biased estimates and the unidentified parameters of the RE approach for PC data (*Figure 9*) are eliminated by the fluorescence data.

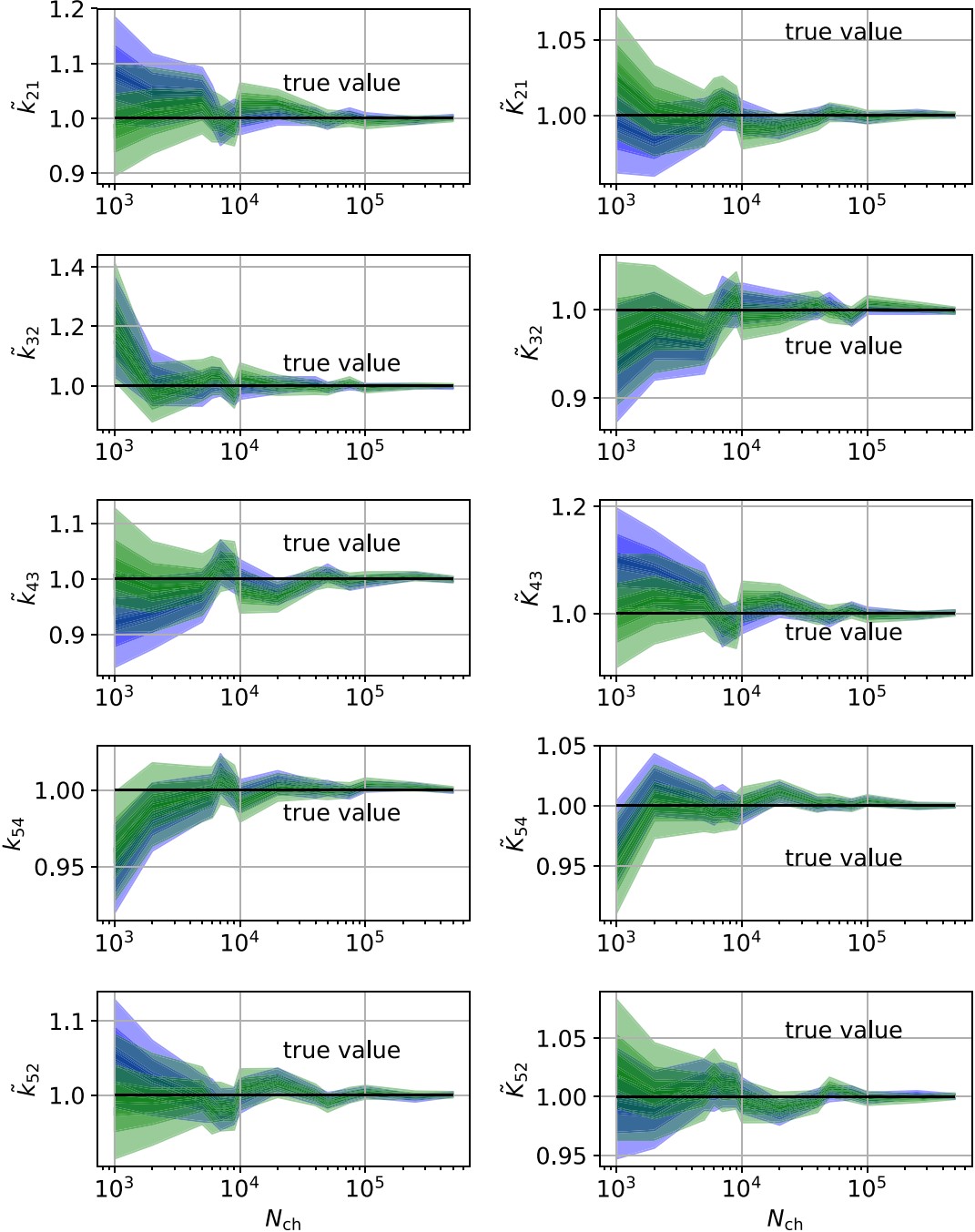

**Appendix 9—figure 1.** HDCIs vs $N_{ch}$ for the 5-state-2-open-states model for cPCF data. The single parameter level corresponds to the Euclidean error of *Figure 7d*. Compared to the main text we switched rows with columns of the panels, the first column represents now the rates and the second column the equilibrium constants. The prior that enforces microscopic-reversibility is identical to the prior mentioned in the caption of *Figure 7*.

Even the over-confidence problem seems to be decreased (compare with *Figures 4b1–6*). In contrast, the HDCIs of the 6-states-1-open-state model for cPCF data (*Appendix 9—figure 2*) display again a much increased over-confidence problem of the rate equation approach. This indicates that not simply counting of states but the actual topology of the kinetic scheme and the structure of the observation matrix **H** determines the scale of the over-confidence problem. Apparently, the more information comes from the fluorescence data relative to the information from the current data the less grave is the over-confidence problem and the less different are the Euclidean errors.

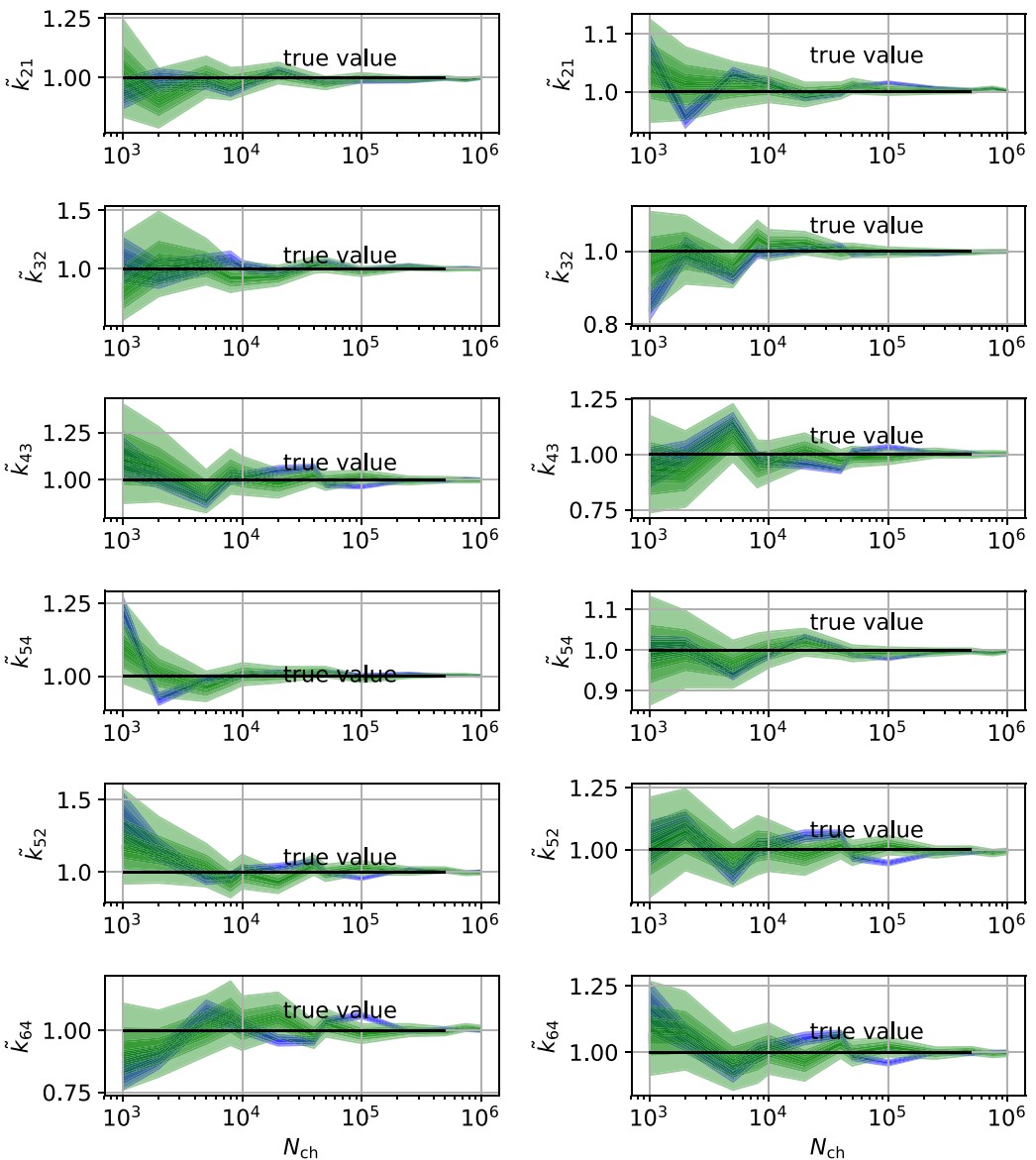

**Appendix 9—figure 2.** HDCIs vs $N_{ch}$ for the 6-states-1-open-state model for cPCF data. The single parameter level corresponds to the Euclidean error of *Figure 7e*. Compared to the main text we switched rows with columns of the panels, the first column represents now the rates and the second column the equilibrium constants. The prior that enforces microscopic-reversibility is identical to the prior mentioned in the *Figure 7*.

