## [Editor Report]

The authors develop a Bayesian approach to modeling signals arising from ensembles of ion channels that can incorporate multiple simultaneously recorded signals such as fluorescence and ionic current. For simulated data from a simple ion channel model where ligand binding drives pore opening, they show that their approach enhances parameter identifiability and/or estimates of parameter uncertainty over more traditional approaches. The developed approach provides a valuable tool for modeling macroscopic time series data including data with multiple observation channels.

---

## [Decision Letter]

**Decision letter after peer review:**

Thank you for submitting your article "Bayesian selection of Hidden Markov models for multi-dimensional ion channel data" for consideration by *eLife*. Your article has been reviewed by 3 peer reviewers, including Marcel P Goldschen-Ohm as the Reviewing Editor and Reviewer #1, and the evaluation has been overseen by Richard Aldrich as the Senior Editor. The following individuals involved in review of your submission have agreed to reveal their identity: Lorin S Milescu (Reviewer #2); Colin Kinz-Thompson (Reviewer #3).

The reviewers have discussed the reviews with one another and the Reviewing Editor has drafted this decision to help you prepare a revised submission.

We would like to draw your attention to changes in our revision policy that we have made in response to COVID-19 (https://elifesciences.org/articles/57162). Specifically, when editors judge that a submitted work as a whole belongs in *eLife* but that some conclusions require additional discussion or a modest amount of additional analysis, as they do with your paper, we are asking that the manuscript be revised to address the points raised by the reviewers.

Our expectation is that the authors will eventually carry out the additional analyses and report on how they affect the relevant conclusions either in a preprint on bioRxiv or medRxiv, or if appropriate, as a Research Advance in *eLife*, either of which would be linked to the original paper.

Summary:

Extracting ion channel kinetic models from experimental data is an important and perennial problem. Much work has been done over the years by different groups, with theoretical frameworks and computational algorithms developed for specific combinations of data and experimental paradigms, from single channels to real-time approaches in live neurons. At one extreme of the data spectrum, single channel currents are traditionally analyzed by maximum likelihood fitting of dwell time probability distributions; at the other extreme, macroscopic currents are typically analyzed by fitting the average current and other extracted features, such as activation curves. Robust analysis packages exist (e.g., HJCFIT, QuB), and they have been put to good use in the literature.

Münch et al. focus here on several areas that need improvement: dealing with macroscopic recordings containing relatively low numbers of channels (i.e., hundreds to tens of thousands), combining multiple types of data (e.g., electrical and optical signals), incorporating prior information, and selecting models. The main idea is to approach the data with a predictor-corrector type of algorithm, implemented via a Kalman filter that approximates the discrete-state process (a meta-Markov model of the ensemble of active channels in the preparation) with a continuous-state process that can be handled efficiently within a Bayesian estimation framework, which is also used for parameter estimation and model selection.

With this approach, one doesn't fit the macroscopic current against a predicted deterministic curve, but rather infers – point by point – the ensemble state trajectory given the data and a set of parameters, themselves treated as random variables. This approach, which originated in the signal processing literature as the Forward-Backward procedure (and the related Baum-Welch algorithm), has been applied since the early 90s to single channel recordings (e.g., Chung et al., 1990), and later has been extended to macroscopic data, in a breakthrough study by Moffatt (2007). In this respect, the study by Münch et al. is not necessarily a conceptual leap forward. However, their work strengthens the existing mathematical formalism of state inference for macroscopic ion channel data, and embeds it very nicely in a rigorous Bayesian estimation framework.

The main results are very convincing: basically, model parameters can be estimated with greater precision – as much as an order of magnitude better – relative to the traditional approach where the macroscopic data are treated as noisy but deterministic (but see comments below). Estimated uncertainty can be further improved by incorporating prior information on parameters (e.g., diffusion limits), and by including other types of data, such as fluorescence.

The manuscript is well written and overall clear, and the mathematical treatment is a rigorous tour-de-force. However, the reviewers raised a number points that need further clarification, better discussion or amendment. These concerns are likely to be addressable largely by changes to the main text and software documentation along with some additional analyses. Once addressed, a revised version of this manuscript would likely be suitable for publication in *eLife*. That said, the study is very nice and ambitious, but clarity is a bit impaired by dealing with perhaps too many issues. The state inference and the bayesian model selection are very important but completely different issues. The authors should consider whether they may be better treated separately, or even perhaps in a more specialized journal where they can be developed in more detail.

Essential revisions:

1. Tutorial-style computational examples must be provided, along with well commented/documented code. The interested readers should be able to implement the method described here in their own code/program. The supplied code is not well documented, and it is unclear whether it is applicable beyond the specific models examined in the paper. It was supplied as.txt files, but looks like C code, and is unlikely to be easily adaptable by others to their own models.

2. The authors should clearly discuss the types of data and experimental paradigms that can be optimally handled by this approach, and they must explain when and where it fails or cannot be applied or becomes inefficient in comparison with other methods. One must be aware that ion channel data are very often subject to noise and artifacts that alter the structure of microscopic fluctuations. It seems that the state inference algorithm would work optimally with low noise, stable, patch-clamp recordings (and matching fluorescence recordings) in heterologous expression systems (e.g., HEK293 cells), where the currents are relatively small, and only the channel of interest is expressed (macropatches?). In contrast, it may not be effective with large currents that are recorded with low gain, are subject to finite series resistance, limited rise time, restricted bandwidth, colored noise, contaminated by other currents that are (partially) eliminated with the P/n protocol with the side effect of altering the noise structure, power line 50/60 Hz noise, baseline fluctuations, etc. This basically excludes some types of experimental data and experimental paradigms, such as recordings from neurons in brain slices or in vivo, oocytes, etc. Of course, artifacts can affect all estimation algorithms, but approaches based on fitting the predicted average current have the obvious benefit of averaging out some of these artifacts.

The discussion in the manuscript is insufficient in this regard and must be expanded. The method should be tested under non-ideal but commonly occurring conditions, such as limited bandwidth and in the presence of contaminating noise. For example, compare estimates obtained without filtering with estimates obtained with 2, 3 times over-filtering, with and without large measurement noise added (whole cell recordings with low-gain feedback resistors and series resistance compensation are quite noisy), with and without 50/60 Hz interference. How does the algorithm deal with limited bandwidth that distorts the noise spectrum? How are the estimated parameters affected? The reader will have to get sense of how sensitive is this method to artifacts. Also, fluorescence data in particular is usually of much lower temporal resolution than current measurements. How does this impact its benefit to parameter estimation?

3. Even more emphasis on the approximation of n(t) as being distributed according to a multivariate normal, and thus being continuous, should be placed in the main text. It seems that this may limit the applicability of the method to data with > ~100s of channels; although the point is not investigated. In Figure 3, the method is only benchmarked to a lower limit of ~500 channels. As shown in Milescu et al., 2005, fitting macroscopic currents is asymptotically unbiased. In other words, the estimates are accurate, unless the number of channels is small (tens or hundreds), in which case the multinomial distribution is not very well approximated by a Gaussian. What about the predictor-corrector method? How accurate are the estimates, particularly at low channel counts (10 or 100)? Since the Kalman filter also uses a Gaussian to approximate the multinomial distribution of state fluctuations, I would also expect asymptotic accuracy. Parameter accuracy should be tested, not just precision.

4. Achieving the ability to rigorously perform model selection is very impressive aspect of this work and a large contribution to the field. However, the manuscript offers too many solutions to performing that model selection itself along with a long discussion of the field (for instance, line 376-395 could be completely cut). Since probabilistic model selection is an entire area of study by itself, the authors do not need to present underdeveloped investigations of each of them in a paper on modeling channel data (e.g., of course WAIC outperforms AIC. Why not cover BIC and WBIC?). The authors should pick one, and maybe write a second paper on the others instead of presenting non-rigorous comparisons (e.g., one kinetic scheme and set of parameters). As a side note, it is strange that the authors did not consider obtaining evidence or Bayes factors to directly perform Bayesian model selection – for instance, they could have used thermodynamic integration since they used MC to obtain posteriors anyway (c.f., Computing Bayes Factors Using Thermodynamic Integration by Lartillot and Philippe, Systematic Biology, 2006, 55(2), 195-207. DOI: 10.1080/10635150500433722)

5. Regarding model selection for the PC data in Figure 7, why does the RE model with BC appear to provide a visually better identification of the true model than the KF model with BC if the KF model does a better job at estimating the parameters and setting non true rates to zero? Doesn't this suggest that RE with cross validation is better than the proposed KF approach in regards to model selection? In terms of parameter estimates (i.e. as shown in Figure 3), how does RE + BC stack up?

6. A better comparison with alternative parameter estimation approaches is necessary. First of all, explain more clearly what is different from the predictor-corrector formalism originally proposed by Moffatt (2007). The manuscript mentions that it expands on that, but exactly how? If it is only an incremental improvement, of interest to a very limited audience, a specialized, technical journal, is more appropriate.

Second, the method proposed by Celentano and Hawkes, 2004, is not a predictor-corrector type but it utilizes the full covariance matrix between data values at different time points. It seems that the covariance matrix approach uses all the information contained in the macroscopic data and should be on par with the state inference approach. However, this method is only briefly mentioned here and then it's quickly dismissed as "impractical". We all agree that it's a slower computation than, say, fitting exponentials, but so is the Kalman filter. Where do we draw the line of impracticability? Computational speed should be balanced with computational simplicity, estimation accuracy, and parameter and model identifiability. Moreover, that method was published in 2004, and the computational costs reported there should be projected to present day computational power. We are not saying that the authors should code the C&H procedure and run it here, but should at least give it credit and discuss its potential against the KF method.

The only comparison provided in the manuscript is with the "rate equation" approach, by which the authors understand the family of methods that fit the data against a predicted average trajectory. In principle, this comparison is sufficient, but there are some issues with the way it's done.

Table 3 compares different features of their state inference algorithm and the "rate equation fitting", referencing Milescu et al., 2005. However, there seems to be a misunderstanding: the algorithm presented in that paper does in fact predict and use not only the average but also – optionally – the variance of the current, as contributed by stochastic state fluctuations and measurement noise. These quantities are predicted at any point in time as a function of the initial state, which is calculated from the experimental conditions. In contrast, the KF calculates the average and variance at one point in time as a projection of the average and variance at the previous point. However, both methods (can) compare the data value against a predicted probability distribution. The Kalman filter can produce more precise estimates but presumably with the cost of more complex and slower computation, and increased sensitivity to data artifacts.

Figure 3 is very informative in this sense, showing that estimates obtained with the state inference (KF) algorithm are about 10 times more precise that those obtained with the "rate equation" approach. However, for this test, the "rate equation" method was allowed to use only the average, not the variance.

Considering this, the comparison made in Figure 3 should be redone against a "rate equation" method that utilizes not only the expected average but also the expected variance to fit the data, as in Milescu et al., 2005. Calculating this variance is trivial and the authors should be able to implement it easily (reviewers are happy to provide feedback). The comparison should include calculation times, as well as convergence.

7. The manuscript nicely points out that a "rate equation" approach would need 10 times more channels (N) to attain the same parameter precision as with the Kalman filter, when the number of channels is in the approximate range of 10^2 … 10^4. With larger N, the two methods become comparable in this respect.

This is very important, because it means that estimate precision increases with N, regardless of the method, which also means that one should try to optimize the experimental approach to maximize the number of channels in the preparation. However, it should be pointed out that one could simply repeat the recording protocol 10 times (in the same cell or across cells) to accumulate 10 times more channels, and then use a "rate equation" algorithm to obtain estimates that are just as good. Presumably, the "rate equation" calculation is significantly faster than the Kalman filter (particularly when one fits "features", such as activation curves), and repeating a recording may only add seconds or minutes of experiment time, compared to a comprehensive data analysis that likely involves hours and perhaps days. Although obvious, this point can be easily missed by the casual reader and so it would be useful to be mentioned in the manuscript.

8. A misunderstanding is that a current normalization is mandatory with "rate equation" algorithms. This is really not the case, as shown in Milescu et al., 2005, where it is demonstrated clearly that one can explicitly use channel count and unitary current to predict the observed macroscopic data. Consequently, these quantities can also be estimated, but state variance must be included in the calculation. Without variance, one can only estimate the product i x N, where i is unitary current and N is channel count. This should be clarified in the manuscript: any method that uses variance can be used to estimate i and N, not just the Kalman filter. In fact, the non-stationary noise analysis does exactly that: a model-blind estimation of N and i from non-equilibrium data. Also, one should be realistic here: in some circumstances it is far more efficient to fit data "features", such as the activation curve, in which case the current needs to be normalized.

9. It's great that the authors develop a rigorous Bayesian formalism here, but it would be a good idea to explain – even briefly – how to implement a (presumably simpler) maximum likelihood version that uses the Kalman filter. This should satisfy those readers who are less interested in the Bayesian approach and will also be suitable for situations when no prior information is available.

10. The Bayesian formalism is not the only way of incorporating prior knowledge into an estimation algorithm. In fact, the reader may have more practical and pressing problems than guessing what the parameter prior distribution should be, whether uniform or Gaussian or other. More likely one would want to enforce a certain KD, microscopic (i)reversibility, an (in)equality relationship between parameters, a minimum or maximum rate constant value, or complex model properties and behaviors, such as maximum Popen or half-activation voltage. A comprehensive framework for handling these situations via parameter constraints (linear or non-linear) and cost function penalty has been recently published (Salari et al/Navarro et al., 2018). Obviously, the Bayesian approach has merit, but the authors should discuss how it can better handle the types of practical problems presented in those papers, if it is to be considered an advance in the field, or at least a usable alternative.

11. The methods section should include information concerning the parameter initialization choices, HMC parameters (e.g. number of steps) and any burn-in period used in the analyses used in Figures 3-6. For example, how is convergence established? How many iterations does it take to reach convergence? How long does it take to run? How does it scale with the data length, channel count, and model state count? How long does it take to optimize a large model (e.g., 10 or 20 states)? Provide some comparison with the "rate equation method".

12. In the section on priors, the entire part concerning the use of a β distribution should be removed or replaced, because it is a probabilistic misrepresentation of the actual prior information that the authors claim to have in the manuscript text. The max-entropy prior derived for the situation described in the text (i.e., an unknown magnitude where you don't know any moments but do have upper and lower bounds; the latter could be from the length from the experiment) is actually P(x) = (ln(x_{max}) – ln(x_{min}))^{-1} * x^{-1}. Reviewers are happy to discuss more with the authors.

13. Here and there, the manuscript somehow gives the impression that existing algorithms that extract kinetic parameters by fitting the average macroscopic current ("fitting rate equations") are less "correct", or ignorant of the true mathematical description of the data. This is not the case. Published algorithms often clearly state what data they apply to, what their limitations are, and what approximations were made, and thus they are correct within that defined context and are meant to be more effective than alternatives. Some quick editing throughout the manuscript should eliminate this impression.

14. The manuscript refers to the method where the data are fitted against a predicted current as "rate equations". However, it is not completely clear what that means. The rate equation is something intrinsic to the model, not a feature of any algorithm. An alternative terminology must be found. Perhaps different algorithms could be classified based on what statistical properties are used and how. E.g., average (+variance) predicted from the starting probabilities (Milescu et al., 2005), full covariance (Celentano and Hawkes, 2004), point-by-point predictor-corrector (Moffatt, 2007).

15. The manuscript needs line editing and proofreading (e.g., on line 494, "Roa" should be "Rao"; missing an equals sign in equation 13). Additionally, in many paragraphs, several of the sentences are tangential and distract from communicating the message of the paper (e.g., line 55). Removing them will help to streamline the text, which is quite long.

[Editors' note: further revisions were suggested prior to acceptance, as described below.]

Thank you for submitting your article "Bayesian inference of kinetic schemes for ion channels by Kalman filtering" for consideration by *eLife*. Your article has been reviewed by 3 peer reviewers, including Marcel P Goldschen-Ohm as the Reviewing Editor and Reviewer #1, and the evaluation has been overseen by Richard Aldrich as the Senior Editor. The following individuals involved in review of your submission have agreed to reveal their identity: Colin Kinz-Thompson (Reviewer #3); Lorin Milescu (Reviewer #4).

Essential revisions:

*Reviewer #1 (Recommendations for the authors):*

The authors develop a Bayesian approach to modeling macroscopic signals arising from ensembles of individual units described by a Markov process, such as a collection of ion channels. Their approach utilizes a Kalman filter to account for temporal correlations in the bulk signal. For simulated data from a simple ion channel model where ligand binding drives pore opening, they show that their approach enhances parameter identifiability and/or estimates of parameter uncertainty over more traditional approaches. Furthermore, simultaneous measurement of both binding and pore gating signals further increases parameter identifiability. The developed approach provides a valuable tool for modeling macroscopic time series data with multiple observation channels.

The authors have spent considerable effort to address the previous reviewer comments, and I applaud them for the breadth and depth of their current analysis.

1. The figure caption titles tend to say what analysis or comparison is being presented, but not what the take home message of the figure is. I suggest changing them to emphasize the latter. This will especially help non-experts to understand what the figures are trying to convey to them.

2. I very much appreciate the GitHub code and examples for running your software. However, I feel that a hand-holding step-by-step explicit example of running a new model on new data is likely necessary for many to be able to utilize your software. Much more hand-holding than the current instructions on the GitHub repo.

3. Figure captions sometimes do not explain enough of what is shown. I appreciate that many of these details are in the main text, but data that is displayed in the figure and not concretely described in the caption can makes the figures hard to follow. e.g. Figure 4a – "With lighter green we indicate the Euclidean error of patch-clamp data set." But what data set do the other colors reflect? It is not stated in the caption. Again, I realize this is described in the main text, but it also needs to be defined in the caption where the data is presented. Figure 4d – Please spell out what "both algorithms" intends. Also, a suggestion: instead of having to refer to the caption for the color codes (i.e. RE vs. KF, etc.) it would speed figure interpretation to have a legend in the figures themselves. Few other examples: Box 1. Figure 2. – Please define the solid vs. dashed lines in the caption. Figure 3c – Please define the solid vs. dashed lines in the caption. Figure 12 – "We simulated 5 different 100kHz signals." What kind of signals? Fluorescence I assume, but this needs to be explicitly defined. I'd check all figures for similar.

*Reviewer #3 (Recommendations for the authors):*

In this revised manuscript, the Münch et al. have addressed all of my original concerns. It is significantly revised, though, and includes many new investigations of the algorithm's performance. Overall, the narrative of this manuscript is now to introduce an approximation to the solution for a Bayesian Kalman Filter, and then spend time demonstrating that this approximation is reasonable and even better than previous methods. In my opinion, they successfully do this, although, as they mention in their comments, their manuscript is very long.

I am not 100% certain, but the approximation that the author's make seems to be equivalent (or at least similar to) an approximation of the chemical master equation using just the 1st and 2nd moments, which is just the Fokker-Planck equation. The authors should discuss any connection to this approximation, as there is a great deal of literature on this topic (e.g., see van Kampen's book).

In Figures 3A and 4D, it is unclear to me what is plotted for the RE and classical Kalman filter (i.e., how is there a posterior if they are not Bayesian methods)? Perhaps it is buried in the methods or appendices, but, if so, it needs to at least be clarified in the figure captions.

The Bayesian statistical tests devised to determine success (e.g., on pgs. 12-13) seem a little ad hoc, but are technically acceptable. I do not see a need for additional metrics.

Line 939: Equation 61 is absolutely not "close to uniform distribution". The α and β parameters of 100.01 are much larger than 1. It is incredibly peaked around 0.5. Perhaps this is a typo?

Line 942: The allowed small breaking of microscopic reversibility in the prior is an interesting idea that I wish the authors would expound upon more.

Line 712: The authors state that the simulation 'code will be shared up-on request'. They should include it with their github pages tutorials for running the examples in case others wish to check their work and/or use it. There is no reason to withhold it.

Line 707: 'Perspectively' is not a commonly used word.

*Reviewer #4 (Recommendations for the authors):*

The authors have addressed all my comments and suggestions. The manuscript is nice and extremely comprehensive, and should advance the field.

Nevertheless, the manuscript is also very long (but justifiably so), and certain statements could be a little clearer. Most of these statements refer to the comparison with the so-called rate equation (RE) methods, with which I'm more familiar. For example:

Abstract: "Furthermore, the Bayesian filter delivers unbiased estimates for a wider range of data quality and identifies parameters which the rate equation approach does not identify."

The first part of this statement is not quite true, as Figure 4 shows clearly that the Bayesian estimates are biased (and the authors acknowledge this elsewhere in the manuscript). If they are biased in one "range of data quality", that probably means they are always biased, just to different degrees. This is not surprising, because the Kalman filter is a continuous state approximation to a discrete state process, and the overall estimation algorithm makes a number of approximations to intractable probability distributions. It would definitely be correct to say that the estimates are very good, but not unbiased.

Second, this statement is also ambiguous. Are you referring to the theoretical non-identifiability caused by having more parameters in the model than what the combination of estimator+experimental protocol can extract from data? In this case, it's not a matter of certain parameters that cannot be identified, but a matter of how many can be identified uniquely. The more information is extracted from the data, the more unique parameters can be identified, so the Kalman filter should do better. Or, alternatively, are you referring to specific parameters that are poorly identified because, for example, they refer to transitions that are rarely undertaken by the channel? In this case, it would be a matter of obtaining looser or tighter estimates with one method or another, but the parameters should be intrinsically identifiable, I imagine, regardless of the algorithm. In any case, it's not clear that the better identifiability is the result of the Bayesian side, or of the predictor-corrector state inference filter. I would guess it is the Kalman filter, but I'm not sure.

Perhaps it would be clearer if you said that the KF method produces good estimates and generally improves parameter identifiability compared to other methods, as it extracts more information from the data?

Introduction:

32: I'm not sure, but if the intention here is to cite mathematical treatments for estimation, you may add references to the "macroscopic" papers by Celentano, Milescu, Stepaniuk and perhaps a few others that use "rate equations". Also, you may cite Qin et al., 1996, as a single channel paper describing a method used in hundreds of studies.

Pg. 3:

51: I remain skeptical that it is a good idea to use "rate equations" (RE) as a term to refer to those methods that are fundamentally different from the approach described here (also see my comment to the first submission). The rate equations must always be used to predict a future state from a past or current state, by all methods, explicitly or implicitly, because REs simply describe the channel dynamics. In this very manuscript, Equation 3, central to the Kalman filter formalism, is nothing but a deterministic rate equation with a stochastic term added to approximate the stochastic evolution of an ensemble of channels. In fact, there are some old papers by Fox and some more recent by Osorio (I'm not exactly sure of the name and I don't remember the years) that discuss that approximation and its shortcomings – perhaps that is the source of bias?

Whether that prediction is then corrected, as in the Kalman filter approach, with a corrector step is irrelevant, as far as the rate equations. Furthermore, it's all a matter of degree. For example, the Milescu et al. approach, which is classified here under the RE umbrella, predicts future states as a mean + variance (or as a mean only), but only using the initial state. There is no correction at each point, as with the Kalman filter method, but there is a correction of the initial state from one iteration to the next (by the way, it's not clear if you implemented that feature here, which would make a big difference). Then, because it considers the stochastical aspect of the data, the Milescu et al. approach should also be considered a stochastic method, just one that doesn't use all the information contained in the data (and so it is fast). Imagine a situation where you ran the same stimulation protocol multiple times, but each time you record only one data point, further and further away from the beginning of the stimulation protocol. A "stochastic" algorithm applied to this type of data would be exactly as described in Milescu et al. Of course, in reality all points are recorded in sequence, but that doesn't mean that the approach is not stochastic, just that it 's simplifying and discarding some information to gain speed. All methods (such as the Kalman filter described here) make some compromises to reduce complexity and increase speed.

The bottom line is that there is the most basic approach of solving the rate equations deterministically, without considering any variance whatsoever, and then there is everything else.

58: "Thus, a time trace with a finite number of channels contains, strictly speaking, only one independent data point."

I don't understand this sentence. Could you please clarify?

74: "The KF is the minimal variance filter Anderson and Moore (2012). Thus, instead of excessive analog filtering of currents with the inevitable signal distortions Silberberg and Magleby (1993) one can apply the KF with higher analysing frequency on less filtered data."

Yes, but I'm not sure why should we use excessive filtering? Where is this idea coming from?

131: "However, even with simulated data of unrealistic high numbers of channels per patch, the KF outperforms the deterministic approach in estimating the model parameters."

It is clearly true from your figures, but please give a number as to what is unrealistic (10,000? 100,000?). Also, outperforms, but by how much? As I commented above and below, all methods tested here seem to produce good estimates under the conditions they were designed for (and even outside), and one might argue that it's not worth adding the additional computational complexity and running time for a possibly small increase in accuracy. How is one to judge that increase in accuracy?

276: Has the RE method been used with the mean + variance or mean only? Also, how was the initial probability vector calculated with the RE method? If you used the mean + variance, then (as mentioned above), you can't call it deterministic. If you used only the mean, then it is indeed a deterministic method, but it's not the approach described in Milescu et al. Please explain.

229: "added, in Equation 9 to the variance … which originates (Equation 38d) from the fact that we do not know the true system state"

Which noise are you referring to? The state noise or the measurement noise?

326: Which are the two algorithms?

278: "Both former algorithms are far away from the true parameter values with their maxima and also fail to cover the true values within reasonable tails of their posterior."

I think the authors might have gotten a little carried away when they made this statement. There is no doubt that the Kalman/Bayesian method produces more accurate estimates, but the other two methods (KF and RE) are very reasonable as well. I see estimates that are within 5, 10, or 20% from the true values, for most parameters. This is not "failure" by any stretch of the imagination. Most people would call this quite good, given the nature of the data.

294: "The small advantage of our algorithm for small … over Moffatt (2007) is due to the fact that we could apply an informative prior in the formulation of the inference problem … by taking advantage of our Bayesian filter. "

This is an interesting and important statement. I interpret it as saying that the Bayesian aspect itself makes only a small contribution to the quality of the estimates, when comparing it with the Moffatt method, which is a Kalman filter as well. The only issue with the Moffatt method is the lack of an explicit formalism for the excess state noise (which could presumably be added). It also suggests to me that any method that tries to use the noise may run into difficulties when the noise model is unknown (typical real-life scenario). The Moffatt method is confronted with unknown noise and it fails. What about when the Bayesian method is confronted with unknown noise? There are some comments in the manuscript, but nothing tangible. Could you please comment?

Figure 3: There are no data sets in the figure. What data were analyzed here?

Is there a typo regarding the colors in a? What are the red, blue, black, green symbols? Please verify.

Assuming the red is KF, there is something very curious about its estimates, which are very different in distribution from the Bayesian estimates. Could there be a coding problem? If red is actually RE, then it would make more sense. Could you explain? Is this the effect of the "excessive" open state noise? If so, I find this situation a bit unfair, because then the 2007 KF algorithm is tested with data that it is not meant to work with.

Also, I think it would be more interesting to have a comparison between the original KF and the newer Bayesian approach, so we can understand what Bayesian estimation does. In any case, I can't really interpret the data until you clarify the colors.

The legend is unclear overall.

303: What means "singular"?

324: What shaded area in Figure 1a? Perhaps you mean Figure 4a?

Figure 4: I don't see any light green.

368: "The estimated true success rate by the RE approach (blue) is ≈ 0.15 and therefore far away from the true success probability. In contrast, the posterior (green) of the true success probability of the KF resides with a large probability mass between the lower and upper limit of the success probability of an optimal algorithm (given the correct kinetic scheme)."

I'm really a bit puzzled here: yes, the KF is more accurate (given the correct kinetic scheme), but what I see in this figure is that both algorithms are biased, yet both are generally within 10% (and quite a bit better for KF) of the true values. If one would plot the log of the rates, to transform into δ Gs, the differences would be even smaller. I would bet that experimental artifacts would contort the estimates to a greater degree anyway.

It's very nice that the RE approach was embedded and tested within a Bayesian framework, but it would still be interesting to know what is the contribution of the Bayesian aspect (unless I missed this point in the manuscript).

436: "…while their error ratio (Figure 7a) seems to have no trend to diminish with increasing NCh".

This would be unexpected, because the Kalman filter will always use more information than RE. Why would the KF approach become relatively more erroneous at higher Nc? I don't see any reason. To me, an important question is when do the overall errors become dominated by external factors, such as recording artifacts, using the wrong model, etc.

511: "Thus, transferring the unusually strictly applied algebraic constraint Salari et al. (2018) of microscopic-reversibility to constraint with scalable softness we can model the lack information if microscopic-reversibility is exactly fulfilled Colquhoun et al. (2004) by the given ion channel instead of forcing it upon the model."

I'm not sure I understand what you mean here. I think it is very usual to constrain microscopic reversibility EXACTLY, whether through the SVD method (Qin et al., Plested et al., Milescu et al., Salari et al), or through some other algebraic method (Plested et al). It's not "forcing" it on the channel, but simply testing the data for that condition. Of course, one could test the condition where there is no reversibility enforced at all (i.e., it's "given by the channel"), or anything in between.

675: "With our algorithm we demonstrate (Figure 3c and 7) that the common assumption that for large ensembles of ion channels simpler deterministic modeling by RE approaches is on par with stochastic modeling, such as a KF, is wrong in terms of Euclidean error and uncertainty quantification (Figure 5a-c and Figure 6a-b)".

I find this statement a little subjective. I think anyone who cares about stochastic vs. deterministic modeling knows enough that any method that uses more information from data should produce better estimates, regardless of the number of channels. I would say that the more likely assumption is that deterministic estimators produce poor estimates with small numbers of channels, perfect estimates with infinitely many channels, and anything in between. In fact, the KF method behaves exactly the same way, just with overall higher accuracy. Looking at the tests in this manuscript, I would say that all the previous studies that modeled macroscopic data using deterministic methods are safe. They don't need to be redone. The future, of course, is a different matter.

---

## [Author Response]

Essential revisions:1. Tutorial-style computational examples must be provided, along with well commented/documented code. The interested readers should be able to implement the method described here in their own code/program. The supplied code is not well documented, and it is unclear whether it is applicable beyond the specific models examined in the paper. It was supplied as.txt files, but looks like C code, and is unlikely to be easily adaptable by others to their own models.

An extended tutorial-style git-hub page for PC data can be found at https://github.com/JanMuench/Tutorial_Patch-clamp_data and for confocal patch-clamp fluorometry data, https://github.com/JanMuench/Tutorial_Bayesian_Filter_cPCF_data.

2. The authors should clearly discuss the types of data and experimental paradigms that can be optimally handled by this approach, and they must explain when and where it fails or cannot be applied or becomes inefficient in comparison with other methods. One must be aware that ion channel data are very often subject to noise and artifacts that alter the structure of microscopic fluctuations. It seems that the state inference algorithm would work optimally with low noise, stable, patch-clamp recordings (and matching fluorescence recordings) in heterologous expression systems (e.g., HEK293 cells), where the currents are relatively small, and only the channel of interest is expressed (macropatches?). In contrast, it may not be effective with large currents that are recorded with low gain, are subject to finite series resistance, limited rise time, restricted bandwidth, colored noise, contaminated by other currents that are (partially) eliminated with the P/n protocol with the side effect of altering the noise structure, power line 50/60 Hz noise, baseline fluctuations, etc. This basically excludes some types of experimental data and experimental paradigms, such as recordings from neurons in brain slices or in vivo, oocytes, etc. Of course, artifacts can affect all estimation algorithms, but approaches based on fitting the predicted average current have the obvious benefit of averaging out some of these artifacts.The discussion in the manuscript is insufficient in this regard and must be expanded. The method should be tested under non-ideal but commonly occurring conditions, such as limited bandwidth and in the presence of contaminating noise. For example, compare estimates obtained without filtering with estimates obtained with 2, 3 times over-filtering, with and without large measurement noise added (whole cell recordings with low-gain feedback resistors and series resistance compensation are quite noisy), with and without 50/60 Hz interference. How does the algorithm deal with limited bandwidth that distorts the noise spectrum? How are the estimated parameters affected? The reader will have to get sense of how sensitive is this method to artifacts. Also, fluorescence data in particular is usually of much lower temporal resolution than current measurements. How does this impact its benefit to parameter estimation?

We developed the algorithm for analysing patch-clamp or patch-clamp fluorometry data though we are convinced that other experimental settings are applicable, as long as normal distributions can be reasonably applied and the white noise assumption for the measurement process holds. We show that it is advantageous to use the data in the ”rawest” form possible, e.g. without additional filtering (Figure 11 and App. 7 for patch-clamp data). Instead of smoothing, averaging, or subtraction, optimal statistical description of noise sources is needed in our approach. For example, in uncompensated whole-cell measurements the RC-circuit of the membrane is a low-pass filter with known characteristics. It should be noted that any active compensation of the series resistance certainly affects the noise characteristics in a hardly predictable way. It should therefore be used with particular care. In contrast, traces corrected by P/n protocols (given sufficiently low open-probability of the inactive channel) should be treatable with our approach when including the additional variance of the noise introduced by the P/n trace adequately. In fact, the assumed fluorescence signal in a confocal patch-clamp fluorometry experiment is a difference signal just as the current signal obtained with the P/n technique. Here, the amount of mean signal of swimming bulk ligands is subtracted from the original signal. Thus, in the current signal in Equation 4 we simply need to add a second noise term whose variance needs to be characterized. Non-stochastic powerline or slow baseline fluctuations are not described by our approach and should be minimized by experimental means. Although, adding a drift term in Equation 4, governed by a Gaussian process with an appropriate kernel, is worth to try. In the similar way, with a periodic kernel powerline errors could be tackled. In the “Kalman filter derived from a Bayesian filter” section (line 244-255) we specify those considerations.

Moreover, we selected now two non-ideal aspects induced by limitations of real experiments to investigate the robustness of the idealizations of our algorithm versus the approach by Milescu et al., 2005. First, we analyzed the bias induced by the analog filtering of confocal patch-clamp fluorometry data before the analog-to-digital conversion (Figure 11). For PC only data see App. 7. Second, we investigated finite integration times (Figure 12) of fluorescence data (as explicitly asked by Reviewer 1). We chose to combine those two aspects because of their similar physical origin. Both of them induce an intrinsic time scale of the recording system which is not represented in the algorithms.

3. Even more emphasis on the approximation of n(t) as being distributed according to a multivariate normal, and thus being continuous, should be placed in the main text. It seems that this may limit the applicability of the method to data with > ~100s of channels; although the point is not investigated. In Figure 3, the method is only benchmarked to a lower limit of ~500 channels.

We expanded for confocal patch-clamp fluorometry data (Figure 4a, 7d,e) the benchmark to channel numbers of 5 · 10^1^ to 10^6^ per patch. At the lower limit below *N*_ch_
*<* 200 we see that highest density volumina (HDCV) do not report the correct success probability of the true parameters to be inside this volume. Below this limit both algorithms have very similar Euclidean errors and the deviations of the highest density intervals for both algorithms for each *N*_ch_ become similar. Therefore, we suspect that the error due to the approximations of the multinomial distribution becomes more relevant than the autocorrelation of the intrinsic noise. In principle, the almost equal errors could also result from a bias induced by the uniform prior over the rate matrix. Therefore, we tested the Jeffreys Prior (loguniform on the dwell times) and dirichilet distribution on the probabilities which transition is chosen and did not detect differences for confocal patch-clamp fluorometry data. For PC data (Figure 3c), we see that below *N*_ch_
*<* 2000 the posterior becomes improper for some parameters (unidentified parameters) such that here a comparison is problematic to interpret. Below *N*_ch_
*<* 2000 the mentioned Jeffreys prior has an enormous effect but to avoid prolongation of the text we save this aspect for a separate paper about priors in kinetic scheme estimation. We added in the main text a reference to Moffat, 2007 (line 201-207). He applied his algorithm even for ion channel numbers as low as *N*_ch_ = 1 and found in Figure 4 an inverse relationship ∝1Nch for the difference between the true log likelihood value and the loglikelihood of his Kalman filter approximation which establishes around *N*_ch_ = 20. Of course, the differences also depend on which data points of the time traces are used for the inference because the quality of the used approximations also depend on where on the probability simplex it is applied. Thus, in this regard our results should be seen as a rule of thumb for the lower limit. Nevertheless, we expect a similar behaviour.

As shown in Milescu et al., 2005, fitting macroscopic currents is asymptotically unbiased. In other words, the estimates are accurate, unless the number of channels is small (tens or hundreds), in which case the multinomial distribution is not very well approximated by a Gaussian. What about the predictor-corrector method? How accurate are the estimates, particularly at low channel counts (10 or 100)?

We expanded the benchmarking for confocal patch-clamp fluorometry data (Figure 4) to low channel counts of 50. We observe that both algorithms have similar Euclidean errors if the channel numbers for confocal patch-clamp fluorometry data for the simple model drops below *N*_ch_
*<* 200. For those small ion channel ensembles, the uncertainty quantification by Bayesian credibility intervals/volume (Figure 4. b1-6) becomes unreliable even for the Kalman filter. Note, we showed that uncertainty quantification by Bayesian credibility volume in combination with the rate equation approach is generally unreliable because the posterior is to narrow (over-confident) leading to bad coverage properties of its Highest Density Credibility Volume/Interval (Figure 5 and 6). Thus, in that regime both algorithms fail in a similar way.

On the one hand, these problems can originate from the used approximations of both algorithms to derive the likelihood. One the other hand, as information from the data gets weaker in the lower regime it could also be a bias from the uniform prior of the rate matrix. This is because a uniform prior is not an unbiased minimum information prior for rates, as indicated by Reviewer 3. Tests with a Jeffreys prior for confocal patch-clamp fluorometry data with the 4-state model did not reveal a significant difference such that we consider in the article only the first option and for clarity took out prior discussions.

We also like to note that the difference between the exact likelihood and a Bayesian/Kalman Filter approximation is investigated for patch-clamp data of a 3-state model in (Moffatt, 2007) (Figure 2, 3 and 4). He shows that the difference of the log likelihood of data between an algorithm with the exact likelihood and the Kalman filter implementation scales with ∝ 1*/N*_ch_ if *N*_ch_
*>* 20.

Since the Kalman filter also uses a Gaussian to approximate the multinomial distribution of state fluctuations, I would also expect asymptotic accuracy.

For a 5-state-2-open-states model for PC data (Figure 9), we demonstrate that the advantage of the KF is a bias reduction compared to the RE approach. Thus, a high accuracy is reached by the KF already at *N*_ch_ = 10^3^. In contrast, we detect bias and inaccurate inferences of the RE approach even for ensemble sizes as large as *N*_ch_ = 5·10^5^. Due to the small number of test data sets a small undetected amount of inaccuracy in parameter inferences of the KF is possible. Additionally, for some data sets of high quality *N*_ch_ = 7*.*5·10^4^ the RE algorithm fails to identify parameters, whereas the KF algorithm has no identification problems.

In contrast, with confocal patch-clamp fluorometry data for all tested models bias and inaccuracy is a smaller but still present issue (Figure 11 and 12). The over-confidence problem of the RE algorithm remains for all tested models.

Parameter accuracy should be tested, not just precision.

consistent estimators (given an identifiable model) as their Euclidean error (distance from the true value) vanishes in distributions with increasing data quality. If any estimator is consistent it has to be asymptotically accurate and precise.

Accurate means

accurate = *E*[*θ_i_* − *θ_i,_*_true_] = 0 (1)

with *i* indicating the *i*-th parameter and *E* means the arithmetic mean of many inferences with different data sets. The common mathematical definition of precision is(2)σ=1/Ninference∑iNinference(θi−E[θ])2.

For a data set of finite quality or amount, the estimators could still be inaccurate (biased) and/or imprecise but it could also be that inferences are perfectly accurate (unbiased) and only imprecise. To see that the last aspect is true, take for example a perfectly accurate inference: Thus, the mean equals exactly the true value *E*[*θ*] − *θ*_true_ = 0 and assume its imprecision is Σ = 1. In this case the Euclidean error ∑iNprametersθi2 , is a random variable that is *χ*-distributed. Thus, the mean Euclidean distance equates to E[error]=2Γ((Nparameter+1)/2)Γ(Nparameter/2)≠0, with Γ() being the γ function. In conclusion, the Euclidean error is not a direct estimator of precision but has good summarizing properties.

We display now in Figure 4b1-6, 8 and 9 and App. 9 accuracy and precision implicitly by the narrowest highest density interval. By implicitly, we mean that both aspects are demonstrated by the local trend (accuracy) and the spread (precision) around the trend vs. *N*_ch_. In this representation, perfect accuracy means *E*[*θ*^˜^*_i_*] = 1 as the parameters are normalized to their true value. The advantage of the single parameter level is that systematically inaccurate (biased) estimates *E*[*θ*^˜^*_i_*] 6 = 1 for a finite data quality can be detected Figure 9 for which the more summarizing Euclidean error is blind.

Conceptually, we prefer the Bayesian highest density credibility interval. In the new version of the article we argue that Euclidean error and Bayesian highest density volumes together are a very effective way to probe the performance of an algorithm (Figure 5. and Figure 6). Highest density credibility volumes (HDCV) allow uncertainty quantification exactly in the way the researcher desires in most cases. HDCVs allow to determine the probability mass of the true parameters to be in a certain volume for a given data set. This guided us to the binomial testing of the overall shape of the posterior. This interpretation is not possible for the usual confidence intervals/volumes of maximum likelihood, Janes, et al., 1976. In the benchmark situation of this article, we used the HDCV to show that the RE approach is generally too confident in the Bayesian parameter uncertainty quantification, which also leads to too confident (narrow) confidence volumes or errorbars in a maximum likelihood context. Note, that this shortcoming had remained undetected if we would have explored only accuracy and precision. Later in the article we worked explicitly with the classical definition of accuracy and precision because we want to understand the mean bias (for all possible random data sets) induced by analog Bessel-filtering (Figure 11 and App. 7 Figure 1) the data before analysing them. The same applies to the mean bias due to limited time resolution of fluorescence data (Figure 12).

4. Achieving the ability to rigorously perform model selection is very impressive aspect of this work and a large contribution to the field. However, the manuscript offers too many solutions to performing that model selection itself along with a long discussion of the field (for instance, line 376-395 could be completely cut). Since probabilistic model selection is an entire area of study by itself, the authors do not need to present underdeveloped investigations of each of them in a paper on modeling channel data (e.g., of course WAIC outperforms AIC. Why not cover BIC and WBIC?). The authors should pick one, and maybe write a second paper on the others instead of presenting non-rigorous comparisons (e.g., one kinetic scheme and set of parameters). As a side note, it is strange that the authors did not consider obtaining evidence or Bayes factors to directly perform Bayesian model selection – for instance, they could have used thermodynamic integration since they used MC to obtain posteriors anyway (c.f., Computing Bayes Factors Using Thermodynamic Integration by Lartillot and Philippe, Systematic Biology, 2006, 55(2), 195-207. DOI: 10.1080/10635150500433722)

We took out the whole model selection part and saved it for a successor publication.

5. Regarding model selection for the PC data in Figure 7, why does the RE model with BC appear to provide a visually better identification of the true model than the KF model with BC if the KF model does a better job at estimating the parameters and setting non true rates to zero? Doesn't this suggest that RE with cross validation is better than the proposed KF approach in regards to model selection? In terms of parameter estimates (i.e. as shown in Figure 3), how does RE + BC stack up?

We are pleased to clarify this point though the model selection will be placed in a second article. In the previous version of the article, we presented in Figure 6 and Figure 7 two related aspects.

First, we define overfitting in the usual way as a performance difference of a trained model when it is used to predict the training data and prediction data. A trained model, no matter if under complex or overly complex performs on average better to predict the already used trainings data than unseen new data. When we choose a too complex kinetic scheme in which the true process is nested we observe, that the RE approach, when applied to a more complex model, has a higher tendency to overfit. We concluded that from the observation that the RE algorithm concentrates less the posterior mass around zero for rates which do not exist in the true process (Figure 6 of the previous article) than the KF. The RE algorithm places during the training more statistical weight on values of rates to explain fluctuations which are specific for the training data set. This happens no matter which model is used just because the amount of data is finite. That leads due to the higher flexibility of a too complex model to even more ways to overfit the data set. This means for a kinetic scheme trained by an RE algorithm, that if the trained model is now used to predict new data, it does not perform (generalize) as good as if the simpler model would have been trained and then used to predict new data (cross validation). In mathematical terms, more statistical weight on structures which are not present in the real process reduces, of course, the value of Equation 16 of the previous version of the article. This tendency to overfit more of the RE approach can be exploited if the model inference is done by Bayesian cross validation (Figure 7a, see Equation 16 of the previous version of the article). Because cross-validation penalizes more the bigger tendency of RE approach on the too complex model to overfit to the trainings data.

Thus, the conclusion of Reviewer 1 is the same as we drew in lines 406 -410 of the previous version of the article “This means, ignoring intrinsic fluctuations of macroscopic data, such as RE approaches do, leads to the inference of more states than present in the true model if the model performance is evaluated by the training data. One can exploit this weakness by choosing the kinetic scheme on cross-validated data, since too complex models derived from the RE approach do not generalize as good to new data as a parsimonious model”. We were intending to state that if one uses a RE approach one is limited to cross validation for the kinetic scheme selection. To the best of our knowledge, cross-validation is not the default strategy how models are selected. But the data (Figure 7a in the previous version of the article) show that it should be the method of choice if an RE approach is used.

The Bayesian filter is more flexible than a RE approach. It is more reliable to detect structures that are over-fitted by asking for which rates is the probability mass close to zero (continuous model expansion). But it can also use estimators of the minimum Kullback-Leibler entropy (Bayesian cross validation or information criteria such as WAIC). Because, Bayesian cross validation is data intensive, it is an advantage to have an algorithm which is able to perform this task on the training data by WAIC.

6. A better comparison with alternative parameter estimation approaches is necessary. First of all, explain more clearly what is different from the predictor-corrector formalism originally proposed by Moffatt (2007). The manuscript mentions that it expands on that, but exactly how? If it is only an incremental improvement, of interest to a very limited audience, a specialized, technical journal, is more appropriate.

We thank the Reviewer for mentioning these concerns. First, we inserted the new Figure 3 to address these concerns. In particular, Figure 3b shows even for tiny standard deviations of the open-channel noise, measured in units of single channel current *σ_op_/i* ≈ 0*.*01, that the previous Kalman filter, Moffat, 2007, produces a larger Euclidean error than our algorithm. The magnitude of the Euclidean error of the parameters of the kinetic scheme inferred by Moffat, 2007, becomes quickly even worse than the Euclidean Error of Milescu et al., 2005 while our algorithm and the one used by, Milescu et al., 2005 is hardly affected. The reason for this is revealed if one considers the scaling of the total amount of open-channel noise in the macroscopic signal. In fact, we interpret our findings such that for most ion channels this generalization from pure additive to additional multiplicative noise (from adding instrumental noise of a constant variance to adding instrumental noise whose variance depends on the ensemble state) is a necessary step to apply the Kalman filter. Second, we discuss the mathematical difficulties now in more detail in the main text in the subsection “Kalman filter derived from a Bayesian filter”.

Second, the method proposed by Celentano and Hawkes, 2004, is not a predictor-corrector type but it utilizes the full covariance matrix between data values at different time points. It seems that the covariance matrix approach uses all the information contained in the macroscopic data and should be on par with the state inference approach. However, this method is only briefly mentioned here and then it's quickly dismissed as "impractical".

Thanks for pointing out that our concerns with the algorithm by Celentano and Hawkes, 2004, were not convincing. We agree that the likelihood of the Celentano and Hawkes algorithm seems to be a mathematical equivalent. But our major concern is computational speed. We agree that there is no sharp borderline between possible and impossible with regard to a Bayesian adaptation of the Celentano and Hawkes, 2004, algorithm and our algorithm. Thus, we softened the tone by writing “non-optimal or even impractical” (line 62-63). Nevertheless, we clarified our reasons to call it impractical here after the next quote and in the main article (line 60-72).

We all agree that it's a slower computation than, say, fitting exponentials, but so is the Kalman filter. Where do we draw the line of impracticability? Computational speed should be balanced with computational simplicity, estimation accuracy, and parameter and model identifiability. Moreover, that method was published in 2004, and the computational costs reported there should be projected to present day computational power.

To clarify our concerns, note that a maximum likelihood optimization usually takes some orders of magnitude fewer likelihood evaluations compared to the number of posterior evaluations when one samples the posterior (App. 1). The reason is not simply the larger number of samples by some orders of magnitude compared to the iteration steps of a maximum likelihood optimizer: For each excepted sample, the Hamiltonian Monte Carlo sampler (Betancourt, 2017) evaluates the derivative of the likelihood and prior distribution many times. This is necessary for numerically integrating the Hamiltonian equations via a leapfrog integrator. The evaluations of the likelihood by Celentano and Hawkes, 2004, have a computational complexity due to matrix inversion from *N_data_*^2*.*3^ to *N_data_*^3^ while the Kalman filter has a linear computational complexity in *N_data_* (App. 1).

We are not saying that the authors should code the C&H procedure and run it here, but should at least give it credit and discuss its potential against the KF method.

We expanded our considerations on the Celentano and Hawkes algorithm, and its offsprings, in the introduction (line 60-72). Based on the Celentano and Hawkes algorithm, Stepanyuk and Borisyuk, 2011, and Stepanyuk, 2014, published an algorithm which evaluates the likelihood quicker. Under certain conditions they claim that their algorithm can be faster than the Kalman filter from Moffatt, 2007.

A second point relates again to the computational complexity. Analog filtering before the analog-to-digital conversion distorts the dynamics of interest. Instead of a extensive analog filtering, it is beneficial to use the Kalman filter with a high analysing frequency because it is mathematically proven to be the minimal variance filter, Anderson et all, 2012, given the assumptions mentioned in the article are fulfilled. Under these conditions the linear computational complexity relative to *N*_data_ becomes even more important.

That said it is not sufficient to just care about the scaling of the computational time. Memory operations (App. 1 Figure 2) of the code between the CPUs become quickly the bottleneck of the total computational time. For example, if the algorithm is asked for each data point to save the covariance matrix and mean vector of the signal, the computational time increases roughly by 5 · 10 − 10^2^ times (App. 1 Figure 2).

A third reason for choosing the KF, is the implementation of a second dimension of the signal which is straightforward for the Bayesian filter as all of the Kalman filter equations are vector and matrix equations.

The only comparison provided in the manuscript is with the "rate equation" approach, by which the authors understand the family of methods that fit the data against a predicted average trajectory. In principle, this comparison is sufficient, but there are some issues with the way it's done.Table 3 compares different features of their state inference algorithm and the "rate equation fitting", referencing Milescu et al., 2005. However, there seems to be a misunderstanding: the algorithm presented in that paper does in fact predict and use not only the average but also – optionally – the variance of the current, as contributed by stochastic state fluctuations and measurement noise. These quantities are predicted at any point in time as a function of the initial state, which is calculated from the experimental conditions. In contrast, the KF calculates the average and variance at one point in time as a projection of the average and variance at the previous point. However, both methods (can) compare the data value against a predicted probability distribution. The Kalman filter can produce more precise estimates but presumably with the cost of more complex and slower computation, and increased sensitivity to data artifacts.

We thank the Reviewer for this criticism because it allows us to point out what we liked to say with Table 3 in the previous version of the article. We used the terminology of the Kalman filter which separates the prediction equations for the mean and covariances of the system from the correction equations. The prediction equation evolves the mean and the covariance in time. The prediction of the mean value of the RE approach is identical to the one of the Kalman filter but there is no prediction equation for the covariance in the terminology sense of the Kalman filter. This does not mean that we intended to say that the RE approach cannot predict a covariance of the signal and state. The RE approach uses the multinomial assumption and the mean values for the prediction of the covariance. But this distinction obviously leads to unnecessary confusion and is not necessary. We therefore deleted the table.

As a rule of thumb at moderate analysing frequencies, the KF is usually 2 to 3 times slower than the RE algorithm (App. 1 Figure 3) for cPCF data. Using higher analysing frequencies for the patch-clamp data (App. 1 Figure 4a) the computational time of the Kalman filter can be even faster. This effect is not present at a lower analysing frequency. (App. 1 Figure 4b). For the two investigated data artifacts (Figure 11, 12 and App. 7) we could show that the Kalman filter performs better.

Figure 3 is very informative in this sense, showing that estimates obtained with the state inference (KF) algorithm are about 10 times more precise that those obtained with the "rate equation" approach. However, for this test, the "rate equation" method was allowed to use only the average, not the variance.Considering this, the comparison made in Figure 3 should be redone against a "rate equation" method that utilizes not only the expected average but also the expected variance to fit the data, as in Milescu et al., 2005. Calculating this variance is trivial and the authors should be able to implement it easily (reviewers are happy to provide feedback). The comparison should include calculation times, as well as convergence.

We benchmarked against an in-house algorithm, but of course, Milescu et al., 2005, and Moffat, 2007, should be considered the gold standard. Therefore, we now challenge (Figure 3) our algorithm for patch-clamp data by the Bayesian version of Milescu et al., 2005 and Moffat, 2007. The error ratio between both algorithms amounts to 5*.*6 ± 1*.*4 for 4-state model and to 6*.*8 ± 2*.*7 for 5-state model (Figure 7).

In particular, the algorithm used by Moffat, 2007 suffers from state-depended noise (Figure 3b). Consequently, his approach is not applicable for Poissonian distributed fluorometry data. Thus, we benchmark only against Milescu et al., 2005 for confocal patch-clamp fluorometry data on a 4-state model (Figure 4), and on two more complex models (Figure 7,8 and 9) with 5 and 6 states. We show that the error ratio between the KF and the RE algorithm gets smaller when the second observable is added. For all tested models (Figure 7) the error ratio is about 1.5-2.0. Thus, adding the variance information, as asked by the Reviewer, made a large difference, similar to patch-clamp data, Milescu et al., 2005. That said, we show in Figure 4,5,6 that even if the error ratios are smaller the posterior sampled by an RE algorithm is meaningless as it is over-confident, which has also consequences for the confidence volume of maximum likelihood estimation (see discussion around Figure 4b,5,6,8,9).

In Figure 7b and d, we show that the advantage of the KF for patch-clamp data depends on the complexity of the process to be inferred. The KF is unbiased and identifies all parameters while the RE algorithm generates on the same data biased estimates and sometimes unidentified parameters.

For patch-clamp data of the most complex tested 6-state model (Figure 7 e) even the KF requires at least *N*_ch_
*>* 4 · 10^4^ channels per patch to identify all parameters. We suspect that this effect is caused by the 12 parameter dimensions of the rate matrix. Probably, the 12 dimensions make it difficult for the likelihood to dominate the bias picked up from the uniform prior.

We discuss calculation times and convergence in App. 1 Figure 2-4. Normally, the Kalman filter is 2-3 times slower than the RE approach. But against intuition, (App. 1 Figure 4a) we show that the Kalman filter can be even faster than a Rate equation type algorithm if the analysing frequency is high. This is only possible if the integration times of the Hamiltonian dynamics of the sampler are taking much longer for the RE approach. We showed that the RE approach creates too confident posteriors which in turn could lead to more integration steps of the leap-frog integrator of the sampler. Thus, algorithms with lower computational complexity in the likelihood evaluation can still be slower regarding the total computational time.

7. The manuscript nicely points out that a "rate equation" approach would need 10 times more channels (N) to attain the same parameter precision as with the Kalman filter, when the number of channels is in the approximate range of 10^2 … 10^4. With larger N, the two methods become comparable in this respect.This is very important, because it means that estimate precision increases with N, regardless of the method, which also means that one should try to optimize the experimental approach to maximize the number of channels in the preparation. However, it should be pointed out that one could simply repeat the recording protocol 10 times (in the same cell or across cells) to accumulate 10 times more channels, and then use a "rate equation" algorithm to obtain estimates that are just as good.

As we were asked to redo the benchmark against the RE approach using the likelihood of Milescu, 2005, instead of least squares, there are many things that changed in our interpretation. First, we do not see anymore, in any of the tested scenarios for patch-clamp data (Figure 3c) our patch-clamp fluorometry data (Figures 4a, 7d,e) that both algorithms become equivalent in terms of the Euclidean error (Figures 3c, 4a, 7d,e) or HDCV (Figures 5c, 6a-b) with increasing channel numbers. They rather have a constant mean error ratio (Figure 7a,b) such that our old interpretation, to repeat the experiment 10 times, does not hold anymore.

Second, the ratio of the Euclidean error changes, such that with the second observable the KF is less better compared to the rate equation approach, given a constant model complexity (Figure 7a).

Third, we show that the rate equation approach is incapable to produce reliable credibility volumes. Thus, the Bayesian benefit that a parameter uncertainty quantification based on one data set is possible requires to use an algorithm which mimics the autocorrelation of the intrinsic noise, even if accuracy differences are small (Figures 5a-c). That is true even under pessimistic signal to experimental noise conditions (Figure 6b). This has consequences also for confidence volumes of maximum likelihood which at best are equivalent to the credibility volume if the likelihood dominates the prior, Jaynes, et all, 1976. For a discussion on limitations and work arounds of maximum likelihood error quantification of a dynamical system, see Joshi, et all, 2006.

Presumably, the "rate equation" calculation is significantly faster than the Kalman filter (particularly when one fits "features", such as activation curves), and repeating a recording may only add seconds or minutes of experiment time, compared to a comprehensive data analysis that likely involves hours and perhaps days. Although obvious, this point can be easily missed by the casual reader and so it would be useful to be mentioned in the manuscript.

Of course, the computational complexity for the calculation of the likelihood in a rate equation approach is smaller than the computational complexity of the Bayesian filter. We confirm the intuition that typically the RE algorithm is 1.8 till 3 times faster (App. 1 Figure 3) with one important exception for which the RE algorithm is slower (App. 1 Figure 4). We hypothesize that this exception is caused by the fact that, the full computational time to gain *N* samples depends not only on the computational complexity and memory operations but also how many integration steps of the Hamiltonian equations need to be applied to suggest a new sample. If an algorithm such as the RE approach generates much narrower posteriors, the curvature of the posterior is higher and, consequently the HMC sampling would require more integration.

But even if our discussed limits of the RE approach are considered to be unimportant, the bottle neck in the whole scientific process data acquisition or data analysis/modeling depends on the one hand on both the speed of the data acquisition and the complexity of the modeling question. On the other hand, it depends on the resources of the laboratory. Data acquisition of fast voltage-gated channels might take many orders of magnitude less time than a conconfocal patch-clamp fluorometry experiments in slowly gating HCN channels. Taking other limitations into account, such as e.g. photo toxicity when working with fluorescent ligands, it is not even a question of the time scales but also a matter of how many repetitions can be done at all.

8. A misunderstanding is that a current normalization is mandatory with "rate equation" algorithms. This is really not the case, as shown in Milescu et al., 2005, where it is demonstrated clearly that one can explicitly use channel count and unitary current to predict the observed macroscopic data. Consequently, these quantities can also be estimated, but state variance must be included in the calculation. Without variance, one can only estimate the product i x N, where i is unitary current and N is channel count. This should be clarified in the manuscript: any method that uses variance can be used to estimate i and N, not just the Kalman filter. In fact, the non-stationary noise analysis does exactly that: a model-blind estimation of N and i from non-equilibrium data. Also, one should be realistic here: in some circumstances it is far more efficient to fit data "features", such as the activation curve, in which case the current needs to be normalized.

We apologise for this error and corrected the introduction accordingly. In our old version this sentence referred rather to the in-house algorithm against which we benchmarked originally. The subtle but important differences in RE approaches were carelessly deleted during the text refinement. We indeed intended this article to be about kinetic scheme estimation not about fitting certain aspects of the data. Thus, we assume a situation where the data is qualified enough to try kinetic scheme inference.

9. It's great that the authors develop a rigorous Bayesian formalism here, but it would be a good idea to explain – even briefly – how to implement a (presumably simpler) maximum likelihood version that uses the Kalman filter. This should satisfy those readers who are less interested in the Bayesian approach and will also be suitable for situations when no prior information is available.

The programming language Stan offers to select from three options: Sampling the posterior, variational Bayesian inference and maximising the posterior. The same code provided by us can be used (see, Appendix 4). If one maximises the parameter of a posterior or simply of a likelihood it is then a matter of statements of prior distributions within the code. There is no need for reimplementation of the code.

10. The Bayesian formalism is not the only way of incorporating prior knowledge into an estimation algorithm. In fact, the reader may have more practical and pressing problems than guessing what the parameter prior distribution should be, whether uniform or Gaussian or other. More likely one would want to enforce a certain KD, microscopic (i)reversibility,

We thank the Reviewer for pointing out that our former text was not stating that adding prior information is not unique to Bayesian statistics. Similar approaches with regard to the maximum likelihood method for algebraic constrains (Salari et al., 2018) and behavioral constraints (Navarro et al., 2018) called penalized maximum likelihood, are now cited. We like to note, that the penalized maximum likelihood is synonymously called maximum a posteriori because the penalizing function is usually the logarithm of a prior (Gelman et all 2013). Thus, we do not think that it is helpful to separate simple prior considerations like uniform, Jeffrey or informative priors on subsets of or single parameters from more complex priors incorporating relations (Navarro et al., 2018) between parameters: For example, we demonstrate in the discussion of the methods section related to (Figures 7 and 9) how enforce from strictly to softly micro reversibility by means of a conditional prior on(3)k5∼P(k5|k1k2k3k4k6k7k8)

In this way we translated the algebraic constraint (Salari et al., 2018) to a rather behavioral constraint (Navarro et al., 2018) which might be chosen to be strict (thus to be effectively algebraic) or soft to allow testing whether for a channel the thermodynamic equilibrium assumption is fulfilled.

As this article is already quite long with investigating the behaviour and the robustness of the likelihood, we hardly dwell into prior selecting questions expect that we demonstrate a way to incorporate micro-reversibility. In general, all problems which can be formulated as log-likelihood with an added penalizing function for the parameters can be expressed as a prior multiplied by the likelihood and sampled (given that exp[penalizing function] can be integrated).

…an (in)equality relationship between parameters, a minimum or maximum rate constant value, or complex model properties and behaviors, such as maximum Popen or half-activation voltage.

The programming language Stan employs base parameter types, whose support (the interval between their minimal and maximal values) can be readily defined by the user. The Stan compiler does then the necessary transformation to an unconstrained parameter space where the sampling is takes places. More complex parameter relationships, such as inequalities, are incorporated by other predefined parameter types such as simplexes, ordered vectors etc. Again, the stan compiler does the nasty parameter transformation to an unconstrained space automatically such that these constraints are fulfilled. Thus, testing model assumptions can be implemented fast. A constraint such as *P*_open_ can quickly be enforced from the fact that

*P*_open_ = *f*(K*,L*) (4)

which means for each parameter sample of the rate matrix K we can calculate the open probability for any ligand concentration *L*. Suppose that we have derived from previous data posteriors P(*P*_open_(*L*)) of a set of open probabilities *P*_open_ for a set ligand concentrations. Then we can simply define the prior by

P(*P*_open_) = P(*f*(K*,L*)) (5)

It is not even necessary that the inverse function *f*^−1^(*P*_open_) = K exists (which it obviously does not). The only prerequisite for doing this information fusion in a mathematically rigorous attempt is that the previous analysis derived a posterior on *P_open_*. Proxy’s such as point estimates and their standard errors of should be used cautiously.

A hard limit for a derived quantity, which is not naturally fulfilled, can be rather tricky. A possible solution is defining a own prior on this derived quantity in the same manner as Equation 2. The Prerequisite is that it can be differentiated such as a sum of Gaussians. With increasing terms in the sum it is possible to model a hard limit as a soft limit. That in turn alleviates hard limit problems.

A comprehensive framework for handling these situations via parameter constraints (linear or non-linear) and cost function penalty has been recently published (Salari et al/Navarro et al., 2018). Obviously, the Bayesian approach has merit, but the authors should discuss how it can better handle the types of practical problems presented in those papers, if it is to be considered an advance in the field, or at least a usable alternative.

Unless the asymptotic large data limiting case of the Bernstein-von Mises theorem hasn’t taken over, it is crucial to select a reasonable prior distribution in the Bayesian context. Nevertheless, we restricted ourselves mainly to the behaviour of the likelihood of both algorithms (in the asymptotic strong data case) which offers many things to explore. If faced with the situation that a substantial amount of information enters the parameter inference via soft or inequality constraints (Salari et al/Navarro et al., 2018) one risks a waste of information if one uses a point estimator. That is because if one for example reports only the parameter vector of a penalized maximum likelihood inference and its covariance matrix most of the time one does not report the sufficient statistics of the corresponding posterior (Gelman et al., Chapter 4.5, 2013 third Edition). Take for example an inequality constraint that cuts at one standard deviation distance from the mean through the sampling distribution of a maximum likelihood inference. The mentioned Bernstein-von Mises theorem is a Bayesian justification for doing maximum likelihood inferences in the asymptotic limit (Gelman et al., Chapter 4.5, 2013 third Edition) when all prior information becomes irrelevant. Thus, we do not see any relevant differences in which statistical framework prior information for a parametric model can be formulated. But Bayesian statistics will make the most of the prior information unless the data completely dominates the prior information. A rather detailed discussion on priors is beyond the scope of this article.

11. The methods section should include information concerning the parameter initialization choices, HMC parameters (e.g. number of steps) and any burn-in period used in the analyses used in Figures 3-6.

We thank the Reviewer for pointing out the missing information: We included in App. 1 a section about the typically used parameter settings and convergence diagnostics. The Stan compiler generates a HMC sampler which adaptively tunes the sampling parameters. Additionally, the no-U-turn implementation (NUTS) automatically selects an appropriate number of leapfrog steps in each iteration. In that way there is much less knowledge of the user required to operate the program. Typically, we used 4 independent sampling chains with a warm up phase of 3 − 9 · 10^3^ iterations per chain followed by the actual iterations which generate the posterior. Convergence is monitored by the *R*ˆ statistics (Appendix 1).

For example, how is convergence established? How many iterations does it take to reach convergence? How long does it take to run? How does it scale with the data length, channel count, and model state count? How long does it take to optimize a large model (e.g., 10 or 20 states)? Provide some comparison with the "rate equation method".

We dedicated App. 1 to this discussion

12. In the section on priors, the entire part concerning the use of a β distribution should be removed or replaced, because it is a probabilistic misrepresentation of the actual prior information that the authors claim to have in the manuscript text. The max-entropy prior derived for the situation described in the text (i.e., an unknown magnitude where you don't know any moments but do have upper and lower bounds; the latter could be from the length from the experiment) is actually P(x) = (ln(x_{max}) – ln(x_{min}))^{-1} * x^{-1}. Reviewers are happy to discuss more with the authors.

We picked up these suggestions. We deleted this section and continued to employ uniform priors. We do not want to advocate a uniform prior for rate matrices. We use the uniform prior to be comparable to plain maximum likelihood, as it is the default method of most researchers, and to reduce the size of this publication. Unfortunately, the article is already quite long which makes us focus on the likelihood side of the Bayesian formalism. Therefore, we restrict the comparison between the algorithms to the situation where the likelihood dominates the uniform prior. Nevertheless, there are clear indications in Figure 3, 7, 8 and 9 that the uniform prior becomes insufficient for patch-clamp data, in particular for more complex (data generating) models and if the RE approach is used. We will present a detailed prior investigation in the near future in the archive version of another article.

13. Here and there, the manuscript somehow gives the impression that existing algorithms that extract kinetic parameters by fitting the average macroscopic current ("fitting rate equations") are less "correct", or ignorant of the true mathematical description of the data. This is not the case. Published algorithms often clearly state what data they apply to, what their limitations are, and what approximations were made, and thus they are correct within that defined context and are meant to be more effective than alternatives. Some quick editing throughout the manuscript should eliminate this impression.

We did not intend to generate this impression. What we liked to communicate was to emphasize both the similarities but in several details also relevant differences in the assumptions adopted by both approaches. We revised the manuscript accordingly.

14. The manuscript refers to the method where the data are fitted against a predicted current as "rate equations". However, it is not completely clear what that means. The rate equation is something intrinsic to the model, not a feature of any algorithm. An alternative terminology must be found. Perhaps different algorithms could be classified based on what statistical properties are used and how. E.g., average (+variance) predicted from the starting probabilities (Milescu et al., 2005), full covariance (Celentano and Hawkes, 2004), point-by-point predictor-corrector (Moffatt, 2007).

We thank the Reviewer for this criticism which helped us to clarify our terminology. We agree that rate equations are something intrinsic of a kinetic scheme but this does not exclude them to be also a feature of an algorithm: A good Bayesian model is always a forward model. This means, given a proper prior, one can simulate parameters and then (given parameters) sample data at every time point from the likelihood which in turn is calculated either by the deterministic rate equation or by the first order Markov process of the KF. For the inverse problem every Bayesian inference algorithm takes the model assumptions of the forward model and uses them to implement a calculation rule for the likelihood and the prior given the data. In this way, it makes sense to us to speak of an algorithm as a mathematical (non-unique) representation of the set of model assumptions/features which allows to calculate the posterior. The same applies to maximum likelihood models except that parameters are fixed and can not be updated once new information is available. They have to be refitted completely. Our terminology is coherent with the terminology of the statistical research community. They mean by statistical model either a likelihood or the combination of likelihood and prior.

15. The manuscript needs line editing and proofreading (e.g., on line 494, "Roa" should be "Rao"; missing an equals sign in equation 13). Additionally, in many paragraphs, several of the sentences are tangential and distract from communicating the message of the paper (e.g., line 55). Removing them will help to streamline the text, which is quite long.

Thanks. We changed Roa to Rao and also streamlined and divided the text. The changes evoked by the detailed criticism slightly prolonged the article.

[Editors' note: further revisions were suggested prior to acceptance, as described below.]

Essential revisions:Reviewer #1 (Recommendations for the authors):The authors develop a Bayesian approach to modeling macroscopic signals arising from ensembles of individual units described by a Markov process, such as a collection of ion channels. Their approach utilizes a Kalman filter to account for temporal correlations in the bulk signal. For simulated data from a simple ion channel model where ligand binding drives pore opening, they show that their approach enhances parameter identifiability and/or estimates of parameter uncertainty over more traditional approaches. Furthermore, simultaneous measurement of both binding and pore gating signals further increases parameter identifiability. The developed approach provides a valuable tool for modeling macroscopic time series data with multiple observation channels.The authors have spent considerable effort to address the previous reviewer comments, and I applaud them for the breadth and depth of their current analysis.

We are glad that our revision could convince the reviewer.

1. The figure caption titles tend to say what analysis or comparison is being presented, but not what the take home message of the figure is. I suggest changing them to emphasize the latter. This will especially help non-experts to understand what the figures are trying to convey to them.

We checked and modified the figure caption titles that all provide now a brief take home message.

2. I very much appreciate the GitHub code and examples for running your software. However, I feel that a hand-holding step-by-step explicit example of running a new model on new data is likely necessary for many to be able to utilize your software. Much more hand-holding than the current instructions on the GitHub repo.

We followed the suggestion and edited the tutorial https://github.com/

JanMuench/Tutorial_Patch-clamp_data/blob/main/README.md

and https://github.com/JanMuench/Tutorial_Bayesian_Filter_cPCF_data/blob/main/ README.md to make it more a step-by-step description of how to analyze a new data set with a new model.

3. Figure captions sometimes do not explain enough of what is shown. I appreciate that many of these details are in the main text, but data that is displayed in the figure and not concretely described in the caption can makes the figures hard to follow. e.g. Figure 4a – "With lighter green we indicate the Euclidean error of patch-clamp data set." But what data set do the other colors reflect? It is not stated in the caption. Again, I realize this is described in the main text, but it also needs to be defined in the caption where the data is presented. Figure 4d – Please spell out what "both algorithms" intends. Also, a suggestion: instead of having to refer to the caption for the color codes (i.e. RE vs. KF, etc.) it would speed figure interpretation to have a legend in the figures themselves. Few other examples: Box 1. Figure 2. – Please define the solid vs. dashed lines in the caption. Figure 3c – Please define the solid vs. dashed lines in the caption. Figure 12 – "We simulated 5 different 100kHz signals." What kind of signals? Fluorescence I assume, but this needs to be explicitly defined. I'd check all figures for similar.

We edited the Figure 3,4 caption in this regard. Every panel has its legend and the color coding of all panel matches. But in order to keep the figure caption and the figure together small enough that they fit onto one page we took away some of the redundant information from the caption. Additionally, Figure 8, 9 got the information how much probability mass is included in each HDCI. We also edited the caption of Figure 12 and state now that we simulated cPCF data.

Reviewer #3 (Recommendations for the authors):In this revised manuscript, the Münch et al. have addressed all of my original concerns. It is significantly revised, though, and includes many new investigations of the algorithm's performance. Overall, the narrative of this manuscript is now to introduce an approximation to the solution for a Bayesian Kalman Filter, and then spend time demonstrating that this approximation is reasonable and even better than previous methods. In my opinion, they successfully do this, although, as they mention in their comments, their manuscript is very long.

We thank the reviewer that he appreciates our work.

I am not 100% certain, but the approximation that the author's make seems to be equivalent (or at least similar to) an approximation of the chemical master equation using just the 1st and 2nd moments, which is just the Fokker-Planck equation. The authors should discuss any connection to this approximation, as there is a great deal of literature on this topic (e.g., see van Kampen's book).

We added a respective sentence in the introduction about protein expression studies which approximates the solution of the chemical master equation (CME) with the linear noise approximation to derive a Kalman filter. The linear noise approximation is the van Kampen’s System size expansion “pursued only up to first order”[13], or a linearized Fokker-Planck equation [11].

We added a paragraph in the conclusion to contextualize our method against the background of the classical approximations of the CME. We conclude due to the first order dynamics of chemical reaction network our approach is actually equivalent to the full chemical Fokker-Planck equation.

In Figures 3A and 4D, it is unclear to me what is plotted for the RE and classical Kalman filter (i.e., how is there a posterior if they are not Bayesian methods)? Perhaps it is buried in the methods or appendices, but, if so, it needs to at least be clarified in the figure captions.

We implemented a full Bayesian Version of Milescu 2005, algorithm and Moffatt 2007, algorithm, thus all results in this paper were obtained by posterior sampling. Since we used uniform priors, the mode of the posterior equals the maximum of the likelihood, i.e. the outcome of the classical implementation as a point estimator. However, our implementation of Milescu 2005 algorithm and Moffatt 2007 algorithms has the advantage that all sorts of priors can be applied. Unfortunately, our wording was somewhat ambiguous: Every Kalman filter is Bayesian filter (Moffatt, 2007) regarding the time varying unknown **n**(*t*) ensemble state. That is independent of how the time constant parameters such as the rate matrix parameters are inferred. The wording “Bayesian Filter” for our algorithm was used to distinguish it from the previous algorithm (Moffatt, 2007), as only our generalized Bayesian filter allows the inclusion of state dependent noise. We added to the manuscript in the caption and the main text that we implemented both algorithms in the full Bayesian framework.

The Bayesian statistical tests devised to determine success (e.g., on pgs. 12-13) seem a little ad hoc, but are technically acceptable. I do not see a need for additional metrics.

Interestingly, we found out is is not that actually not that “ad hoc” as we thought in the beginning. We in fact test the frequentist coverage property of “credibility” intervals or volumes, see [8] Equation 2.1. Usually for parametric Bayesian statistics it is at least asymptotically clear by the Bernstein von-Mises theorem (when the data swamped the prior) that coverage property should hold the frequentist interpretation of probability [9]. Just like us, others interpret the Bayesian uncertainty quantification as “invalid” if this frequentist coverage does not hold. [10]. We add a sentence “Noteworthy, that this is a empirical test of how sufficient the Bayesian filter and the RE approach hold frequentist coverage property of their HDCVs.” to indicate the theoretical foundation of our approach.

Line 939: Equation 61 is absolutely not "close to uniform distribution". The α and β parameters of 100.01 are much larger than 1. It is incredibly peaked around 0.5. Perhaps this is a typo?

This was indeed a left-over text fragment from the previous version of the article. We changed it to ”sharply peaked β distribution”.

Line 942: The allowed small breaking of microscopic reversibility in the prior is an interesting idea that I wish the authors would expound upon more.

Thanks for this encouragement. Indeed, we plan to go more into detail on this topic in the future.

Line 712: The authors state that the simulation 'code will be shared up-on request'. They should include it with their github pages tutorials for running the examples in case others wish to check their work and/or use it. There is no reason to withhold it.

The simulation code is exemplified here:

https://cloudhsm.it-dlz.de/s/QB2pQQ7ycMXEitE and now also in the manuscript.

Line 707: 'Perspectively' is not a commonly used word.

Thanks. We changed “Perspectively, extensions …” to “Prospective extensions …”

Reviewer #4 (Recommendations for the authors):The authors have addressed all my comments and suggestions. The manuscript is nice and extremely comprehensive, and should advance the field.Nevertheless, the manuscript is also very long (but justifiably so), and certain statements could be a little clearer. Most of these statements refer to the comparison with the so-called rate equation (RE) methods, with which I'm more familiar. For example:

We thank the Reviewer for his appreciation of our work.

Abstract: "Furthermore, the Bayesian filter delivers unbiased estimates for a wider range of data quality and identifies parameters which the rate equation approach does not identify."The first part of this statement is not quite true, as Figure 4 shows clearly that the Bayesian estimates are biased (and the authors acknowledge this elsewhere in the manuscript). If they are biased in one "range of data quality", that probably means they are always biased, just to different degrees.

We agree that it is right to say “If they are biased in one “range of data quality” this probably means they are always biased, just to different degrees”. The approximations lead to all sorts of errors including bias. This argument is also supported by the findings of Moffatt, 2007 (Figure 11) in which he detects bias in the deterministic approach (Milescu, 2005) and his own Kalman filter. It can certainly be stated that Figure 4b indicates a bias. However, even for optimistically high signal-to-noise in the fluorescence signal as in Figure 4, reasonable large HDCIs such as the 0*.*95−HDCIs of Figure 4c are covering the true value. Thus, the bias can be interpreted as negligible relative to the parameter uncertainty quantification. One can actually observe how the size (the probability mass) of the HDCV of the Bayesian filter overcomes the bias in Figure 5c. We toned down our statement by adding the word “negligible” in the context with “biased estimates” in the abstract. Considering the differences in the bias in Figure 9, this seems justified to us. We also mention the bias now in the discussion of Figure 4.

This is not surprising, because the Kalman filter is a continuous state approximation to a discrete state process, and the overall estimation algorithm makes a number of approximations to intractable probability distributions. It would definitely be correct to say that the estimates are very good, but not unbiased.

We agree with the reviewer and added to the discussion of Figure 4 a statement indicating the bias of both algorithms.

Second, this statement is also ambiguous. Are you referring to the theoretical non-identifiability caused by having more parameters in the model than what the combination of estimator+experimental protocol can extract from data? In this case, it's not a matter of certain parameters that cannot be identified, but a matter of how many can be identified uniquely. The more information is extracted from the data, the more unique parameters can be identified, so the Kalman filter should do better. Or, alternatively, are you referring to specific parameters that are poorly identified because, for example, they refer to transitions that are rarely undertaken by the channel? In this case, it would be a matter of obtaining looser or tighter estimates with one method or another, but the parameters should be intrinsically identifiable, I imagine, regardless of the algorithm.

In the context of the whole article this sentence seems to be unambiguous to us. The abstract alone leaves of course room for interpretation but we defined what we mean by unidentified parameters by Equation 16 in the main article. To avoid confusion we changed in the abstract the sentence to “For some data sets it identifies more parameters than the rate equation approach”. By our definition we are able to detect many cases of both practical and structural unidentifiability (Middendorf,Aldrich 2017). But our definition is not equivalent to the definition by (Middendorf,Aldrich 2017). We added an explanatory paragraph after Equation 16.

In any case, it's not clear that the better identifiability is the result of the Bayesian side, or of the predictor-corrector state inference filter. I would guess it is the Kalman filter, but I'm not sure.

Indeed, it is only the “predictor-corrector” side of the algorithm. Unfortunately, our wording was somewhat ambiguous which gives rise to a couple of the Reviewer’s following questions: Every Kalman filter is a Bayesian Filter but not every Bayesian Filter is (the classical) Kalman filter. In fact, Moffatt, 2007 derives his Kalman filter as a Bayesian forward algorithm in continuous space. He thus shows that the forward algorithm and the Kalman filtering performed for discrete and continuous state Markov processes, respectively, are the same. We come to the same conclusion herein, see Equation 5-9. (Nevertheless, he optimizes the rate matrix etc. via maximum likelihood optimization.)

We additionally describe in Equation 11 the consequences of the open-channel noise, that Equation 11 cannot be derived by the original Kalman filter equations. A more flexible noise modeling is necessary. This leads to our algorithm which is exact up to the second moment. That also includes other state-dependent noise such as photon counting noise. We believe that this explicit noise description is the reason for the performance advantage of our algorithm. We would also expect this advantage if our algorithm would be implemented as point estimator. The full Bayesian implementation results in a fully sampled posterior and, in principle, allows for arbitrary priors. Since we used uniform priors, the mode of the posterior equals the maximum of the likelihood, i.e. the outcome of the classical implementation as point estimator.

Perhaps it would be clearer if you said that the KF method produces good estimates and generally improves parameter identifiability compared to other methods, as it extracts more information from the data?

As mentioned above, we changed the abstract to avoid misunderstandings.

Introduction:32: I'm not sure, but if the intention here is to cite mathematical treatments for estimation, you may add references to the "macroscopic" papers by Celentano, Milescu, Stepaniuk and perhaps a few others that use "rate equations". Also, you may cite Qin et al., 1996, as a single channel paper describing a method used in hundreds of studies.

Thank you for pointing out the Qin paper, 1996. We also refer now to the other proposed papers at that position in the text.

Pg. 3:51: I remain skeptical that it is a good idea to use "rate equations" (RE) as a term to refer to those methods that are fundamentally different from the approach described here (also see my comment to the first submission).

We follow with our terminology on stochastic processes that of other groups in the community [6], [4], (Walczak 2011). In this terminology, the rate equation or (reaction-)rate equation is a deterministic ordinary differential equation describing the time evolution of the first moment (the mean value) of the chemical master equation (CME) [3, 4]. To our understanding the rate equation is the basis of the work of Milescu 2005 which justifies the wording.

The rate equations must always be used to predict a future state from a past or current state, by all methods, explicitly or implicitly, because REs simply describe the channel dynamics. In this very manuscript, Equation 3, central to the Kalman filter formalism, is nothing but a deterministic rate equation with a stochastic term added to approximate the stochastic evolution of an ensemble of channels.

Indeed, all approximations (deterministic or stochastic) which make physically sense should have as the large system limit the solution of the RE and thus solve at least implicitly the RE. We hesitate to use today only the celebrated traditional chemical kinetics perspective [5] that the Res describe the (full) channel dynamics. But we acknowledge that some others do. Also, we are aware of its central importance as one aspect of the evolution of a chemical system. However, we prefer the accepted terminology [4, 5, 12] that the CME is the governing equation of a well stirred ensemble of chemically reacting molecules or in other words of a Markov jump process. Our Bayesian filter uses only stochastic approximation Equation 3 of the CME, not the CME itself.

Strictly speaking, Equation 3 is the solution of the rate equation plus a stochastic perturbation. The perturbation around the mean evolution defines it as a stochastic approach. It is a standard technique to approximate the solution of the CME around the mean evolution (the RE) with a Gaussian perturbation [4],[12]. In this regard our method does not differ from the cited methods but it does differ from the RE approach by its rigorous evaluation of the likelihood. Consequently the solution of the RE without any perturbation does also occur in the Bayesian filter as the time evolution of the mean value of the ensemble vector *E*[n(*t*+∆*t*)|n(*t*)] = Tn(*t*). Everything else would be mathematically surprising.

In fact, there are some old papers by Fox and some more recent by Osorio (I'm not exactly sure of the name and I don't remember the years) that discuss that approximation and its shortcomings – perhaps that is the source of bias?

We would be really interested to discuss the mentioned papers but without further comments we are not able to find them. We agree that the approximations are the origin of the bias.

Whether that prediction is then corrected, as in the Kalman filter approach, with a corrector step is irrelevant, as far as the rate equations.

It is true that in-between the correction steps the solution of the RE is used to predict the future mean state and similarly a matrix expression is used to deterministically predict the future covariance. But the defining stochastic perturbation of Equation 3 allows to go beyond a deterministic mean evolution (RE approach) of the system. This is exactly induced by the “correction step”. Therefore, we believe it is appropriate to call Milescu 2005 a deterministic RE approach. On a further note, if one would be only interested in solving the dynamic equations without relation to experimental data, we agree that the difference between REs with added variance and Equation 3 would be one of wording; it is, however, the combination of the stochastic dynamic model with the rigorous likelihood computation via the correction steps which makes the Kalman filter perform differently from RE approaches when applied to data.

Furthermore, it's all a matter of degree. For example, the Milescu et al. approach, which is classified here under the RE umbrella, predicts future states as a mean + variance (or as a mean only), but only using the initial state. There is no correction at each point, as with the Kalman filter method, but there is a correction of the initial state from one iteration to the next (by the way, it's not clear if you implemented that feature here, which would make a big difference). Then, because it considers the stochastical aspect of the data, the Milescu et al. approach should also be considered a stochastic method, just one that doesn't use all the information contained in the data (and so it is fast).

To our understanding we are consistent with the common terminology when distinguishing deterministic (Milescu, 2005) and stochastic modelling (Moffat, 2007) in that way: An illustration of this distinction can be seen in Figure 2. panel A and B in Moffatt, 2007: The RE approach (panel A in Moffatt, 2007) creates a mean signal which is continuous and derivatives can be taken. Panel B and C are only continuous but one cannot take the derivative of the mean signal (at least not in the strict sense of a mathematical limit), which is one central mathematical problem of stochastic methods. In particular that is true for stochastic differential equations which therefore are usually represented in an integrated form such as Equation 3. We like to state that we do appreciate the work of (Milescu, 2005) and (Moffat, 2007) because it is the basis for our stochastic extension elaborated herein. We only disagree with the suggested wording. The following refers to the statement in brackets: We like to draw the Reviewer’s attention to two instances of the article where we state how initialisation was done:

1. Directly after Equation 8 line 224 (which is maybe a bit indirect).

2. After Equation 59 which is explicit.

Similar for the RE approach the equilibrium assumption is used as described in (Milescu, 2005).

Imagine a situation where you ran the same stimulation protocol multiple times, but each time you record only one data point, further and further away from the beginning of the stimulation protocol. A "stochastic" algorithm applied to this type of data would be exactly as described in Milescu et al.

Indeed, in this case the deterministic RE approach (Milescu, 2005) derives a solution which is identical to the solution of the master equation of a closed system if the initial conditions are multinomial [6]. We agree that Milescu, 2005 is a sound approximation of a stochastic process, however, it is not stochastic modeling itself (to our understanding of the terminology), even if the derived likelihood is identical in the mentioned example to solving the master equation.

Of course, in reality all points are recorded in sequence, but that doesn't mean that the approach is not stochastic, just that it's simplifying and discarding some information to gain speed. All methods (such as the Kalman filter described here) make some compromises to reduce complexity and increase speed.The bottom line is that there is the most basic approach of solving the rate equations deterministically, without considering any variance whatsoever, and then there is everything else.

All methods make compromises and so does the Bayesian filter. Using the word deterministic simply points to the details of the assumptions which are adopted to calculate a likelihood, given the terminology that we chose.

58: "Thus, a time trace with a finite number of channels contains, strictly speaking, only one independent data point." I don't understand this sentence. Could you please clarify?

This is exactly meant in the way the Reviewer constructed his example. The RE plus the multinomial assumption is exact if one uses only one data point per relaxation because implicitly (Milescu, 2005) models each data point as an independent draw from a multinomial distribution (plus the experimental noise) whose probability vector evolves in time governed by the rate equation. But in the common perception that the dynamics of the ion channel can be described by a first order Markov process, every following data point has some conditional dependency on the first one. Thus, every following data point is statistically not independent.

74: "The KF is the minimal variance filter Anderson and Moore (2012). Thus, instead of excessive analog filtering of currents with the inevitable signal distortions Silberberg and Magleby (1993) one can apply the KF with higher analysing frequency on less filtered data."Yes, but I'm not sure why should we use excessive filtering? Where is this idea coming from?

We thank the Reviewer for his careful reading. We changed the sentence to “The KF is the minimal variance filter [**?**]. Instead of strong analog filtering of currents to reduce the noise, but with the inevitable signal distortions (Silberberg and Magleby (1993)), we suggest to apply the KF with higher analyzing frequency on minimally filtered data.”

131: "However, even with simulated data of unrealistic high numbers of channels per patch, the KF outperforms the deterministic approach in estimating the model parameters."It is clearly true from your figures, but please give a number as to what is unrealistic (10,000? 100,000?). Also, outperforms, but by how much? As I commented above and below, all methods tested here seem to produce good estimates under the conditions they were designed for (and even outside), and one might argue that it's not worth adding the additional computational complexity and running time for a possibly small increase in accuracy. How is one to judge that increase in accuracy?

We added a “more than a couple of thousands within one patch” to specify typical channels numbers among patches. In our opinion, evaluating the performance of an algorithm by one number (in terms of accuracy only) does not give all the relevant information. There is the relative distance to the true values which the Reviewer has his focus on. But there is also the problem with the uncertain quantification of the RE (Figure 4,5,6,8,9) approach and with the fact that the rate equation approach sometimes does not identify all parameters (Figure 9) which can be identified by the Bayesian filter. All of these aspects are central points of this article and are lumped together in the word “outperforms”. We do not see how to break this down to a simple number.

To be less vague, we added a sentence at the end of the introduction saying which aspects of performance are considered ”We consider the performance of our algorithm against the gold standards in four different aspects: (I) The relative distance of the posterior to the true values, (II) the uncertainty quantification, here in the form of the shape of the posterior, (III) parameter identifiability, and (IV) robustness against typical misspecifications of the likelihood (such as ignoring that currents are filtered or that the integration time of fluorescence data points is finite) of real experimental data.”

276: Has the RE method been used with the mean + variance or mean only?

We used the (Milescu, 2005) algorithm with the variance information.

Also, how was the initial probability vector calculated with the RE method? If you used the mean + variance, then (as mentioned above), you can't call it deterministic. If you used only the mean, then it is indeed a deterministic method, but it's not the approach described in Milescu et al. Please explain.

The publication (Milescu, 2005) describes how to initialize the time evolution by making use of the equilibrium assumption over all states. In the same way, we solve the transition matrix for its equilibrium vector (explained after Equation 59). For the RE approach the probability vector is evolved in time whereas for the Bayesian filter both the mean vector and the covariance matrix of the initial distribution are evolved in time.

To our understanding the variance is a deterministic parameter describing a stochastic process, but is not stochastic itself. Its definition is based on a mean value(1)var(y)=E[(y−E[y])2]=∑y(y−E[y])2p(y).

To the best of our knowledge the deterministic differential equation for the evolution of the variance is derived by applying the definition above to the CME [15].

It is the same situation for the initial distribution. The definition(2)E[y]=∑yy∙P(y)

of the mean value which is a deterministic object enables to express the starting probability vector as(3)≡(t0)=1NchE[n(t0)]=∑n(t0)n(t0)∙Pequilibrium(n(t0)).

This equation holds independently of the fact that all investigated algorithms calculate their initial probability vector by Equation 14 of (Milescu, 2005).

We are aware that an estimator of the mean or the variance or the location of the maximum of the likelihood in parameter space are mathematically random variables because they are data-dependent. However, to our understanding of the terminology, that does not necessarily mean that stochastic modeling has been used to define the estimator (of a mean value, a variance or a full likelihood). To exemplify this for the initial probability vector **Ξ**(*t*_0_) both algorithms start with the identical multinomial assumption. But given the current parameter sample the Bayesian filter updates from the multinomial distribution by accounting the first data point *y*(*t*_0_) the first two moments *E*[n(*t*_0_)] and P(*t*_0_) before the two moments are evolved deterministically in time. This update takes explicitly the stochastic nature of the process into account but it is not present in the RE approach. In our opinion that is meant with the distinction between deterministic and stochastic in the terminology.

229: "added, in Equation 9 to the variance … which originates (Equation 38d) from the fact that we do not know the true system state"Which noise are you referring to? The state noise or the measurement noise?

Both noise aspects are influencing this term: P(*t*) is the covariance matrix of the aleatory uncertainty about n(*t*). Thus, the covariance matrix of the prior distribution over n(*t*) before the information of the current data Equation 58-59 has been taken into account (before the correction step). So it is influenced by the combination of both intrinsic and experimental noise sources Equation 38d. Only at the initial step, where the equilibrium distribution of the channel is used, P(*t*) is the covariance matrix of the intrinsic noise alone. To clarify this in the article, we added the sentence “P*_t_* is the covariance of the prior distribution over n(*t*) before the KF took y(*t*) into account” and we used the word “aleatory”.

326: Which are the two algorithms?

We changed “the two algorithms” to “KF and the RE approach”.

278: "Both former algorithms are far away from the true parameter values with their maxima and also fail to cover the true values within reasonable tails of their posterior."I think the authors might have gotten a little carried away when they made this statement. There is no doubt that the Kalman/Bayesian method produces more accurate estimates, but the other two methods (KF and RE) are very reasonable as well. I see estimates that are within 5, 10, or 20% from the true values, for most parameters. This is not "failure" by any stretch of the imagination. Most people would call this quite good, given the nature of the data.

We replaced “far” by “further” to avoid any subjective definition about what is close and what is far. We added a quantification of parameter divergence: “[…] E.g., the relative error of the maximum of the posterior are ∆*k*_21_ ≈ 200% for [1] and ∆*k*_32_ ≈ 240% for [2]. The 4 other parameters including the three equilibrium constants behave less problematic judged by their relative error […]” We also like to direct the attention of the reader to our statement “[…] also fail to cover the true values within reasonable tails of their posterior”, this means that the uncertainty quantification is indeed failing (the posteriors are too narrow). This points to a central aspect of our work (Figures 3, 4, 5, 6, 8, and 9). With this statement, we did not refer to the relative errors which the Reviewer uses to evaluate the figure. To clarify this, we added: “… Additionally, if one does not only judge the performance by the relative distance of maximum (or some other significant point) of the posterior but considers the spread of the posterior as well, it becomes apparent, that the marginal posterior of both former algorithms fail to cover the true values…”

To clarify our evaluation perspective, we added a paragraph at the end of the introduction. We thank the reviewer for inspiring these clarifications.

294: "The small advantage of our algorithm for small … over Moffatt (2007) is due to the fact that we could apply an informative prior in the formulation of the inference problem … by taking advantage of our Bayesian filter. "This is an interesting and important statement. I interpret it as saying that the Bayesian aspect itself makes only a small contribution to the quality of the estimates, when comparing it with the Moffatt method, which is a Kalman filter as well. The only issue with the Moffatt method is the lack of an explicit formalism for the excess state noise (which could presumably be added).

We suggest that this interpretation is based on a misunderstanding. There are two layers of Bayesian statistics in our algorithm and one in the previous Kalman filter. We added the word “generalizations” to make clear that one of these layers of our algorithm uses a similar Bayesian update (in other words prediction-correction) as in (Moffatt, 2007). As argued above, each KF is a Bayesian filter. The derivation of (Moffatt, 2007) is done as a Bayesian version of the forward algorithm. Nevertheless, his implementation of the Kalman filter infers the rate matrix via maximum likelihood. The statement “The only issue with the Moffatt method is the lack of an explicit formalism for the excess state noise (which could presumably be added)” does not reflect that our article demonstrates exactly how to change the calculation of the likelihood of (Moffatt, 2007) to include “excess state noise” (Equation 44-46c) and Poisson noise (Equation 52 57). The central mathematical point is how to circumvent that Equation 8 cannot be solved in the way it was done by Moffatt, 2007, for state-dependent noise (Equation 10-11). Our solution is independent of which statistical framework (either maximum likelihood or Bayesian statistics) is used for the inference of the rate matrix. (Also, our implementation in Stan can be used as a point estimator or to sample the full posterior.)

The conclusion “I interpret it as saying that the Bayesian aspect itself makes only a small contribution to the quality of the estimates” additionally skips, over the fact that the (Moffatt, 2007) algorithm can be seen as the limiting algorithm of ours for *σ*_op_ → 0. The more one goes to the left of Figure 3b the more the two algorithms become identical and thus also their results. In case that the Reviewer meant the Bayesian posterior sampling of the rate matrix etc. we agree. But according to our arguments above, a parameter whose influence becomes less and less important in the algorithm as a whole (going to the left in Figure 3b) can asymptotically be fixed at a maximum likelihood inference, Simply, because also its error becomes more and more irrelevant. But this should not be the case going further to the right in Figure 3b.

Another point of view is that one can see the (Moffatt, 2007) algorithm as a likelihood misspecification for data with state-dependent noise. The more we go to the left in Figure 3b the smaller the misspecification is.

It also suggests to me that any method that tries to use the noise may run into difficulties when the noise model is unknown (typical real-life scenario). The Moffatt method is confronted with unknown noise and it fails. What about when the Bayesian method is confronted with unknown noise? There are some comments in the manuscript, but nothing tangible. Could you please comment?

It depends on what the Reviewer means with unknown. It is not the issue that in Figure 3b (Moffatt, 2007) *σ* and *σ*_op_ are unknown. The problem of (Moffatt, 2007) is that his likelihood is not flexible enough and that there is no structure in the algorithm (Moffatt, 2007) to account for *σ*_op_. Inspired by the question of the Reviewer we added the sentence “Further, (Figure 3b) indicates the importance that the functional form of likelihood is flexible enough to capture the second order statistics of the noise of the data sufficiently.” We observed also for cPCF data the same effect at the early stages of the project when coding mistakes made the conditional likelihood of the experiment flawed.

In general, misspecifications of the likelihood will always lead to sometimes small and sometimes huge deviations in the inferred parameters compared to a fictitious inference with an ideal likelihood and prior. But this holds for both maximum likelihood inference and Bayesian statistics.

The learning of the unknown noise statistics is briefly discussed in Figure 10e. It displays the posterior of *σ*_op_^2^ for PC or cPCF data if only an informative prior on *σ*^2^ is used. Information about the baseline variance is always possible to be characterized prior to the inference of interest. Also Figure 11 only assumes a pre-characterization of *σ* and then *σ*_op_ is inferred from the actual data (even though we did not display *σ*_op_).

Figure 3: There are no data sets in the figure. What data were analyzed here?Is there a typo regarding the colors in a? What are the red, blue, black, green symbols? Please verify.Assuming the red is KF, there is something very curious about its estimates, which are very different in distribution from the Bayesian estimates. Could there be a coding problem? If red is actually RE, then it would make more sense. Could you explain? Is this the effect of the "excessive" open state noise? If so, I find this situation a bit unfair, because then the 2007 KF algorithm is tested with data that it is not meant to work with.

We improved figure 3 and its legend and used the colors more consistently. We display in the new Figure 3d the data which were used to generate the posteriors of panel a and b. Yes, red is our own Bayesian implementation of the previous Kalman filter (Moffatt, 2007) and (blue) is our Bayesian implementation of (Milescu, 2005). To increase clarity, we made the blue colors identical which were not identical in the previous version. It is indeed the effect of the open-channel noise which we state in the main text by “Varying *σ*_op_*/i* reveals the range of the validity (Figure 3b) of the algorithm (red) from Moffatt (2007)”. The figure legend was adapted accordingly.

We like to state further that it was one of the central questions in the first comment by the reviewer to demonstrate that our algorithm is not only an incremental step forward. If we compare the algorithm in a situation where the standard deviation of the excessive open channel *σ*_op_ = 0, our algorithm and the previous Kalman filter (Moffatt, 2007) produce an (almost) identical posterior because the likelihoods of both algorithms are essentially identical (see Methods section). The posteriors are only almost identical because our algorithm allows some uncertainty in *σ*_op_. Figure 3b shows a limitation of (Moffatt, 2007) which is not a limitation of our algorithm.

Also, I think it would be more interesting to have a comparison between the original KF and the newer Bayesian approach, so we can understand what Bayesian estimation does. In any case, I can't really interpret the data until you clarify the colors.

Back to the misunderstanding above: Both Kalman filters are Bayesian filters. Our one is a more flexible version of the previous one. It is the role of Figure 3b to illustrate this comparison. Thus, Figure 3b explains when our generalization is necessary. The question about what advantage the sampling of the posterior of the rate matrix gives vs. a maximum likelihood inference is not in scope here. The sampling of the posterior, if dominated by data and not the prior itself does not improve parameter identification, however, enables a more detailed uncertainty evaluation. As we are using a uniform prior for all parameters the mode of the posterior corresponds to the maximum likelihood inference.

The legend is unclear overall.

We revised the legend.

303: What means "singular"?

We added now also in the main text “meaning that the fisher information matrix is singular (Watanabe, 2007)”.

324: What shaded area in Figure 1a? Perhaps you mean Figure 4a?

We are not sure what the Reviewer means here because the figure reference where the shaded area is mentioned in the main text indeed points to Figure 4a.

Figure 4: I don't see any light green.

We changed the light green to a dotted green line.

368: "The estimated true success rate by the RE approach (blue) is ≈ 0.15 and therefore far away from the true success probability. In contrast, the posterior (green) of the true success probability of the KF resides with a large probability mass between the lower and upper limit of the success probability of an optimal algorithm (given the correct kinetic scheme)."I'm really a bit puzzled here: yes, the KF is more accurate (given the correct kinetic scheme), but what I see in this figure is that both algorithms are biased, yet both are generally within 10% (and quite a bit better for KF) of the true values.

The quoted sentence refers to Figure 4c which is used to introduce a Bayesian way to look at the spread of the probability mass versus the distance to the true value. We disagree with the Reviewer that one can deduce a biased *P*_success_ from the posterior of the binomial test.

Possibly, the Reviewer refers to a different aspect than we do. We assume he was wondering why the observed bias of Figure 4b is not visible in Figure 4c. The answer is because we tested in Figure 4c for 0.95 probability mass which, even with the bias included, covers the true values according to the binomial interpretation.

We plotted Figure 4c to have a gentle introduction to the concept. The final solutions are discussed in Figures 5 and 6. We assume that the Reviewer expected to see that binomial statistics do not match the data that well. But Figure 4c is not the optimal figure to detect this because the probability mass is chosen too big. But one can see this in Figure 5c because here we plot success vs. probability mass. And indeed, one might discover the bias again (if one knows that it is there) in the slight under-performance of the Bayesian filter for probability masses *<* 0*.*5. Note, that these plots are for uncertainty quantification but are not optimally suited for bias detection because the direction of the bias is lost, and all data sets are lumped together in one binomial test. They test if the posterior (its highest density volumes) holds frequentist properties.

If one would plot the log of the rates, to transform into δ Gs, the differences would be even smaller. I would bet that experimental artifacts would contort the estimates to a greater degree anyway.

Indeed, the ratio of the accuracies is small, but it is the essential message of Figure 4 that even if the accuracies are almost the same the uncertainty quantification of the RE approach is defective.

Of course, depending on the experimental setting, experimental artifacts can mess up everything if they are not correctly accounted for.

It's very nice that the RE approach was embedded and tested within a Bayesian framework, but it would still be interesting to know what is the contribution of the Bayesian aspect (unless I missed this point in the manuscript).

As mentioned above the posterior sampling itself (when using uniform priors) does not influence the accuracy of the parameter estimates. The mode of the posterior equals the maximum of the likelihood. Using more elaborate priors is not in the scope of the paper.

436: "…while their error ratio (Figure 7a) seems to have no trend to diminish with increasing NCh".This would be unexpected, because the Kalman filter will always use more information than RE. Why would the KF approach become relatively more erroneous at higher Nc? I don't see any reason.

Indeed, this was an awkward use of words. What we like to say is that the error ratio does not approximate 1. The error of the RE approach does not seem to converge to the error of the KF. There is obviously no reason to assume that the Bayesian filter becomes more erroneous than the RE. We changed the sentence to “to have no trend to approach 1 with increasing *N*_ch_”. In fact, it is exactly our conclusion to say that the Kalman filter will always use more information (see lines 467-773).

To me, an important question is when do the overall errors become dominated by external factors, such as recording artifacts, using the wrong model, etc.

Of course, there might be experimental situations where these problems are not the major concern. So, there might be other opportune choices for algorithms but then there is also the possibility to adapt the algorithm for those unwanted perturbations (likelihood misspecifications) of the signal. In Figures 11 and 12 we show some aspects of typical problems in realistic experimental data. One can actually see the whole article as a discussion of various aspects of likelihood misspecifications.

511: "Thus, transferring the unusually strictly applied algebraic constraint Salari et al. (2018) of microscopic-reversibility to constraint with scalable softness we can model the lack information if microscopic-reversibility is exactly fulfilled Colquhoun et al. (2004) by the given ion channel instead of forcing it upon the model."I'm not sure I understand what you mean here. I think it is very usual to constrain microscopic reversibility EXACTLY, whether through the SVD method (Qin et al., Plested et al., Milescu et al., Salari et al), or through some other algebraic method (Plested et al).

Thanks for the careful reading to detect this typo. It indeed changed the meaning of the sentence. We changed the sentence to “Thus, we can transfer the usually strictly applied algebraic constraint (Salari,2018) of microscopic reversibility to a constraint with scalable softness. In that way we can model the lack of information if microscopic-reversibility is exactly fulfilled (Colquhoun,2004impose) by the given ion channel instead of enforcing the strict constraint upon the model.”

It's not "forcing" it on the channel, but simply testing the data for that condition. Of course, one could test the condition where there is no reversibility enforced at all (i.e., it's "given by the channel"), or anything in between.

To be more clear we changed the part of the sentence to “enforcing the strict constraint”.

To the best of our knowledge, often microscopic reversibility is used to constrain the parameter space such that the inference algorithm converges to some global optimum rather than having some unidentified or marginally identified parameters (which does not mean that the constraint is actually fulfilled in reality). In this case it makes sense to model the uncertainty as a hierarchical Bayesian model because the data sets will likely be vague about the strict microscopic reversibility hypothesis. The Bayesian framework is a good way to model this uncertainty because the sampler automatically marginalizes over this hierarchical structure in the model which is induced by the uncertainty. Maximum likelihood or maximum posterior methods have to integrate this uncertainty analytically which is not always possible. If the data are ambiguous about microscopic reversibility, the posterior of the model is going to have the ambiguity of the prior and the data, too.

675: "With our algorithm we demonstrate (Figure 3c and 7) that the common assumption that for large ensembles of ion channels simpler deterministic modeling by RE approaches is on par with stochastic modeling, such as a KF, is wrong in terms of Euclidean error and uncertainty quantification (Figure 5a-c and Figure 6a-b)".I find this statement a little subjective. I think anyone who cares about stochastic vs. deterministic modeling knows enough that any method that uses more information from data should produce better estimates, regardless of the number of channels. I would say that the more likely assumption is that deterministic estimators produce poor estimates with small numbers of channels, perfect estimates with infinitely many channels, and anything in between. In fact, the KF method behaves exactly the same way, just with overall higher accuracy.

We think here it is matter of perspective. If one’s concern is accuracy alone than we agree. We do not doubt that the rate equation is accurate for an infinite amount of ion channels.

But for an inference algorithm we advocate that increasing accuracy is necessary but not sufficient. We meant with line 675 that *“uncertainty quantification”* is flawed for the RE approach independently of the amount of channels. Or at least inside the large tested ranges. This does not seem trivial to us and the summary of the Reviewer “In fact, the KF method behaves exactly the same way, just with overall higher accuracy*”* seems not to reflect this. Sometimes it might be opportune to not care about uncertainty quantification but for a full Bayesian analysis that does not make sense. To the best of our knowledge correct uncertainty quantification is the merit or origin of all merits of Bayesian statistics.

Looking at the tests in this manuscript, I would say that all the previous studies that modeled macroscopic data using deterministic methods are safe. They don't need to be redone. The future, of course, is a different matter.

We agree that previous analysis of macroscopic currents were essential for valuable insight into channel physiology. Still our results suggest that more information can be gained from the data and that possibly some previous error estimates should not be relied on too heavily. We hope that our contribution will move the field forward.

References

Milescu, Lorin S., Gustav Akk, and Frederick Sachs. (2005) Maximum likelihood estimation of ion channel kinetics from macroscopic currents*.*,Moffatt, Luciano (2007) Estimation of ion channel kinetics from fluctuations of macroscopic currents*.*,Daniel T. Gillespie (1979) Approximating the master equation by Fokker-

Planck-type equations for single variable chemical systems,

Gillespie, Daniel T The chemical Langevin equation*. (2000)*,Gillespie, Daniel T. Stochastic simulation of chemical kinetics*.* Annu. Rev. Phys. Chem. 58 (2007): 35-55,Tobias Jahnke·Wilhelm Huisinga (2007) Solving the chemical master equation for monomolecular reaction systems analytically,Aleksandra M. Walczak Andrew Mugler Chris H. Wiggins Analytic methods for modeling stochastic regulatory networks (2012),Rubin, Donald B., and Nathaniel Schenker (1986) Efficiently simulating the coverage properties of interval estimates*.*,Van der Vaart, A. W. (1998) Asymptotics Statistics *(1998)* Cambridge series in statistical and probabilistic mathematics.”,Martin, Ryan, and Bo Ning. Sankhya A Empirical priors and coverage of posterior credible sets in a sparse normal mean model. ,Gardiner, Crispin W. Handbook of stochastic methods,Wallace, E. W. J., et al. Linear noise approximation is valid over limited times for any chemical system that is sufficiently large*.*,Wallace, Edward WJ. A simplified derivation of the linear noise approximation*.*Grima, Ramon. Linear-noise approximation and the chemical master equation agree up to second-order moments for a class of chemical systems*.* Physical Review E 92.4 (2015): 042124.Munsky, Brian, Brooke Trinh, and Mustafa Khammash. Listening to the noise: random fluctuations reveal gene network parameters*.*